JCB Journal of Cell Biology

**TOOLS**

# ILEE: Algorithms and toolbox for unguided and accurate quantitative analysis of cytoskeletal images

Pai Li[1,2]*, Ze Zhang[3]*, Yiying Tong[3], Bardees M. Foda[4,5], and Brad Day[1]

The eukaryotic cytoskeleton plays essential roles in cell signaling and trafficking, broadly associated with immunity and diseases in humans and plants. To date, most studies describing cytoskeleton dynamics and function rely on qualitative/quantitative analyses of cytoskeletal images. While state-of-the-art, these approaches face general challenges: the diversity among filaments causes considerable inaccuracy, and the widely adopted image projection leads to bias and information loss. To solve these issues, we developed the Implicit Laplacian of Enhanced Edge (ILEE), an unguided, high-performance approach for 2D/3D-based quantification of cytoskeletal status and organization. Using ILEE, we constructed a Python library to enable automated cytoskeletal image analysis, providing biologically interpretable indices measuring the density, bundling, segmentation, branching, and directionality of the cytoskeleton. Our data demonstrated that ILEE resolves the defects of traditional approaches, enables the detection of novel cytoskeletal features, and yields data with superior accuracy, stability, and robustness. The ILEE toolbox is available for public use through PyPI and Google Colab.

## Introduction

Higher eukaryotes have evolved a complex suite of cellular signaling mechanisms to regulate many biological processes, such as growth, development, movement, reproduction, and response to environmental stimuli. Not surprisingly, to maintain the integration and sustainability of a conceptual signaling network, another "physical network" is deployed as the framework to organize all subcellular structures and the conduit to transport molecules and information (Lian et al., 2021)—namely, the cytoskeleton. There are four basic types of eukaryotic cytoskeletal elements: actin, microtubule, intermediate filament, and septin. Shared by plants and animals are actin and microtubule, which can be generalized as a cytoplasmic web-like matrix, physically integrating the plasma membrane, vesicles, and organelles, and functionally connecting intercellular signaling to the extracellular environments through a highly dynamic series of temporally and spatially regulated changes of architecture (Blanchoin et al., 2014; Brouhard, 2018). However, structurally complete and functionally identical intermediate filaments or septin have not yet been identified in plants (Utsunomiya et al., 2020; Onishi and Pringle, 2016). Altogether, the cytoskeleton controls numerous cellular processes such as movement, shaping, cellular trafficking, and intercellular communication (Nick, 2011). It also provides the mechanical force required for chromosome separation and plasma membrane division during mitosis and meiosis (Carlton et al., 2020). In addition to its role within the cytoplasm, the cytoskeleton also participates in various processes within the nucleus, including RNA polymerase recruitment, transcription initiation, and chromosome scaffolding (Kristó et al., 2016).

As the cytoskeleton is vital for cell activities and life, the dysfunction of cytoskeletal dynamics generally leads to severe diseases. For example, in the development of Alzheimer's disease, amyloid development triggers the dysregulation of actin dynamics within dendritic spines, leading to synaptotoxicity (Rush et al., 2018). Similarly, the dysfunction of microtubule dynamics can trigger neuropsychiatric disorders (Marchisella et al., 2016). In breast cancer, the cancer cell can deploy aberrant actin aggregation to resist cytotoxic natural killer (NK) cells from the immune system (Al Absi et al., 2018). In the case of plant diseases, the function of the cytoskeleton is similarly required. Recent work has demonstrated that the dynamics of the actin and microtubule of the host are specifically manipulated by pathogens to paralyze plant immunity during an infection (Li and Day, 2019). Indeed, the eukaryotic cytoskeleton not only serves as an integrated cell signaling and transportation platform but it is also "a focus of disease development" at the

[1]Department of Plant, Soil and Microbial Sciences, Michigan State University, East Lansing, MI; [2]Department of Plant Biology, Michigan State University, East Lansing, MI; [3]Department of Computer Science and Engineering, Michigan State University, East Lansing, MI; [4]Department of Pharmacology and Toxicology, Michigan State University, East Lansing, MI; [5]Molecular Genetics and Enzymology Department, National Research Centre, Dokki, Egypt.

*P. Li and Z. Zhang contributed equally to this paper. Correspondence to Pai Li: lipai@msu.edu

A preprint of this paper was posted in bioRxiv on February 25, 2022.

molecular level for humans and plants. Hence, understanding the architecture and dynamics of the cytoskeleton is broadly associated with life, health, and food security, attracting significant interest across different fields of biology and beyond.

Over the past several decades, confocal microscopy-based methods using fluorescent markers have been developed to monitor changes in the cytoskeletal organization (Melak et al., 2017). While showing advantages in real-time observation and offering an intuitive visual presentation, these approaches possess critical limitations—namely, they are subject to human interpretation and, as a result, often suffer from bias. As a step to reduce such limitation(s), the emergence of computational, algorithm-based analyses offers a solution to describe the quantitative features of cytoskeletal architecture. However, while previous studies introduced the concept of using generalizable image processing pipelines (Lichtenstein et al., 2003; Shah et al., 2005) to transfer the task of evaluation away from humans to computer-based quantitative indices, several key bottlenecks were still prevalent. First, most previous quantitative algorithms are limited to 2D images. As a result, these approaches require users to manually generate z-axis projections from raw data, a process that results in an incredible amount of information loss, especially within the perpendicular portion of the cytoskeleton. Second, many current approaches require users to set thresholds manually to segment cytoskeletal components from images, a task that results in a sampling bias. Lastly, the performance of existing algorithms varies significantly depending on the sample source. This hurdle imposes a considerable disparity in the algorithm performance for plants—which possess a dominance of curvy and spherically distributed filaments—compared to the animal cytoskeletal organization, which is generally straight and complanate (Liu et al., 2020, 2018; Alioscha-Perez et al., 2016). In fact, while the sample source dramatically impacts the ability to evaluate features of cytoskeletal function across all eukaryotes, most current approaches are developed based on cytoskeletal images from animal cells, indicating potential systemic bias when applied to other types of image samples, such as plants.

Previous work has described the development of a global-thresholding-based pipeline to define and evaluate two key features of cytoskeleton filament organization in living plant cells: cytoskeletal density, defined by occupancy, and bundling, defined by statistical skewness of fluorescence (Higaki et al., 2010). While this approach utilizes manual global thresholding (MGT), which can potentially introduce a certain level of user bias, it still outperforms most standardized adaptive/automatic global or local thresholding techniques, such as Otsu (Otsu, 1979) and Niblack (Niblack, 1985) methods. More recent advances in MGT-based approaches include the introduction of the coefficient of variation (CV) of fluorescence as an index to measure the degree of filament bundling, which serves as an improvement, substituting skewness with advanced overall robustness and utility of the original algorithm (Higaki et al., 2020). However, this analysis pipeline still consumes a considerable amount of time and effort from users for massive sample processing and leaves unaddressed two critical issues of rigor in image processing and analysis: information loss and human bias.

If the biological question of a study does not necessarily require an address of the thickness of the cytoskeleton, another strategy is to directly obtain the skeletonized image (thinned filament, with 1-pixel width), while bypassing the segmentation of entire filaments. Previous successful approaches utilized a linear feature detector (Gan et al., 2016) or a steerable filter for ridge recognition (Kittisopikul et al., 2020) to extract the skeleton information from 2D datasets, which enabled the automated computation of cytoskeleton orientation/directionality and potentially some additional features (e.g., topology). However, such strategies face potential challenges if filament thickness or other features depending on the binary image of the entire filament (e.g., occupancy) is required. While the linear feature detector, as aforementioned, may not be appropriate for plant samples with a curvy cytoskeleton, the use of steerable filters (Jacob and Unser, 2004) actually arouses significant inspiration—instead of ridges, edges can potentially define the entire filament (area within the edge) for a complete cytoskeleton analysis algorithm. However, if such filters were directly applied to edges, the obtained first-/second-order derivatives would still require subjective human input for thresholding. Additionally, it is also challenging to categorize multiple pieces of edges to one filament since edges may be separated or linked by fault because of noise and filament diversity of the samples. Therefore, how to properly utilize gradient information becomes a primary concern for developing a high-performance cytoskeleton analysis algorithm.

In this study, we developed the Implicit Laplacian of Enhanced Edge (ILEE), a 2D/3D compatible unguided local thresholding algorithm for cytoskeletal segmentation and analysis based on native brightness, first-order derivative (i.e., gradient), and second-order derivative (i.e., Laplacian) of the cytoskeleton image altogether (see Fig. 1). The study herein supports ILEE as a superior image quantitative analytic platform that overcomes current limitations such as information loss through dimensional reduction, human bias, and intersample instability; ILEE can accurately recognize cytoskeleton from images with a high variation of filament brightness and thickness, such as those of live plant samples.

As a key advancement in the development of ILEE, we constructed an ILEE-based Python library, namely ILEE_CSK, for the fully automated quantitative analysis of cytoskeleton images. To improve ILEE_CSK's capability to explore the features of the cytoskeleton, we proposed several novel indices—linear density, diameter_TDT, diameter_STD, segment density, and static branching activity to enable/enhance the measurement of (de-)/polymerization, bundling, severing-and-nucleation, and branching dynamics of the cytoskeleton. Together with other classic indices transplanted from previous studies, ILEE_CSK supports 12 cytoskeletal indices within five primary classes: density, bundling, connectivity, branching, and directionality. Our data suggested that ILEE outperforms other classic algorithms by its superior accuracy, stability, and robustness for the computation of cytoskeleton indices. Furthermore, we provided evidence demonstrating higher fidelity and reliability of 3D-based cytoskeletal computational approaches over traditional 2D-based approaches. In addition, using a series of experiment-based images from

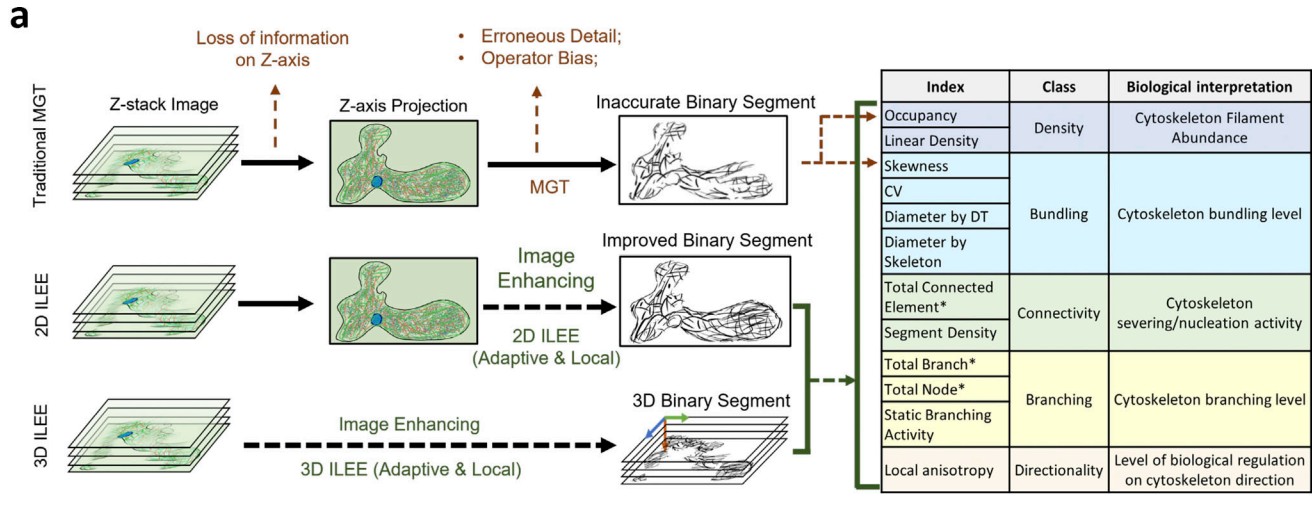

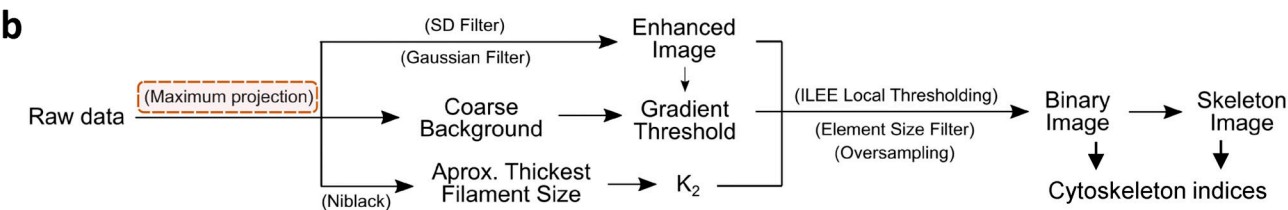

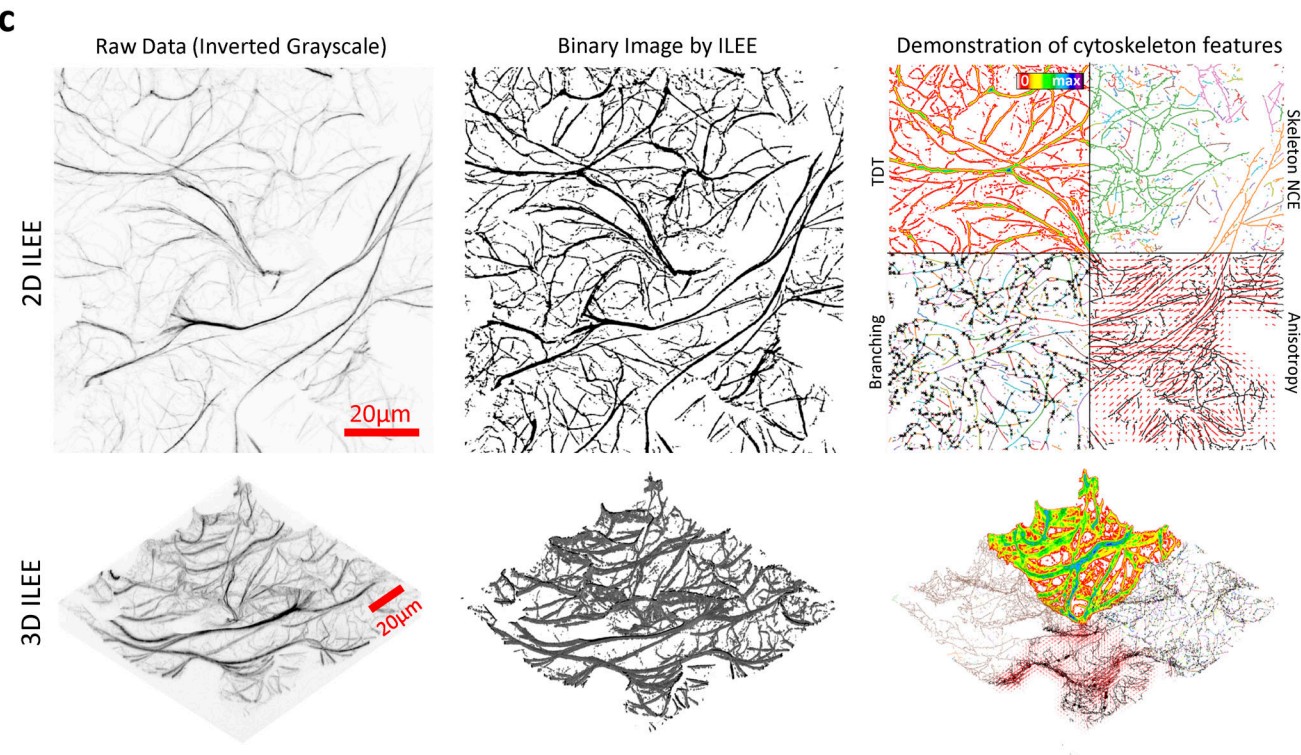

Figure 1.  **ILEE pipeline and the demonstration of cytoskeletal indices. (a)** ILEE is an adaptive local thresholding approach that applies to both 2D and 3D data structures, with an output of 12 cytoskeletal indices. Please note that total connected element, total branch, and total node (marked "*") are non-normalized indices for development purposes of advanced users and cannot be directly used for biological interpretation. **(b)** Schematic diagram of the ILEE algorithm. ILEE requires a sample image with enhanced edge gradients, a computed gradient threshold, and an implicit Laplacian smoothing coefficient, $K_2$, to generate a binary image and skeletonized image for index computation. Z-axis maximum projection (red box) is only conducted in the 2D mode. **(c)** Visualized demonstration of ILEE performance. The raw data, binary image generated by ILEE, and demonstration of computed features by both 2D and 3D modes are shown. TDT, total distance transformation map, used to compute bundling indices; skeleton NCE, non-connected element (filament pieces) of the skeleton

image with each element by different colors, used to estimate connectivity indices; branching, the skeleton image with each branch in different colors and each node by a black cross; anisotropy, local anisotropy level, shown as length of red lines, and direction of the first eigenvector, shown as the direction of red lines. Scale bars are marked as red solid line.

pathogen-infected plant cells, we demonstrated that ILEE has an improved sensitivity to distinguish biological differences and the potential to reveal novel biological features previously omitted due to insufficient approaches. In sum, this platform not only enables the efficient acquisition and evaluation of key cytoskeleton filament parameters with high accuracy from both projected 2D and native 3D images but also enables accessibility to a broader range of cytoskeletal status for biological interpretation.

The library, ILEE_CSK, is publicly released on GitHub (https://phylars.github.io/ILEE_CSK/). We also developed the ILEE Google Colab pipelines for data processing, visualization, and statistical analysis, which is a convenient, user-friendly, crossplatform interface requiring no programming experience.

## Results

### The ILEE pipeline

Raw images from a confocal laser scanning microscope are obtained from the sensor detection of in-focus photons from a fixed focal plane. Since the cytoskeleton is a 3D structure that permeates throughout the cell, current approaches used to capture the filament architecture rely on scanning pixels of each plane along the z-axis at a given depth of step; finally, these stacks are reconstructed to yield a 3D image. However, due to limited computational biological resources (e.g., imaging conditions with low signal-to-noise ratio, lack of suitable algorithms and tools, and insufficient computational power), most studies have exclusively employed the z-axis-projected 2D image, which results in substantial information loss and systemic bias in downstream analyses.

In our newly developed algorithm, we integrated both 2D and 3D data structures into the same processing pipeline to ameliorate the aforementioned conflict (Fig. 1 a). In short, this pipeline enabled automatic processing of both traditional 2D and native 3D z-stack image analysis. As shown in Fig. 1 b, cytoskeleton segmentation using ILEE requires three inputs: an edge-enhanced image, a global gradient threshold that recognizes the edges of potential cytoskeletal components, and the Laplacian smoothing coefficient $K$ (described below). With these inputs, a local adaptive threshold image is generated via ILEE, and the pixels/voxels with values above the threshold image are classified as cytoskeletal components, which generate a binary image (Fig. 1 c). The binary image is further skeletonized (Lee et al., 1994) to enable the downstream calculations of numerous cytoskeleton indices, the sum of which comprises the quantitative features of cytoskeletal dynamics (Fig. 1 c). Additionally, because the 2D and 3D modes share a common workflow, all of the calculated cytoskeleton indices also share the definition for both dimensions. This feature enables a horizontal comparison of both modes by the user, which we assert will significantly contribute to the community involvement and production of large image datasets for further examination and development.

In general, the ultimate goal of this approach and the resultant algorithm is to construct a pipeline that enables the automated detection of the eukaryotic cytoskeleton from complex biological images in an accurate and unbiased manner.

### Image decomposition and processing strategy

One of the central problems of automated cytoskeletal image processing is how to accurately recognize cytoskeletal components—a task that is highly challenging because object pixels (i.e., cytoskeleton components) generally have a high dynamic range of signal intensity within and among individual samples due to varied bundle thickness, the concentration of the fluorescent dye, and its binding efficiency. As a framework to further understand this challenge, an image from confocal microscopy is conceptually comprised of three components: (1) true fluorescence, which is emitted by the dye molecules within the pixel, (2) the diffraction signal transmitted from neighboring space, and (3) the ground noise generated by the sensor of the imaging system (Fig. 2 a). During confocal imaging, the ground noise fluctuates around a constant due to the fixed setting of photon sensors, while the diffraction signal is positively correlated with the actual local fluorescence. Therefore, an ideal cytoskeleton segregation algorithm should be an adaptive local thresholding approach that refers to both ground noise and local signal intensity.

### Identification of coarse background

To identify the approximate level of the ground noise for downstream fine thresholding, a general strategy is to define a highly representative "feature value" of ground noise, based on which a global threshold can be estimated to segment the "coarse background." Guided by this strategy, we first tested a method to directly use the peak-of-frequency of the image brightness as the representative value. Unfortunately, this approach is not reliable because the brightness distribution near the theoretical peak is always turbulent and non-monotone (Fig. S1 a), which makes the peak inaccurate and unmeasurable. However, during our experiments, we noticed an interesting phenomenon whereby global thresholding always rendered the maximum number of segmentations of the background approximately at the peak-of-frequency of ground noise. This is because individual pixels in ground noise are subject to a normal-like distribution; therefore, two adjacent pixels would likely fall into different binary categories when the threshold equals the mean (peak) of the distribution. Inspired by this discovery, we designed a different algorithm that determines the coarse background based on the morphological feature of the ground noise—namely, non-connected negative element scanning (NNES; Fig. 2 b).

In brief, NNES scans the total number of non-connected negative elements (NNE) at different thresholds to identify one that results in the maximum count of NNE (i.e., the peak;

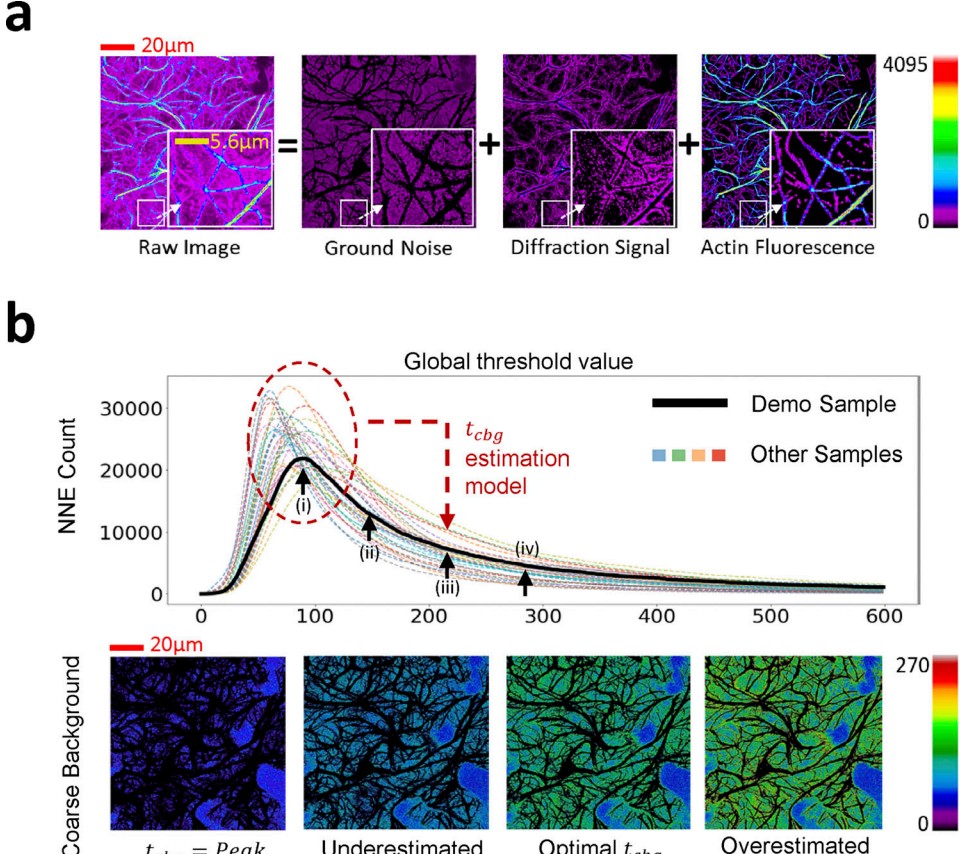

Figure 2. **NNES global thresholding. (a)** The conceptual decomposition of a confocal fluorescence image of the cytoskeleton. An Arabidopsis leaf confocal microscopic image of actin, as an example of the eukaryotic cytoskeleton, comprises of three components: ground noise, the mechanical noise of the sensor regardless of the true fluorescence signal, diffraction light, the unavoidable diffraction signal of fluorescence component around, and true actin signal. They correspond to the noise filtered by coarse background, noise additionally filtered by ILEE, and segmented actin components in the algorithm. **(b)** The scheme and demonstration of NNES. The curve reflects the NNE (negative non-connected component) count when certain global thresholding is applied to the raw images of 30 randomly selected samples. Their extremely smooth shape makes it easy to detect the peak as a feature value as the input for the coarse background estimation model. (i–iv) The demonstration of the filtered background at positions (i), (ii), (iii), and (iv) are shown above, where (iii) is adopted. The black area surrounded by colored area is the foreground information to be further processed. Scale bars are marked as red/yellow solid line.

Fig. 2 b i) to be used as the representative value. Our analysis suggested that NNES generates a curve with a smooth, easily measurable maximum, which is ideal for representative value identification (Fig. S1, b and c). Next, the global threshold for the coarse background (Fig. 2 b iii) will be determined using a linear model trained by the representative value rendered by NNES and manual global thresholding (MGT), a global threshold determined by researchers experienced in manual cytoskeleton image analysis (Figs. S1 and S2). Generally, NNES maintains stability and accuracy over different samples that vary in brightness distribution. Therefore, the global threshold identified through NNES and the estimation model is used in this study to define the coarse background area for downstream processing. Additionally, we further improved the 3D coarse background estimation model by introducing another published dataset (Cao et al., 2022) with image augmentation for diverse ground noise pattern (Fig. S3). In total, these improvements serve to further extend the applicability and reliability of the ILEE_CSK library beyond this study.

**Cytoskeleton segmentation by ILEE**

The core strategy of ILEE is to accurately identify the edges of all cytoskeletal filament components and apply an implicit Laplacian smoothing (Desbrun et al., 1999) on only the edges. This leads to the generation of a threshold image where high gradient areas (i.e., the edges of cytoskeleton filaments) smoothly transition into each other (Fig. 3 a). As illustrated in Fig. S4 a, the general local threshold trend changes as a function of the baseline values of the cytoskeleton edges. This is because ILEE selectively filters out high-frequency noise while preserving salient geometric features of individual actin filaments by leveraging the spectral characteristics of Laplacian operators (Fig. S4, b and c). Thus, ILEE can filter the background regardless of the general local brightness level, demonstrating that its function does not require an operating kernel, which would otherwise restrict the performance when evaluating filaments of varying thickness. Moreover, the edges of cytoskeletal components can be smoothed and elongated using a significant difference filter (SDF; Fig. S5) and a Gaussian filter, the sum of

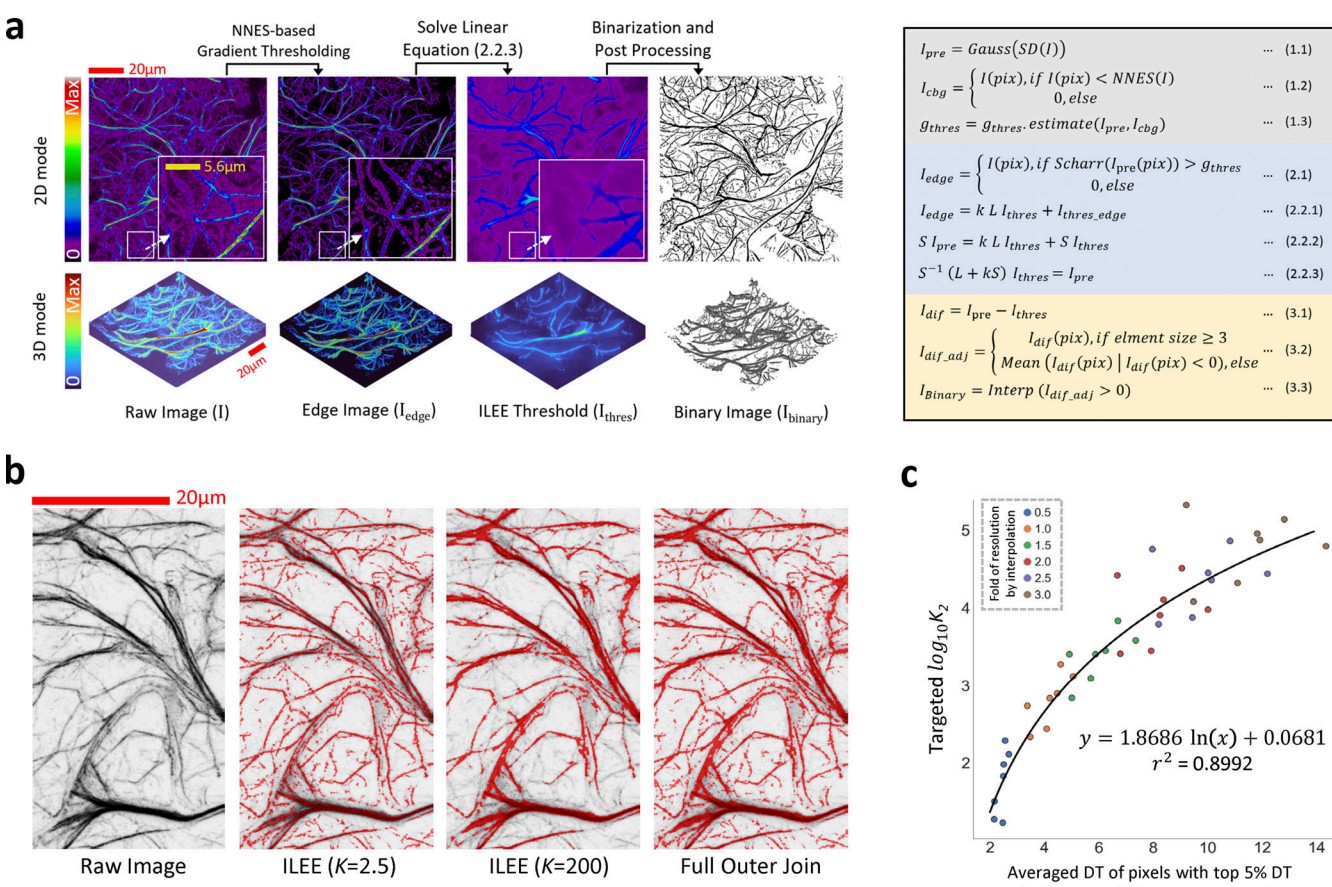

**a**

NNES-based Gradient Thresholding → Solve Linear Equation (2.2.3) → Binarization and Post Processing

2D mode / 3D mode

Raw Image (I) — Edge Image ($I_{edge}$) — ILEE Threshold ($I_{thres}$) — Binary Image ($I_{binary}$)

$$I_{pre} = Gauss(SD(I)) \qquad \ldots \text{(1.1)}$$

$$I_{cbg} = \begin{cases} I(pix), \text{if } I(pix) < NNES(I) \\ 0, else \end{cases} \qquad \ldots \text{(1.2)}$$

$$g_{thres} = g_{thres}. \, estimate(I_{pre}, I_{cbg}) \qquad \ldots \text{(1.3)}$$

$$I_{edge} = \begin{cases} I(pix), \text{if } Scharr(I_{pre}(pix)) > g_{thres} \\ 0, else \end{cases} \qquad \ldots \text{(2.1)}$$

$$I_{edge} = k \, L \, I_{thres} + I_{thres\_edge} \qquad \ldots \text{(2.2.1)}$$

$$S \, I_{pre} = k \, L \, I_{thres} + S \, I_{thres} \qquad \ldots \text{(2.2.2)}$$

$$S^{-1} \, (L + kS) \, I_{thres} = I_{pre} \qquad \ldots \text{(2.2.3)}$$

$$I_{dif} = I_{pre} - I_{thres} \qquad \ldots \text{(3.1)}$$

$$I_{dif\_adj} = \begin{cases} I_{dif}(pix), \text{if element size} \geq 3 \\ Mean \, (I_{dif}(pix) \mid I_{dif}(pix) < 0), else \end{cases} \qquad \ldots \text{(3.2)}$$

$$I_{Binary} = Interp \, (I_{dif\_adj} > 0) \qquad \ldots \text{(3.3)}$$

**b**

Raw Image — ILEE ($K$=2.5) — ILEE ($K$=200) — Full Outer Join

**c**

Fold of resolution by interpolation: 0.5, 1.0, 1.5, 2.0, 2.5, 3.0

Targeted $log_{10}K_2$ (y-axis)

$$y = 1.8686 \ln(x) + 0.0681$$
$$r^2 = 0.8992$$

Averaged DT of pixels with top 5% DT in the manual binary ground truth (x-axis)

Figure 3. **Cytoskeletal identification by ILEE. (a)** Visual demonstration and summarized mathematical process of ILEE. On the left, the visualized intermediate images of ILEE process are presented; on the right, an abbreviated mathematical process of ILEE is shown (gray, image pre-processing; blue, ILEE in a narrow sense; yellow, post-processing; also see Materials and methods for the detailed computational algorithm). **(b)** The value of implicit Laplacian smoothing coefficient $K$ influences ILEE performance. When $K$ is small (e.g., 2.5), the rendering of faint and thin filaments is accurate, but the edge of thick filaments tends to be omitted. Conversely, when $K$ is large (e.g., 200), rendering of thick filaments is accurate but thin and faint filaments are omitted. A solution is applied by creating a full outer join threshold image by a fixed $K_1$ = 2.5 and an estimated universal $K_2$ for an entire biological batch of samples. **(c)** $K_2$ estimation model. A regression model to compute any universal $K_2$ for a given sample batch (see Fig. S10 for detail). To augment the training samples, seven images with manually portrayed binary ground truth are interpolated by a new resolution of different folds to the original, into 42 samples (shown as single dots) that cover the general range of actin thickness. A non-linear regression estimation model is trained using the highest 5% DT (distance transform) values in the ground truth (representing filament thickness) and an anticipated $K_2$ with a specific satisfactory capability to detect thick filament (see Fig. S10 for detail). In the standard ILEE pipeline, the mean of top 5% DT of all input samples will be calculated by Niblack thresholding to obtain the universal $K_2$ for ILEE. Scale bars are marked as red/yellow solid line.

which serves to enhance the continuity of edges and contributes to the accuracy of edge detection (Fig. 1 b and Fig. 3 a). For this reason, we name our algorithm "Implicit Laplacian of Enhanced Edge."

For computation, the ILEE algorithm builds a linear system based on Laplacian operators to achieve local adaptive thresholding for edges of cytoskeletal components (see Materials and methods for detail). In brief, we first used the information from the coarse background to estimate a global threshold of the gradient magnitude ($g_{thres}$) and classified pixels/voxels above $g_{thres}$ as boundary elements ($I_{edge}$; Fig. 3 a, 2.1). Following boundary identification, we constructed an $n \times n$ ($n$ is the total number of pixel/voxel in the image) selection matrix $S$ and a sparse diagonal matrix whose diagonal corresponds to the flattened pixel/voxel in order; the diagonal element was 1 if the pixel/voxel possessed a norm of the gradient above $g_{thres}$. As the

pixels/voxels belonging to the boundary of the cytoskeleton filament were marked, $I_{edge}$ was mathematically defined as shown in Fig. 3 a, equations 2.2.1 and 2.2.2, where $L$ is the Laplacian matrix and $K$, or the implicit Laplacian smoothing coefficient, is a weight that adjusts the influence of the Laplacian operator. Therefore, the local threshold image can be rendered by solving the linear equation shown in Fig. 3 a, 2.2.3.

For a given image input, the performance of ILEE depends on two parameters: $g_{thres}$, which defines the edge, and $K$, which determines the weight of detail (i.e., high-frequency components) to be filtered. To calculate $g_{thres}$ for an input image, we used brightness values of the area identified as the coarse background by NNES. Since the ground noise is approximately subject to a normal distribution, we hypothesized a deducible statistical relationship between the image gradient, defined by the Scharr operator (Scharr, 2000), and the native brightness of

pixels/voxels within the coarse background. Using a normal random array that simulates the noise with a 2D or 3D data structure, we demonstrate that the distribution of the background gradient magnitude is also normal-like, and both mean ($\mu_G$) and standard deviation ($\sigma_G$) of the gradients are directly proportional to the standard deviation of their native pixel values ($\sigma_{cbg}$). Hence, we calculated the proportionality coefficient (see Fig. S6). For the 3D mode, since the voxel size on the *x*- and *y*-axis is different from that of the *z*-axis, the proportionality coefficient of $\mu_G$ and $\sigma_G$ over $\sigma_{cbg}$ will vary for different ratios of the *x–y* unit:*z* unit (see Fig. S7). To solve this problem, we simulated artificial 3D noise images and trained a multiinterval regression model that accurately ($R^2 > 0.999$) calculates the proportionality coefficient of $\mu_G$ and $\sigma_G$ over $\sigma_{cbg}$ for different *x–y*:*z* ratios of the voxel. Finally, using this approach and randomly selected actin image samples, we trained a model, $g_{thres} = \mu_G + k(\sigma_{cbg}) * \sigma_G$, to determine the $g_{thres}$ as an ILEE input for this study (Fig. S8). For 3D mode, we additionally developed enhanced $g_{thres}$ estimation models to further extend the applicability and reliability of ILEE_CSK library beyond the current study (Fig. S9).

To determine the appropriate setting of *K*, we first tested how different *K* values influenced the result of the local threshold image ($I_{thres}$ of Fig. 3 a). As shown in Fig. S10 a, at the optimal $g_{thres}$, a low value of *K* generated an $I_{thres}$ that was highly consistent with the selected edge; when *K* increases, the total threshold image shifted toward the average value of the selected edges, with an increasing loss of detail. As for the resultant binary image, a lower *K* enables the accurate recognition of thin and faint actin filament components, and yet cannot cover the entire width of thick filaments. Conversely, a high *K* value detects thick actin filaments with improved accuracy, resulting in a binary image that is less noisy; however, thin and/or faint filaments tend to be omitted as false-negative pixels (Fig. 3 b and Fig. S10 a). To overcome this dilemma, we applied a strategy using a lower $K_1$ and a higher $K_2$ to compute two different binary images that focus on thin/faint components and thick components, respectively. Then, we generated a full outer-joint image that contains all cytoskeleton components in these two binary images. This approach led to improved recognition of the cytoskeleton with varying morphologies (see Fig. 3 b).

As described above, $K_1$ controls the performance of thin and faint filaments. Since the theoretical minimum thickness of a distinguishable cytoskeletal component for most scenarios is approximately equal to one pixel/voxel unit, $K_1$ can be fixed as a constant to recognize the finest cytoskeletal components from a complex and heterogeneous set of input samples. Using this approach, we identified an empirical optimal $K_1$ of 2.5. However, users may adjust $K_1$ if they are processing exceptional images with an ultrahigh resolution, where the cytoskeleton monomer is larger than two pixels/voxels for extreme conditions. On the other hand, since different image samples have different distributions of cytoskeleton thickness, $K_2$, which controls the performance over-thick filaments, must be guided according to the maximum thickness among all samples in an experiment batch. To ensure that the algorithm is fully unguided, our strategy is to estimate an appropriate $K_2$ from an estimated

maximum thickness using all samples from a single batch of experiments, including multiple biological replicates (if applicable). To do this, we used Niblack thresholding (Niblack, 1985; API Ref. [1]) first to generate a coarse binary image (which is sufficiently accurate for the thickest portion of the filament); from this, we calculated the mean of the top 5% of the Euclidian distance transformation (DT) values of all positive pixels (see Materials and methods for additional information). Next, the top 5% means of all individual images were averaged to estimate $K_2$ via a trained model using a sample set with manual portraited binary ground truth (Fig. 3 c and Fig. S10, b–d). Therefore, individual images from all groups in the same batch of an experiment can be processed by $K_2$ as estimated using the model above. In this, the bias of human input was avoided. When processing 3D images, we additionally provided a parameter that allows using a single *K*, rather than $K_1$ and $K_2$, which balances the accuracy over thin/faint and thick/bright filaments while saving computation time.

## Computational analysis of cytoskeleton indices

Through the image processing pipeline, cytoskeletal indices were automatically calculated from the binary image generated by ILEE. As a substantial expansion from three previously defined cytoskeletal indices (e.g., occupancy, skewness, and CV; Higaki et al., 2010; Higaki et al., 2020), we introduced 12 indices in total (Fig. 1 a). Particularly, we focused on 9 of the 12 indices in this study as they are normalized and ready for biological interpretation (see Fig. S11 for visualized demonstration of each index). The other three indices (total connected element, total branch, and total node) serve as critical intermediate data that potentially contribute to the development of novel custom methods for advanced users and are callable through our library. All indices fall within five classes—density, bundling, connectivity, branching, and directionality, which describe the quantifiable features and functions of cytoskeletal morphology and dynamics. It is noteworthy that we interpolated the *z*-axis for 3D images to make a 1:1:1 length unit ratio for the three axes (i.e., cubic voxel) because the cuboid voxel generated by a confocal system is not compatible with most of the indices. Additionally, the indices require a certain level of image postprocessing (e.g., oversampling) to further enhance their accuracy. The detailed mathematical definition and processing of each index are described in Materials and methods.

Here, we define a novel set of cytoskeletal indices to enable the measurement of certain cytoskeletal features. For the class "density," we developed linear density, a feature that measures filament length per unit of 2D/3D space. For the class "bundling," we developed two new highly robust indices, the diameter by total DT (diameter_TDT) and the diameter by skeleton DT (diameter_SDT), both of which compute the thickness from the filament DT data but with different sampling strategies (see Materials and methods). Both quantify the physical thickness of filament bundles, while another two indirect indices, skewness and CV, estimate the relative bundling level based on the statistical distribution of fluorescence intensity. For the class "connectivity," we defined segment density, which estimates the sum of severing and nucleation activity within each unit of

length of the cytoskeleton. A severing event separates one filament into two halves, while a nucleation event generates a new segment of the filament. As both generate new segments, it is impossible to distinguish them through static images. However, if a researcher can assume one of these activities does not change significantly among samples at specific experimental conditions, the change in segment density can be attributed to the other activities. This is an important consideration in terms of the biological behavior of the cytoskeleton as it enables the decoupling of actin depolymerization and severing, key activities facilitated by the actin depolymerizing factor (ADF) and cofilin family of proteins (Tanaka et al., 2018). For the class "branching," our algorithm is based on Skan, a recently developed Python library for the graph-theoretical analysis of cytoskeletons (Nunez-Iglesias et al., 2018). To further explore the relationship between filament morphology and the biological activity of branching, we specifically designed a novel index, static branching activity, which we define as the total number of branches emerging from any non-end-point node per unit length of the filament. In total, this index measures the abundance/frequency of cytoskeletal branching. Finally, our library supports estimating the level of directional cytoskeletal growth by the local anisotropy index, which measures how local filaments tend to be directional or chaotic. This approach is reconstructed from an ImageJ plug-in FibrilTool (Boudaoud et al., 2014), but we extended this algorithm to 3D (Fig. 1 c).

It is noteworthy that segment density and static branching activity estimate the intensity of dynamic processes through static images. By the most rigorous mathematical criteria, a dynamic status cannot be directly defined without introducing time. However, the introduction of the time dimension (using a video sample) generally sacrifices spatial resolution due to the technical limit of microscopy devices. While total internal reflection fluorescence microscopy (TIRF) or variable-angle epifluorescence microscopy (VAEM) are used to collect 2D videos (Henty-Ridilla et al., 2014), such methods may be limited by the insufficient quantity of filaments in the field of vision and manual counting of the dynamic events, both of which may introduce bias compared with a larger 3D data structure. Alternatively, we introduce segment density and static branching activity as indirect indices measuring these activities. Here, we assume that the occurrence of severing/nucleation/branching events and the net regeneration speed of the cytoskeleton has established (or is very near to) a dynamic equilibrium; at this point, the number of pieces or branches of the filament is positively correlated with the speed of severing/nucleation or branching. In practice, most experimental schemes set an interval between the treatment (if any) and imaging, which is long enough to reach the dynamic equilibrium. Therefore, we expect these two novel indices to provide interpretable estimates of severing/nucleation and branching, especially for confocal images.

## ILEE displays high accuracy and stability over actin image samples

To evaluate the performance of ILEE in terms of its accuracy and compatibility over diverse samples, we selected a dataset of actin images from *Arabidopsis* leaves with a diverse morphology and compared ILEE with numerous traditional global and local thresholding algorithms, including MGT. First, to evaluate the accuracy of each algorithm in terms of filament segregation, we manually prepared the ground truth binary image from each of the samples using a digital drawing monitor (Fig. 4 a, ground truth). Next, we used each ground truth binary image as a reference and compared the filament architecture obtained by ILEE, MGT, and six additional adaptive thresholding algorithms. These additional thresholding algorithms include Otsu (Otsu, 1979), Triangle (Zack et al., 1977), Li (Li and Tam, 1998), Yan (Yen et al., 1995), Niblack (Niblack, 1985), and Sauvola (Sauvola and Pietikäinen, 2000; Fig. 4). Out of additional concern for rigor, we anticipated that false-positive pixels might be obtained due to operator's bias during the generation of each of the ground truth images (even when the operator is experienced in the actin imaging field). Therefore, we further analyzed each segment of false-positive pixels by its shape and connectivity to the matched filaments and identified the actin-like false-positive pixels as possible real actin components (see Materials and methods).

As shown in Fig. 4 a (visualized demonstration), Fig. 4 b (quantitative analysis), and Fig. 4 c (bias analysis), ILEE offers superior performance with improved accuracy, reduced false-positive/negative occurrences, and the lowest bias over local filament thickness compared with current approaches. It is noteworthy, however, that the adaptive global thresholding approaches (from Otsu to Yan) were generally able to detect some thick and bright bundles of the cytoskeleton. However, these approaches are unable to capture faint filaments, and as a result, generate a high false-negative rate. Conversely, both adaptive local thresholding approaches, Niblack and Sauvola, generate numerous block-shaped false-positive elements and fail to capture the near-edge region of thick filaments. For MGT and Li methods, although they showed satisfactory match rates and lower averaged false-positive/negative rates, their performance is far less stable than ILEE (Fig. 4 b).

Next, we evaluated the accuracy and stability of cytoskeletal indices using ILEE versus other commonly used image analysis algorithms. To do this, we first computed the ground truth indices from the manually generated binary images; then, quantitative measurements were collected from all methods and normalized by the relative fold to the results generated from the corresponding ground truth image. As shown in Fig. 4 d, ILEE showed the best overall accuracy and cross-sample stability compared with all other quantitative approaches, including the highest accuracy for occupancy, skewness, CV, and diameter_TDT. However, we also observed that in terms of the morphology-sensitive indices (i.e., linear density, segment density, and branching activity), ILEE did not fully conform to the data collected from ground truth binary images. Upon further inspection, we determined that this is because the manually portrayed ground truth images and ILEE results showed different tendencies in judging the pixels in the narrow areas between two bright filaments (see Discussion). While other approaches displayed obvious, somewhat predictable, inaccuracies, the MGT and Li methods still generated satisfactory

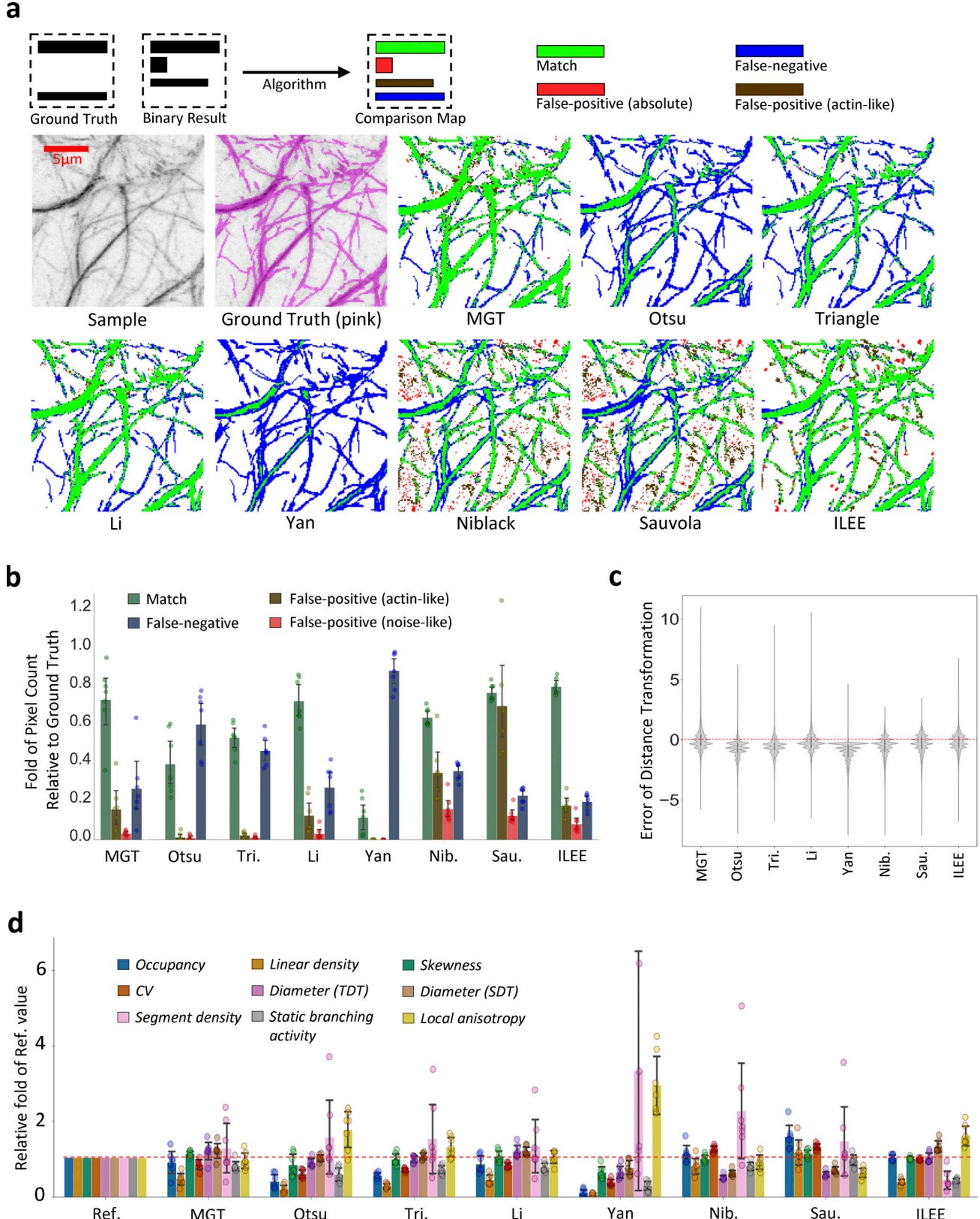

Figure 4. **ILEE shows superior accuracy and stability over classic thresholding approaches.** The manually portrayed binary ground truth of a diverse image set (*n* = 7, 400*400) was compared with corresponding binary images rendered by ILEE, MGT, four global thresholding algorithms (Otsu, Triangle, Li, Yan), and two local thresholding algorithms (Niblack and Sauvola). **(a)** Visualized demonstration of ILEE versus other approaches. Pixels with different colors are defined as green: match of rendered binary image and ground truth; blue: false-negative, the pixels omitted by the algorithm; red: noise-like false-positive, the pixels that are rendered by the algorithm but not in the ground truth without a shape of filament; brown: actin-like false-positive, the false-positive pixels

within a filament-like component, which cannot be judged as false result at high confidence. Scale bars are marked as red solid line. **(b)** Quantitative comparison of pixel rendering accuracy. The seven images annotated with single-pixel ground truth are tested. ILEE has a superior accuracy and with the highest match rate and a stably low error rate; MGT and Li also have acceptable performance. Error bar = standard deviation. **(c)** Comparison of the distribution of distance transformation error. Single pixel errors of all the images were merged ($n$ = ~2.5e5) and summarized as a violin plot. Red dashed line indicates zero error, or results identical to the ground truth. ILEE has a symmetric and centralized distribution, indicating an accurate and unbiased filament segmentation. **(d)** Comparison of computational accuracy of cytoskeletal indices. Nine biologically interpretable cytoskeleton indices computed using the binary images rendered by different algorithms were compared with the ground truth. The index values were normalized to the fold of ground truth. Red dashed line indicates 1-fold, or identical to the ground truth. Error bar = standard deviation. Routine significance tests do not apply to b, c, and d because both mean and standard deviation serve as independent quantitative metrics of algorithm performance rather than indicators of significant differences. Related results are shown in Figs. S12, S13, S14, and S15.

results, which echoes their performance in actin segmentation. However, the performance of these two algorithms among diverse and complex biological samples seemed to be less stable than ILEE as visually suggested by their standard deviation.

To accurately evaluate the stability and robustness of ILEE performance, we continued to analyze the variance coefficient of all groups (Fig. S12), uncovering that ILEE is the only approach that simultaneously maintained high accuracy and stability. Next, we tested the robustness of ILEE and other approaches against noise signal disturbance by adding different levels of Gaussian noise to the image dataset (Figs. S13, S14, and S15). Using this approach, we observed that ILEE is still the best-performing algorithm, maintaining stable and accurate image binarization and cytoskeleton indices against increasing noise. Taken together, these results demonstrate that ILEE has superior accuracy, stability, and robustness over MGT and other classic automated image thresholding approaches in terms of both cytoskeleton segmentation and index computation.

### 3D ILEE preserves more information and gains higher fidelity over 2D ILEE

While the 2D ILEE generally outcompetes MGT and other commonly used automated algorithms (Fig. 4), the comparison of cytoskeleton segmentation approaches using the manually painted ground truth is currently limited to 2D images because it is extremely difficult and error-prone to select the "3D ground truth" voxel by voxel. However, as the 3D mode theoretically preserves information from the z-axis and renders higher accuracy, it is necessary to circumvent these challenges and verify its merits. Therefore, we turned to a different strategy using synthetic artificial 3D actin images with known ground truth to investigate the performance of the 3D mode over 2D (Fig. 5 a and Fig. S16).

First, we constructed an actin image simulation model, which mimics three different fractions of image brightness (real actin, ground noise, and diffraction noise; see Fig. 2 a) using three independent statistical solutions (Fig. S16 a; also see Materials and methods). In brief, we utilized a training dataset of 31 diverse 3D samples to generalize the principles that describe the position-based brightness distributions of voxels for the three fractions independently. Next, we adopted only the skeleton image generated by 3D ILEE from the training samples as the topological frame and refilled the whole image with new brightness values via the actin image simulation model. The advantage of this, rather than generating it de novo, was to ensure that the topological structure of the artificial filaments

mimics the shape of the genuine actin to the maximum. Accordingly, we generated 31 3D artificial images and processed them using the ILEE 2D and 3D modes.

Our primary question was how 3D ILEE structurally improves the information loss by the 2D pipeline and whether it yields an improved accuracy of the cytoskeletal indices. As shown in Fig. 5 b, we observed that the 2D pipeline resulted in considerable information loss with reduced accuracy. Indeed, while the 3D segmentation almost accurately covers the filament voxels, the 2D pipeline only captured 8% of the data points as pixels and 42% of the total length of filaments, suggesting a potentially biased sampling. Consequently, 3D ILEE generally produces index values closer to the ground truth than 2D ILEE, which indicates the 3D mode indeed possesses a better absolute accuracy/fidelity.

In some scenarios, studies are interested in the relative difference between experimental groups for biological interpretations rather than the absolute accuracy of quantifiable features. Therefore, we also measured the "comparative sensitivity" of 3D and 2D ILEE, which reflects their capability to determine relatively high and low index values. To learn this, the index values of the ground truth images, as well as those computed through 3D and 2D modes, were first normalized to the fold of the minimum value of the corresponding group, followed by linear correlation analysis between the ground truth and ILEE outputs of each sample. As shown in Fig. 5 c, the 3D mode has a higher correlation for seven of the nine indices, among which diameter (TDT), segmentation activity, and anisotropy were considerably improved. Interestingly, we observed that the 2D mode performed slightly better for indices of the density class (e.g., occupancy and linear density). We posit that this is because z-axis projection may increase the contrast between samples with low and high filament density. In conclusion, our data suggest that the 3D mode of ILEE indeed eliminates the issues related to image projection and provides reliable results with higher accuracy and comparative sensitivity.

### ILEE leads to the discovery of new features of actin dynamics in response to bacteria

The primary impetus for creating the ILEE algorithm was to develop a method to define cytoskeleton organization from complex samples, including those during key transitions in cellular status. For example, previous research has demonstrated that the activation of immune signaling is associated with specific changes in the cytoskeletal organization (Henty-Ridilla et al., 2014; Henty-Ridilla et al., 2013; Li et al., 2017; Lu

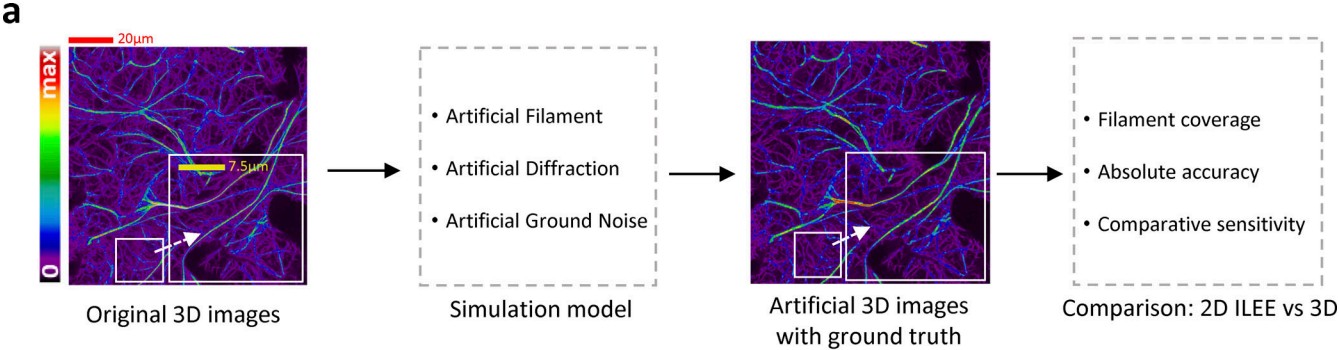

**a**

Original 3D images → Simulation model → Artificial 3D images with ground truth → Comparison: 2D ILEE vs 3D

Simulation model:
- Artificial Filament
- Artificial Diffraction
- Artificial Ground Noise

Comparison: 2D ILEE vs 3D:
- Filament coverage
- Absolute accuracy
- Comparative sensitivity

**b**

| Result (Relative to ground truth) | Ground truth | 3D ILEE | Projected ground truth | 2D ILEE |
|---|---|---|---|---|
| Filament coverage (by voxel) | 100% | 108.0±14.4% | 10.7±0.7% | 8.0±0.7% |
| Filament coverage (by length) | 100% | 105.1±11.6% | 59.4±4.5% | 42.0±4.7% |
| Match rate | 100% | 81.5±13.0% | NC | NC |
| False-positive rate | 0 | 26.5±7.1% | NC | NC |
| False-negative rate | 0 | 2.9±1.4% | NC | NC |
| *Occupancy* | 100% | 127.7±11.6% | 1017.0±67.6% (!) | 766.8±71.5% (!) |
| *Linear density* | 100% | 106.6±11.6% | 1792.9±121.5% (!) | 1218.2±135.6% (!) |
| *Brightness Skewness* | 100% | 103.6±3.2% | 100.4±4.3% | 96.2±5.1% |
| *Brightness CV* | 100% | 109.3±3.3% | 99.8±2.9% | 91.3±3.6% |
| *Diameter (TDT)* | 100% | 109.0±1.4% | 144.7±6.9% | 115.6±3.8% |
| *Diameter (SDT)* | 100% | 116.3±2.6% | 128±4.0% | 117.5±4.3% |
| *Segment density* | 100% | 67.7±8.0% | 43.3±8.0% | 25.6±3.8% |
| *Static branching activity* | 100% | 97.1±7.0% | 137.1±4.1% | 91.3±0.7% |
| *Local anisotropy* | 100% | 97.4±9.4% | 34.1±8.0% (!) | 58.5±10.2% (!) |

**c**

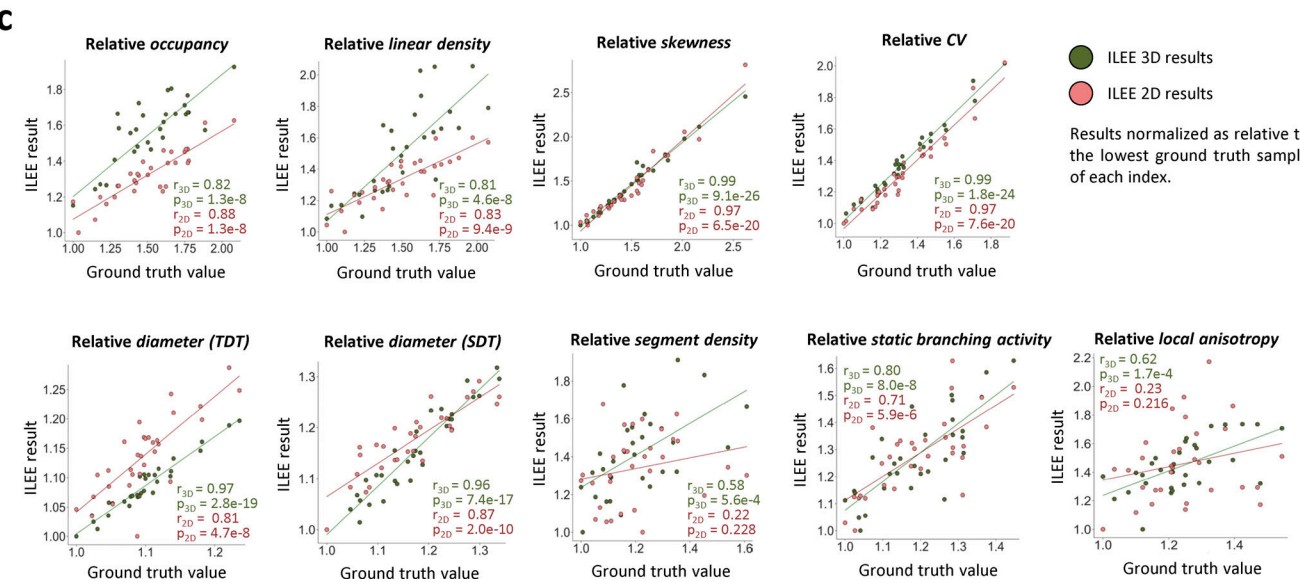

ILEE 3D results
ILEE 2D results

Results normalized as relative to the lowest ground truth sample of each index.

Relative *occupancy*: $r_{3D}$ = 0.82, $p_{3D}$ = 1.3e-8, $r_{2D}$ = 0.88, $p_{2D}$ = 1.3e-8

Relative *linear density*: $r_{3D}$ = 0.81, $p_{3D}$ = 4.6e-8, $r_{2D}$ = 0.83, $p_{2D}$ = 9.4e-9

Relative *skewness*: $r_{3D}$ = 0.99, $p_{3D}$ = 9.1e-26, $r_{2D}$ = 0.97, $p_{2D}$ = 6.5e-20

Relative *CV*: $r_{3D}$ = 0.99, $p_{3D}$ = 1.8e-24, $r_{2D}$ = 0.97, $p_{2D}$ = 7.6e-20

Relative *diameter (TDT)*: $r_{3D}$ = 0.97, $p_{3D}$ = 2.8e-19, $r_{2D}$ = 0.81, $p_{2D}$ = 4.7e-8

Relative *diameter (SDT)*: $r_{3D}$ = 0.96, $p_{3D}$ = 7.4e-17, $r_{2D}$ = 0.87, $p_{2D}$ = 2.0e-10

Relative *segment density*: $r_{3D}$ = 0.58, $p_{3D}$ = 5.6e-4, $r_{2D}$ = 0.22, $p_{2D}$ = 0.228

Relative *static branching activity*: $r_{3D}$ = 0.80, $p_{3D}$ = 8.0e-8, $r_{2D}$ = 0.71, $p_{2D}$ = 5.9e-6

Relative *local anisotropy*: $r_{3D}$ = 0.62, $p_{3D}$ = 1.7e-4, $r_{2D}$ = 0.23, $p_{2D}$ = 0.216

Figure 5. **The 3D mode of ILEE renders cytoskeleton indices at much higher accuracy than the 2D mode, due to advanced data structure.** We utilized image samples (*n* = 31, 25*800*800, Fig. 2) to train an actin image simulation model that generates 31 artificial 3D actin images with ground truth. By comparing the performance of 3D and 2D ILEE over the artificial images, it was demonstrated that the 3D mode displays high fidelity to the ground truth while the 2D mode suffered from loss of information and systemic bias. **(a)** The general experimental scheme. The raw 3D images are analyzed to train an actin image simulation model which uses statistical approaches to mimic the genuine actin filament, diffraction noise, and ground noise. Then 31 artificial actin images are regenerated with specific ground truth in terms of both actin segmentation and cytoskeleton indices. The 3D and 2D ILEE were applied to the artificial images, whose results were then compared to their corresponding ground truth. Scale bars are marked as red/yellow solid line. See Fig. S16 and Materials and methods for a detailed description of the algorithm. **(b)** A table comparing the absolute accuracy of ILEE by the 3D and 2D mode. It was demonstrated that 2D mode losses ~90% of pixels (voxels) and ~50% of the total length of actin samples and is less accurate for most of the indices. Results are presented as mean ± standard deviation. Exclamation mark (!): contrast difference due to different definitions of dimensional space in their units; not rigorously comparable. **(c)** The

linear correlation levels between ILEE results and the ground truth for all indices, by 2D and 3D mode. The absolute values of each index of the ground truth, 3D ILEE, and 2D ILEE are normalized by making their relative folds to the minimum data point of each group. A higher correlation coefficient r indicates a better capability to differentiate indices at different levels relatively, regardless of their absolute fidelity. The 3D mode has a generally better capability of differentiating high and low values. "r" represents Pearson correlation coefficient; "p" represents possibility to reject the null-hypothesis of no existent correlation by two-sided $t$ test. Data distribution was assumed to be normal but not formally tested.

---

et al., 2020). Complementary to these studies, other research identified the temporal and spatial induction of changes in the cytoskeletal organization as a function of pathogen (e.g., *Pseudomonas syringae*) infection and disease development (Guo et al., 2016; Kang et al., 2014; Shimono et al., 2016). The sum of these studies, which broadly applied MGT-based quantitative analysis of cytoskeleton architecture, concluded that the virulent bacterial pathogen triggers elevated density (by occupancy) but did not induce changes in filament bundling (by skewness) in the early stages of infection. Since one of our major motivations herein was to develop an ILEE-based toolkit—supported by novel cytoskeletal indices—to investigate the process of pathogen infection and immune signaling activation, we collected raw data from a previous study (Lu et al., 2020) describing a bacterial infection experiment using *Arabidopsis* expressing an actin fluorescence marker (i.e., GFP-fABD2; Fig. 6 a), followed by confocal imaging analysis by ILEE as well as MGT conducted by three independent operators with rich experience in actin quantificational analysis (Fig. 6, c–n). Additionally, because researchers sometimes apply a universal global threshold to all images from a batch of biological experiments to avoid tremendous labor consumption, we included this approach and aimed to observe its performance as well. In this experiment, the only categorical variant is whether sampled plants are treated with bacteria (EV) or not (mock). In total, nine indices that cover features of density, bundling, severing, branching, and directional order are measured and compared.

Our first question is whether the operator's bias generated by MGT will influence the results and conclusions of the experiment. We thereby analyzed the correlation of the MGT results by the three individual operators and found only a weak correlation between different operators (Fig. 6 b), which indicates MGT indeed introduces bias and potentially impacts quantitative results. Interestingly, while minor statistical discrepancies between MGTs by different operators are found in some indices (i.e., skewness and segment density), most of the MGT results (either adaptive or fixed) showed the same trend as 2D ILEE, but with far higher standard deviation or lower stability (Fig. S17 a) over a certain biological treatment. This indicates that the historical data based on MGT should be generally trustworthy despite the biased single data points, but an accurate conclusion must be based on a high sampling number that hedges the deviation of individual data points. Since ILEE has less systemic error over biological repeats, we also tested whether ILEE renders higher statistical power to identify potential significant differences. As suggested by Fig. S17 b, we found that ILEE has the lowest P values among $t$ tests conducted for indices with a trend of difference. We believe these advantages demonstrate ILEE as a better choice for actin quantification.

Next, we attempted to understand whether different indices of the same class, particularly density and bundling, can reflect the quantitative level of the class in accordance, or instead show inconsistency. For density, we correlated the occupancy and linear density values of all methods over actin images of both mock and EV groups and found that occupancy and linear density measurements are in high conformity, with a Pearson coefficient at 0.98 (Fig. S18). For bundling indices, we were interested in their level of conformity because direct indices (based on the binary shape) and indirect indices (based on the brightness frequency distribution) are entirely different strategies to measure bundling. Using the same approach of correlation analysis, we found that diameter_TDT and diameter_SDT display a strong positive correlation (Fig. 6 j), while skewness and CV have merely a medium-low correlation (Fig. 6 i), which echoes the previous report demonstrating that skewness and CV have different performance on the bundling evaluation (Higaki et al., 2020). Unexpectedly, we also found that CV (as a representative of indirect indices) and diameter_SDT (as a representative of direct indices) have a striking zero correlation (Fig. 6 k). This is perplexing as it raises the question of whether skewness or CV should be regarded as an accurate measurement of bundling (see Discussion). This discrepancy is also observed by 3D ILEE, whose CV and diameter_SDT over mock versus EV revealed the converse results with significant differences. In general, we believe that the biological conclusion that the *Pst* DC3000 treatment renders increased actin bundling should be further studied.

Lastly, we asked if additional features of plant actin cytoskeletal dynamics in response to virulent bacterial infection can be identified by the newly introduced indices and the enhanced performance of ILEE. As shown in Fig. 6, we observed a significantly regulated segment density, local anisotropy, and static branching activity triggered by the bacteria. At a minimum, these discoveries potentially lead to new biological interpretations and contribute to identifying other immune-regulated processes as a function of actin dynamics. However, while most 2D approaches were consistent and in agreement with the other indices, the segment density estimated by 3D ILEE indicates a significant but opposite conclusion. As suggested by the ground-truth-based comparison on accuracy between 3D and 2D ILEE (Fig. 5), we believed this discrepancy was due to the information loss and misinterpretation by the z-axis projection during the 2D pipeline. Hence, we have higher general confidence in the results and conclusions through the 3D mode.

**ILEE has broad compatibility with various sample types**
Cytoskeleton imaging from live plant samples is arguably one of the most challenging types of images to evaluate due to the

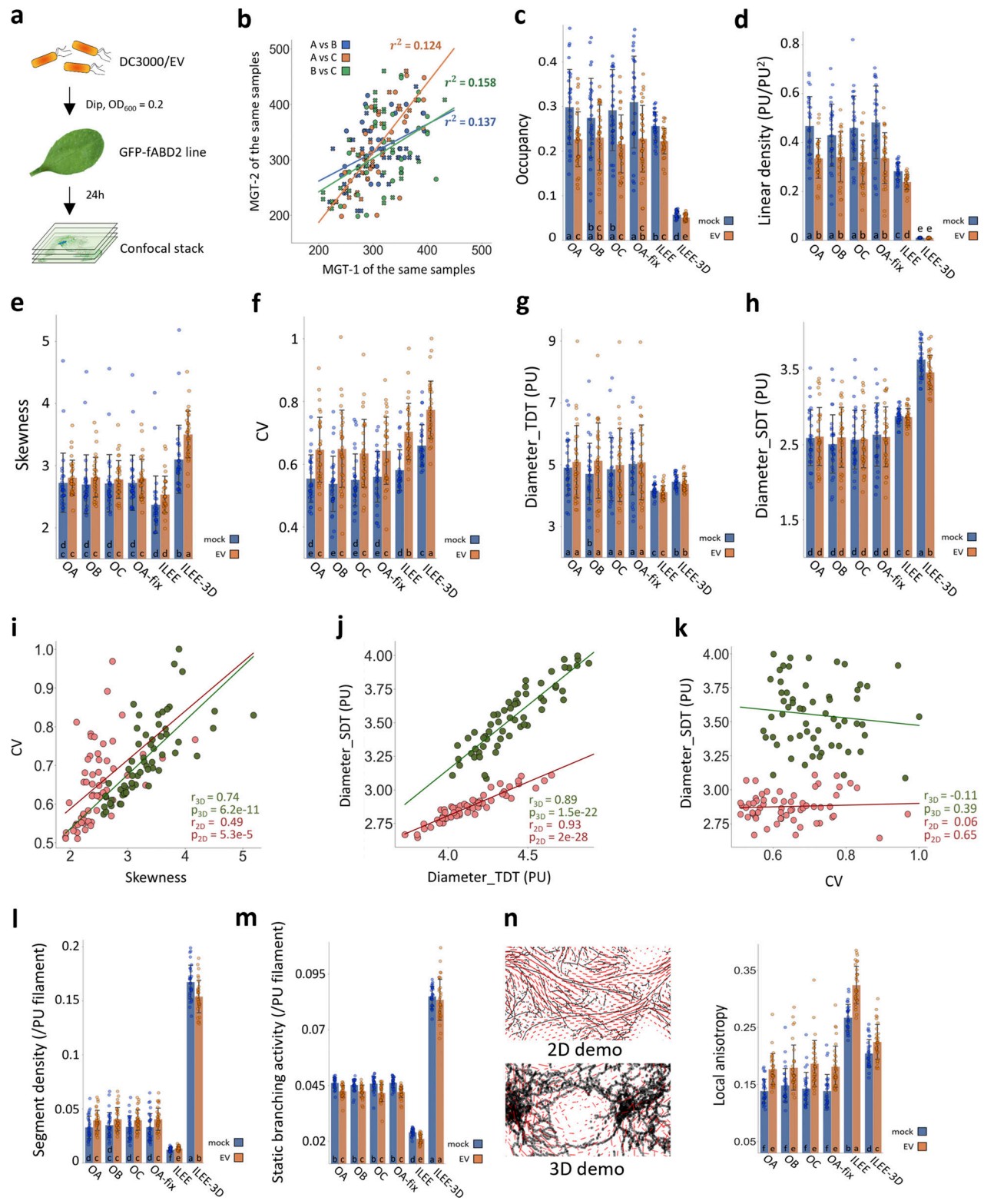

Figure 6. **ILEE library enables the discovery of actin dynamic features of bacteria infected leaf tissue.** Leaves of the Arabidopsis actin marker line Col-0/ GFP-fABD2 were inoculated with mock or a virulent bacterial pathogen, *Pseudomonas syringae* pv. tomato DC3000 with empty vector (EV, identical to wild type); images (*n* = 28 for mock, *n* = 31 for EV) of epidermal cells were captured by laser scanning confocal microscopy at 24 h post-inoculation (hpi). MGT and ILEE were applied to generate binary images and all indices were computed using ILEE_CSK. Double-blinded samples were provided to three operators (experienced researchers) OA, OB, and OC for comparison. OA additionally provides data using a universal threshold (OA-fix). **(a)** Experimental schematic diagram. **(b)** Corelative comparison of MGTs of individual samples determined by different operators. A very low correlation between each pair of operators indicates MGT has an increased risk of bias and is potentially inaccurate. **(c–h, l, m, and n)** Output indices occupancy, linear density, skewness, CV, diameter_TDT, diameter_SDT, segment density, static branching activity, and local anisotropy, respectively. Error bar = standard deviation. Multiple comparisons

are conducted using a two-sided *t* test without family-wise error correction (because image samples themselves are ground truths, and data are exact measurements of methods, rather than approximations). Groups without overlapping letters have a P-value lower than 0.05. Data distribution was assumed to be normal but not formally tested. For n, a visual illustration of the concept of local anisotropy is attached, where each red line segment points to the "averaged" direction of actin growth in the local area and the length shows the intensity of consistency of the direction. **(i, j, and k)** Comparisons of different indices in bundling class by individual image samples. The linear correlations between bundling indices are plotted and measured by Pearson coefficient. Skewness and CV have medium-weak correlation; diameter_TDT and diameter_SDT have strong correlation; diameter_SDT, as a representative of direct indicator, *and CV*, as a representative of indirect indicator, have literally no correlation. "r" represents Pearson correlation coefficient; "p" represents the possibility to reject the null-hypothesis of no existent correlation by two-sided *t* test. Data distribution was assumed to be normal but not formally tested.

---

dynamic topology and uneven brightness of actin filaments. While we demonstrated that ILEE shows superior performance over plant actin samples, ILEE and the ILEE_CSK library are generally designed for non-specific confocal images of the cytoskeleton, and therefore applicable to other types of samples. To investigate the compatibility of ILEE with other types of image samples, we tested ILEE on both plant microtubules (Faulkner et al., 2017) and animal cell actin images (Fig. S19). Importantly, we found that ILEE and the ILEE_CSK library can process both plant and animal images with a satisfying performance. This is encouraging as ILEE can substitute or improve the Hough transform, a straight-line detection algorithm commonly used for animal cytoskeleton (generally straight and thick), but with some limitations in neglecting and miscalculating curvy cytoskeleton fractions (Liu et al., 2020; Liu et al., 2018). With the advancement of ILEE, Hough transform-based analysis may not be essential, and the potential cytoskeleton indices that rigorously require the Hough transform can still utilize ILEE as a provider of binary image input for more accurate results.

## Discussion

Herein, we describe the creation of ILEE, an accurate and robust filament segregation algorithm for the unguided quantitative analysis of the organization of the eukaryotic cytoskeleton. As presented, this approach supports the in vivo analysis of both 2D and native 3D data structures, enabling an unbiased evaluation of cytoskeletal organization and dynamics. In addition to the development of key methods, we also generated a publicly available Python library that supports the automated computation of 12 filament indices of five classes of morphological features of the cytoskeleton. As described above, the data indicate that ILEE shows superior accuracy, robustness, and stability over existing cytoskeleton image analysis algorithms, including the widely adopted MGT approaches (Higaki et al., 2010; Lu and Day, 2017). As a result of this new approach, we have developed an open-access library to conduct ILEE-based cytoskeleton analysis, which eliminates limitations imposed by the traditional 2D MGT approaches, including the introduction of user bias and different types of information loss. Here, we would like to further explain and discuss several interesting discoveries and issues involved in this research.

### Robustness of the NNES-based prediction of global gradient thresholds

The gradient threshold ($g_{thres}$) defines the selected edge of actin filaments for implicit Laplacian transformation, the appropriateness of which greatly determines the performance of ILEE. To calculate $g_{thres}$ without imposing a user-biased input, our strategy utilized specific feature values collected from the NNES curve and human-guided MGT to train a prediction model for the rapid rendering of a coarse area of the image background. Through this approach, we were able to deduce the corresponding $g_{thres}$ by the mathematical relationship between the statistical distribution of the native pixel values and the Scharr gradient magnitude of the coarse background (Figs. S6, S7, S8, and S9, and Materials and methods). This step might first appear unnecessary since, alternatively, the most straightforward strategy was to directly train a prediction model using the image gradient histogram and human-determined $g_{thres}$. However, as the gradient operator (see Materials and methods) for any given object pixel is influenced by the values of the surrounding pixels, the calculated gradient on the edge of the background is highly impacted (overestimated) by the contrast of foreground (cytoskeleton) versus the background. In other words, the brightness frequency distribution of the background gradient will change at elevated cytoskeleton fluorescence levels, even though the background per se remains the same. The outcome of this is a significant decrease in the accuracy of gained $g_{thres}$. For this reason, we assert that $g_{thres}$ should be mathematically deduced from a predetermined background region rather than directly predicted via human-trained models or calculated from the frequency distribution of gradients even using the predetermined background.

### ILEE and visual inspection have different tendencies on the topology between two bright filaments

Our data demonstrate that ILEE generally shows dominant robustness and accuracy for most indices compared with manually portrayed ground truth binary images. However, there are three indices (linear density, segment density, and static branching activity) where ILEE renders stable yet dramatically lower output than those derived from ground truth images (Fig. 4 d). Since the results are stable, we anticipate there may be systemic inclinations in either the human-portrayed ground truth or the ILEE algorithm—or both. After inspecting the binary images generated by ILEE compared with the ground truth, we identified a potentially critical reason: ILEE is less likely to judge the faint, ambiguous thin "connections" inside the narrow space between two bright filaments as positive filaments. As demonstrated in Fig. S20, while the ground truth and ILEE binary image look very similar, their skeleton images—which represent their topological structure—show a discrepancy between two bundles of bright filaments. Considering their procedure of generation, we speculate this is because (1) ILEE fully outer-joins

the binary results by a lower $K_1$ and a higher $K_2$ (Fig. 3 c), among which $K_2$ sacrifices the sensitivity to local filaments at high signal regions to improve the sensitivity at low signal regions, including the edges of thick filaments and (2) human eyes may recognize "imaginary filaments" that may not exist in such complex areas of images. According to our knowledge, there is no overwhelming evidence suggesting either 2D ILEE or the human eye is more accurate, but 2D ILEE is indeed more stable and conservative. However, 3D ILEE may solve this paradox because many "adjacent" bright bundles are artifacts out of the z-axis projection, which is distant enough in 3D space to offer ILEE a satisfactory resolution to split the filaments.

### The 2D and 3D modes of ILEE may lead to different conclusions
For the analysis of leaf samples upon bacterial pathogen treatment (Fig. 6), 2D ILEE generally agreed with MGT while offering higher robustness and stability. Interestingly, this is not always the case for 3D ILEE since it led to different statistical conclusions from 2D approaches in linear density, skewness, diameter (SDT), segment density, and static branching activity. However, such differences are understandable because the 3D mode indeed computes indices with higher accuracy and overcomes many limitations of the 2D data structure. As demonstrated in Fig. 5, b and c, 3D ILEE provide more or less improved absolute fidelity and comparative sensitivity for skewness, diameter (SDT), segment density, and static branching activity; therefore, our current data support that 3D ILEE is more trustworthy for these indices. Similarly, the discrepancy of linear density over the 2D mode versus 3D in Fig. 6 d also agrees with our observation that 2D ILEE has higher comparative sensitivity for indices reflecting density. Hence, in terms of any potential discrepancy between the 2D and 3D modes, we offer this general recommendation: for occupancy only, 2D ILEE has better capability to distinguish biological differences (but not necessarily the absolute quantification), and for all other cytoskeleton features, 3D ILEE is more trustworthy and dependable. Although it is currently difficult to make a definite conclusion about whether the current 2D or 3D mode is more accurate across all scenarios, it is predictable that the 3D computation will gradually substitute for the classic 2D data structure, with mainstream personal computers gaining increased computational power in the future.

### Potential development and prospect
While ILEE has already remedied many disadvantages of traditional methods such as MGT, we are still working to further advance the ILEE approaches presented herein. Our goal is to ultimately establish algorithms and a toolbox that provide cytoskeletal indices at perfect accuracy and effectively answer the specific demands in this field of study. As such, we offer the following as a list of potential upgrades and applications to be integrated into the library:

Deep learning–based cytoskeleton segmentation algorithm with a "foreign object" removal function. As presented herein, ILEE enables the generation of trustworthy binary images on a large scale, which enables the construction of deep learning models to identify cytoskeleton components from confocal images with potentially better performance. Deep learning is also the key to solving the ultimate problem of all current cytoskeleton segmentation algorithms (including ILEE)—the inability to detect and erase non-cytoskeleton objects with high fluorescence, such as the nucleus and cytoplasm. As one approach to circumvent this limitation, free fluorescence proteins can be used as a cell-permeable false signal. This will enable training a model to recognize and exclude the non-cytoskeleton-like foreign objects, ideally rendering pure cytoskeletal datasets.

Vectorized intermediate image. After generating the different image (i.e., $I_{dif}$, Fig. 3 a) using ILEE, one computational limitation of our Python-based algorithm is the tradeoff between the demand for unlimited high-resolution imaging versus limited computational power. Accordingly, an ideal strategy is to transfer the pixel/voxel image to a 2D vector image or 3D polygon mesh for index calculation. This will further enhance the accuracy of ILEE given an acceptable requirement of computational power.

Regions of interest and organelle segmentation. There is currently a high demand in plant cell biology to quantify cytoskeletal parameters in stomatal guard cells, as well as additional cellular and subcellular architectures. In future releases of ILEE, we aim to develop approaches to enable automated recognition and selection of regions of interest, such as stomata, for various demands by the community.

Compatibility to x-y-t and x-y-z-t data, where t represents time. We are in the process of developing a 4D-compatible analysis of cytoskeletal dynamics that tracks filament organization over time. This approach will provide a temporal evaluation of supported indices with high accuracy and robustness.

## Materials and methods
### Plant genotypes and growth
*Arabidopsis thaliana* ecotype Col-0 (wild type) expressing the actin cytoskeleton fluorescence marker GFP-fABD2 (Lu et al., 2020) was used in this study. *Arabidopsis* seeds were stratified for 2 days in the dark at 4°C and then sown into the soil. All plants were grown in a BioChambers model FLX-37 walk-in growth chamber (BioChambers) at 20°C under mid-day conditions (12 h of light/12 h of dark) with 60% relative humidity and a light intensity of ~120 µmol photons m$^{-2}$s$^{-1}$.

### Bacteria growth and plant inoculation
Plant pathogenic bacteria *Pseudomonas syringae* pv. *Tomato* DC3000 carrying an empty broad host vector pVSP61 (*Pst* DC3000/EV; Loper and Lindow, 1994) was grown on NYGA plates (5 g/l peptone, 3 g/l yeast extract, 2% glycerol, and 15 g/l agar) plus 50 µg/ml rifampicin and 50 µg/ml kanamycin for cultivation and inoculation. Bacterial treatments for actin dynamics analysis were conducted following previously described methods (Lu et al., 2020). Briefly, 2-wk-old *Arabidopsis* Col-0/GFP-fABD2 was dipped for 30 s in Dip-inoculation solution (10 mM MgCl2 + 0.02% Silwet-77) with DC3000/EV at a concentration OD$_{600}$ = 0.2 (ca. 2 × 10$^7$ colony forming units/ml). Confocal images were collected 24 h after inoculation.

### Mouse cancer cells sample

Yale University Mouse Melanoma line YUMMER1.7D4 cells (#SCC243; EMD Millipore) were cultured in DMEM (#30-2006; ATCC) supplemented with 10% FBS (#10437-028; Gibco), 1% Pen-strep (#15140122; Thermo Fisher Scientific), and 1% NEAA (#11140035; Gibco). For staining actin stress fibers, ~10,000 cells were seeded onto a glass coverslip maintained in a six-well plate overnight. First, semiconfluent cells were fixed with 3.7% formaldehyde for 15 min at RT, washed three times with PBS, then blocked in PBS supplemented with 2% BSA and 0.1% Triton for 1 h at RT. Next, cells were incubated with 100 nM rhodamine–phalloidin (#PHDR1; Cytoskeleton, Inc) in a blocking buffer for 30 min at RT in the dark and then washed with PBS three times each for 5 min with gentle shaking at RT. Stained cells on the coverslip were mounted in ProLong Glass Antifade Mountant with DAPI (#P36982; Thermo Fisher Scientific). Slides were cured overnight at room temperature and then imaged.

### Confocal microscopy

For plant leaf actin imaging, 2-wk-old Col-0/GFP-fABD2 plants were used for data collection and analysis. Images of epidermal pavement cells and guard cells were collected using a laser scanning confocal microscope (Olympus FV1000D) at RT. Optical setting: 65×/1.42 PlanApo N objective with a 488 nm excitation laser and 510–575 nm emission filter (For GFP fluorescence); and the imaging medium is Olympus Low Autofluorescence Immersion Oil (Olympus # IMMOIL-F30CC). Z-stack Images were collected at a resolution of 800 × 800 × 25 (x–y–z) with a 12-bit dynamic range. Voxel size was 0.132 µm at the x- and y-axis and 0.5 µm at the z-axis. For animal cell actin images, YUMMER1.7D4 cell stained by rhodamine–phalloidin were sampled by the same confocal system. Optical setting: 100×/1.40 UPLSAPO with a 559 nm excitation laser and 570–670 nm emission filter (for rhodamine). The same immersion oil was used. Z-stack images were collected at a resolution of 800 × 800 × 10 (x–y–z) with a voxel size of 0.039 µm at the x- and y-axis and 0.16 µm at the z-axis. The microscope operation software is FV10-ASW, ver. 04.02.02.09 (Olympus Corporation).

### Manually portrayed ground truth binary image

Seven raw projected images of 400 × 400 pixel size with diverse visual appearance (e.g., actin density, shape, thickness, and fluorescence brightness) were selected from our actin image database. Using a pen-drawing display (HUION KAMVAS 22 Plus, GS2202) and GIMP software, we slightly enhanced the brightness of low-value pixels to clarify the actin structure and carefully portrayed the actin component of the selected image sample at a single-pixel level. The portrayed layer was extracted and transferred to the binary format for further evaluation.

### Double-blind MGT analysis

For the MGT process of the mock versus DC3000/EV-inoculated sample pool in Fig. 6, we erased the name, randomized the order of the image files, and distributed them to three independent cell biologists with rich experience in cytoskeleton analysis (referred to as OA, OB, and OC) to let them determine the global threshold value of each sample manually using the approach described previously (Lu et al., 2020). Briefly, ImageJ, or equivalent GUI(s), were used to real-timely mark the pixels over a manually tunable threshold, letting the operator visually determine a value segmenting the cytoskeleton at the best performance according to his/her subjective opinion. Once completed, we restored the grouping of the samples for batch analysis. We use Python to mimic the MGT pipeline that was generally conducted by ImageJ using the determined thresholding value. Operator OA also provided a universal threshold value (referred to as OA_fix, Fig. 6) that applies to all samples as a commonly used fast MGT approach.

### Statistical analysis and data visualization

All data analysis was conducted in "Spyder" IDE (integrated development environment) with Python 3.8/3.10 environment. Image and data visualization, including built-in calculation of mean and standard deviation, were conducted by "matplotlib" and "seaborn" libraries (API Ref. [10, 11]). $t$ tests, including $t$-test-based multiple comparisons, were conducted using "scikit-posthocs" library (API Ref. [12]). Linear correlation analysis was conducted by "SciPy" library (API Ref. [3]). Linear and non-linear modeling was conducted using "scikit-learn" library (API Ref. [13]). Fitting and generalization of specific statistical distribution were conducted by "SciPy" library (API Ref. [3]). The rationale of statistical analysis is described in the corresponding figure legends.

### Determination of $K_2$ for sample batches

For both 2D and 3D modes of ILEE, each 12-bit single-channel 3D image $I(x,y,z)$ in a batch of samples was transferred into a 2D image $I_{proj}(x,y)$ by z-axis maximum projection, where each pixel was $I_{proj}(x, y) = \max\{I(x, y, z) | z \in N^+, z < z_{max}\}$. $I_{proj}$ was processed by a Niblack thresholding (API Ref. [1]) to render $I_{nibthres}(x,y)$, with the parameter optimized to $k = -0.36$ and $window\_size = 2int(25l)+1$ for best performance, where $l$ is the mean of the x and y resolutions of $I_{proj}$. A binary image defined as $I_{binary}(x, y) = \{1, \ if \ I_{proj}(x, y) > I_{nibthres}(x, y); \ 0, \ else\}$ was generated. The binary image was processed through Euclidean distance transformation (API Ref. [2]), and the mean of the highest 5% values was used as the input of the $K_2$ estimation model (see Fig. 3 c) that outputs individual recommended $K_2$. Finally, the mean of all individual $K_2$ will be output as the recommended $K_2$ for the entire group.

### ILEE

An abbreviated workflow of ILEE is illustrated in Fig. 3 b. Here, we describe the overall process in further detail. For the 2D mode, the input image structure is $I(pix) = I_{proj}(x, y) = \max\{I(x, y, z) | z \in N^+, z < z_{max}\}$. For the 3D mode, $I(pix) = I(x,y,z)$. First, $I$ is treated by a significant difference filter (SDF; Fig. S5) to render $I_{SDF}$, where:

$$I_{surround} = \{I(x \pm 1, y(, z)), I(x, y \pm 1(, z)), (0.293I(x, y(, z)) + 0.707 \\ I(x \pm 1, y \pm 1(, z)))\},$$

$$I_{SDF}(x, y(, z)) = \\ \begin{cases} I(x, y(, z)), if \ |Mean(I_{surround}) - I(x, y(, z))| < 2STD(I_{surround}) \\ Mean(I_{surround}), else. \end{cases}$$

It is noteworthy that the coefficients 0.293 and 0.707 represent a linear interpolation to estimate the value on the diagonal that is 1 pixel from the center (Fig. S5). In total, the SDF substitutes a pixel by the mean of the eight adjacent pixels if the absolute difference between it and the mean is higher than two folds of the standard deviation of the surrounding pixels. Then, $I_{SDF}$ is input to a discrete Gaussian filter with a $3 \times 3 (\times 3)$ weighting kernel at $\sigma = 0.5$ to render $I_{pre} = Gauss(I_{SDF})$, which is the smoothed preprocessed image. Since confocal microscopy has different resolutions on the $x/y$ and $z$ axes (hereby named as $U_{xy}$ and $U_z$), we adjusted the weighting kernel from $O_{Gauss}$ to $O_{Gauss}'$ in the 3D mode particularly by a scaling operator, as shown below:

$$f = U_{xy}/U_z,$$

$$O_{scalar} = \frac{3}{1+2f} \begin{bmatrix} \begin{bmatrix} f & f & f \\ f & f & f \\ f & f & f \end{bmatrix} \\ \begin{bmatrix} 1 & 1 & 1 \\ 1 & 1 & 1 \\ 1 & 1 & 1 \end{bmatrix} \\ \begin{bmatrix} f & f & f \\ f & f & f \\ f & f & f \end{bmatrix} \end{bmatrix},$$

$$O'_{Gauss} = O_{Gauss} \circ O_{scalar}.$$

From $I_{pre}$, the gradient magnitude image $G$ is rendered through the Scharr operator (Scharr, 2000) as:

$$G_x = \begin{bmatrix} 3 & 0 & -3 \\ 10 & 0 & -10 \\ 3 & 0 & -3 \end{bmatrix} * I_{pre},$$

$$G_y = \begin{bmatrix} 3 & 10 & 3 \\ 0 & 0 & 0 \\ 3 & -10 & -3 \end{bmatrix} * I_{pre},$$

$$G = \frac{\sqrt{G_x^{\circ 2} + G_y^{\circ 2}}}{32\sqrt{2}}$$

for the 2D mode, or:

$$G_x = \begin{bmatrix} \begin{bmatrix} 9 & 0 & -9 \\ 30 & 0 & -30 \\ 9 & 0 & -9 \end{bmatrix} \\ \begin{bmatrix} 30 & 0 & -30 \\ 100 & 0 & -100 \\ 30 & 0 & -30 \end{bmatrix} \\ \begin{bmatrix} 9 & 0 & -9 \\ 30 & 0 & -30 \\ 9 & 0 & -9 \end{bmatrix} \end{bmatrix} * O_{scalar} * I_{pre},$$

$$G_y = \begin{bmatrix} \begin{bmatrix} 9 & 30 & 9 \\ 0 & 0 & 0 \\ 9 & -30 & -9 \end{bmatrix} \\ \begin{bmatrix} 30 & 100 & 30 \\ 0 & 0 & 0 \\ -30 & -100 & -30 \end{bmatrix} \\ \begin{bmatrix} 9 & 30 & 9 \\ 0 & 0 & 0 \\ -9 & -30 & -9 \end{bmatrix} \end{bmatrix} \circ O_{scalar} * I_{pre},$$

$$G_z = \begin{bmatrix} \begin{bmatrix} 9 & 30 & 9 \\ 30 & 100 & 30 \\ 9 & 30 & 9 \end{bmatrix} \\ \begin{bmatrix} 0 & 0 & 0 \\ 0 & 0 & 0 \\ 0 & 0 & 0 \end{bmatrix} \\ \begin{bmatrix} -9 & -30 & -9 \\ -30 & -100 & -30 \\ -9 & -30 & -9 \end{bmatrix} \end{bmatrix} \circ O_{scalar} * I_{pre},$$

$$G = \frac{\sqrt{G_x^{\circ 2} + G_y^{\circ 2} + G_z^{\circ 2}}}{256}$$

for the 3D mode. Next, to calculate the gradient threshold ($g_{thres}$) as the input for ILEE, a global threshold $t_{cbg}$ to determine the coarse background ($I_{cbg}$, as flattened image) was calculated by the non-connected negative element scanning (NNES) function that satisfies the following:

$$I_{binary.NNE}(x) = \{I(pix)|I(pix) > x\},$$
$$Count.of.NNE(I_{binary.NNE}(x_{peak})) = Count.of.NNE(I_{binary.NNE}(x))_{max},$$

$$t_{cbg} = \begin{cases} 1.163 \ x_{peak} + 101.68, & if \ I \ is \ 2D \ (see \ Fig. \ S1 \ c) \\ 7.334 \ x_{peak}, & if \ I \ is \ 3D \ (see \ Fig. \ S2 \ b) \end{cases},$$
$$I_{cbg} = \{I(pix)|I(pix) \le t_{cbg}\}.$$

If $I$ is processed using the 2D mode, as demonstrated previously (see Fig. S6), the statistical mean and STD of gradient magnitude of $I_{cbg}$ ($\mu_{G.cbg}$ and $\sigma_{G.cbg}$, respectively) are univariate proportional functions of the STD of $I_{cbg}$, $\sigma_{cbg}$, not influenced by the mean of $I_{cbg}$. Particularly, $\mu_G$ and $\sigma_G$, of the coarse background area of the gradient image is:

$$\mu_{G.cbg} = 0.8542 \ \sigma_{cbg},$$
$$\sigma_{G.cbg} = 0.4469 \ \sigma_{cbg}.$$

With these, we established a $g_{thres}$ estimation model, where $g_{thres} = \mu_{G.cbg} + k\left(\sigma_{cbg}\right)\sigma_G$. Using the optimized parameters described in Fig. S8, $g_{thres}$ is deduced as:

$$k(\sigma_{cbg}) = 0.040018 \ \sigma_{cbg},$$
$$g_{thres} = \mu_{G.cbg} + k(\sigma_{cbg})\sigma_{G.cbg},$$

If $I$ is processed by the 3D mode, the inconformity of $U_{xy}$ and $U_z$ will not only impact the weighting of Scharr operator but also influence the proportional coefficient of $\sigma_{cbg}$ to $\mu_G$ and $\sigma_G$. We simulated the accurate mathematical relationship between the proportional coefficient $k_s$ and $U_z/U_{xy}$ (see Fig. S7), so $g_{thres}$ for 3D can be calculated as:

$$\mu_{G.cbg} = k_{s.\mu} \ \sigma_{cbg},$$
$$\sigma_{G.cbg} = k_{s.\sigma} \ \sigma_{cbg},$$
$$k(\sigma_{cbg}) = 0.040519 \ \sigma_{cbg},$$
$$g_{thres} = \mu_{G.cbg} + k(\sigma_{cbg})\sigma_{G.cbg}.$$

The total process above to compute $g_{thres}$ is referred to as the function $g_{thres}.estimate(I_{pre}, I_{cbg})$ in Fig. 3 a (1.3).

The critical step to generate the threshold image of the input sample $I$ is implicit Laplacian smoothing. This algorithm builds a linear system using the Laplacian operator to achieve local adaptive thresholding based on the edges of cytoskeletal components. Leveraging the spectral characteristics of discrete

Laplacian operators, we could filter out high-frequency components (aka, high fluorescence fractions of the cytoskeleton) while preserving the low-frequency salient geometric features of background fluorescence. First, an edge image is defined as $I_{edge}(pix) = I_{pre}(pix)$, if $G(pix) > g_{thres}$; $0, else$, but we transform $I_{edge}$ into a flattened image vector $\overrightarrow{I_{edge}}$, which can be represented as:

$$\overrightarrow{I_{edge}} = S\ \overrightarrow{I_{pre}},$$

where $S$ is a (sparse) selection matrix, which is a diagonal matrix with $i$-th diagonal entry being 1 if the $i$-th pixel has a gradient above $g_{thres}$.

Next, according to the concept of the Laplacian operator, we constructed the (sparse) Laplacian matrix $L$ that satisfies:

$$L = \begin{cases} L_4\ \overrightarrow{I_{flt}} = \Delta_{p\_4} * I(x,y)\ (2D\,alternative) \\ L_8\ \overrightarrow{I_{flt}} = \Delta_{p\_8} * I(x,y)\ (2D\,default) \\ L_6\ \overrightarrow{I_{flt}} = \Delta_{p\_6} * I(x,y,z)\ (3D) \end{cases}$$

where the Laplacian operators in different modes are defined as:

$$\Delta_{p\_4} = \begin{bmatrix} 0 & -1 & 0 \\ -1 & 4 & -1 \\ 0 & -1 & 0 \end{bmatrix},$$

$$\Delta_{p\_8} = \begin{bmatrix} -\sqrt{2}/2 & -1 & -\sqrt{2}/2 \\ -1 & 4+2\sqrt{2} & -1 \\ -\sqrt{2}/2 & -1 & -\sqrt{2}/2 \end{bmatrix},$$

$$\Delta_{p\_6} = \begin{bmatrix} \begin{bmatrix} 0 & 0 & 0 \\ 0 & -1 & 0 \\ 0 & 0 & 0 \end{bmatrix} \\ \begin{bmatrix} 0 & -1 & 0 \\ -1 & 6 & -1 \\ 0 & -1 & 0 \end{bmatrix} \\ \begin{bmatrix} 0 & 0 & 0 \\ 0 & -1 & 0 \\ 0 & 0 & 0 \end{bmatrix} \end{bmatrix}.$$

Therefore, we can establish an implicit Laplacian linear system that targets the edge:

$$(L + KS)\ \overrightarrow{I_{thres}} = \overrightarrow{I_{edge}} = S\ \overrightarrow{I_{pre}},$$

where $\overrightarrow{I_{thres}}$ is the unknown to be solved by the conjugate gradient method and restored to the 2D/3D array $I_{thres}$, and $K$ is a weight that adjusts the influence the Laplacian operator has on the result (see Fig. 3 b).

Finally, a difference image $I_{dif}$ is calculated as $I_{dif} = I_{pre} - I_{thres}$. To further remove noise, a temporary binary image $I_{binary.temp}(pix) = \{1, if\ I_{dif}(pix) > 0;\ 0,\ else\}$ is generated, and the sizes of all positive connected components (elements) are counted. An adjusted difference image $I_{dif\_adj}$ is generated by lowering the values of potential noise into the mean of negative pixels:

$$I_{dif\_adj} = \begin{cases} I_{dif}(pix),\ if\ element\ size \geq 3 \\ Mean\ (I_{dif}(pix)|I_{dif}(pix) < 0),\ else \end{cases}.$$

Using $I_{dif\_adj}$, the various versions of the binary image for the computation of different indices are calculated as:

$$I_{binary}(pix) = \{1, if\,I_{dif\_adj}(pix) > 0;\ 0, else\},$$
$$I_{binary.ovsp}(pix) = \{1,\ if\ Interp_{3,3}(I_{dif\_adj}(pix)) > 0;\ 0,\ else\}$$

for the 2D mode, where $Interp_{3,3}(I_{dif})$ is an oversampled image by bicubic interpolation whose resolution on both $x$- and $y$-axis are upscaled by threefold, or

$$I_{binary.ori}(pix) = \{1,\ if\ I_{dif}(pix) > 0;\ 0,\ else\},$$
$$I_{binary}(pix) = \{1,\ if\ Interp_{1,1,1/f}(I_{dif}(pix)) > 0;\ 0,\ else\},$$
$$I_{binary.ovsp}(pix) = \{1,\ if\ Interp_{3,3,3/f}(I_{dif}(pix)) > 0;\ 0,\ else\}$$

for the 3D mode, where $Interp_{1,1,1/f}(I_{dif}(pix))$ aims to restore the voxel to cubic shape by only interpolating the $z$-axis to $1/f$ folds of its original resolution. $Interp_{3,3,3/f}(I_{dif})$ additionally enhances the resolution for all three axes threefold as an optional process.

**Computation of cytoskeletal indices**

**Occupancy:** the frequency of the positive pixels in the computed binary image, or $\sum I_{binary}(pix)/N$ for the 2D mode and $\sum I_{binary.ori}(pix)/N$ for the 3D mode, where $N$ is the number of total pixels.

**Linear density:** the length of the skeletonized filament per unit of 2D or 3D space. For 2D mode, $I_{binary.ovsp}$ is skeletonized using Lee's approach (Lee et al., 1994; API Ref. [9]) to render the skeletonized image $I_{sk}$. Then, linear density is calculated as:

$$linear\ density = \sum length(x,y)/N,$$
$$length(x,y) = \begin{bmatrix} \sqrt{2}/2 & 1/2 & \sqrt{2}/2 \\ 1/2 & 0 & 1/2 \\ \sqrt{2}/2 & 1/2 & \sqrt{2}/2 \end{bmatrix} * I_{sk}(x,y).$$

For the 3D mode, $I_{sk}$ is rendered by $I_{binary}$, and we use the sum of the Euclidean lengths of all (graph theory defined) branches obtained by the Skan library (Nunez-Iglesias et al., 2018) as the total length of the skeletonized filament and divide it by $N$. This is because sphere-cavities structures existent in the 3D skeletonized images are not applicable to the concept of length.

**Skewness:** the (statistical) skewness of the fluorescence value of positive pixels, or mathematically:

$$skewness = \frac{1}{N_{pos}} \sum_{I_{binary}(pix=1)} \left( \frac{1(pix) - \mu_{pos}}{\sigma_{pos}} \right)^3$$

where $N_{pos}$, $\mu_{pos}$, and $\sigma_{pos}$ represents the count, mean, and standard deviation of positive pixels in the raw image $I$.CV: the (statistical) coefficient of variance of the fluorescence value of positive pixels, or mathematically:

$$CV = \frac{\sigma_{pos}}{\mu_{pos}}.$$

**Diameter_TDT:** average filament diameter estimated by the Euclidian distance transformation of the total binary image. The Euclidian distance transformation map $I_{dis}$ is calculated as an image with the same shape as $I_{binary.ovsp}$, but the value of each pixel is:

$$I_{dis}(x,y(,z)) = \begin{cases} \sqrt{\sum_{i=x,y(,z)} (i - i_{nnp})^2}, & \text{if } I_{binary.ovsp}(x,y(,z)) = 1 \\ 0, & \text{if } I_{binary.ovsp}(x,y(,z)) = 0 \end{cases},$$

where $i_{nnp}$ is the coordinate of the nearest negative pixel to $(x,y,z)$. Therefore, the *Diameter_TDT* is mathematically defined as:

$$\text{Diameter\_TDT} = 4 \, \text{Mean}\{I_{dis}(pix)|I_{dis}(pix) > 0\}.$$

**Diameter_SDT:** average filament diameter estimated by Euclidian distance transformation values of $I_{binary.ovsp}$, but sampled using only positive pixels on $I_{sk}$, or mathematically as:

$$\text{Diameter\_SDT} = 2 \, \text{Mean}\{I_{dis}(pix)|I_{sk}(pix) = 1\}.$$

**Total connected element:** the number of connected components in the binary image. For images captured in the 2D mode, it is $Count.of.NNE(I_{binary.ovsp})$. In the 3D mode, the count of non-connected elements is given by the Skan library (Nunez-Iglesias et al., 2018) using $I_{sk}$ as the input. Total connected element is a nonstandardized intermediate index for the developer's use.

**Segment density:** the count of connected components in the binary image per length unit of the skeletonized filament. For images captured in the 2D mode, it is $Count.of.NNE(I_{binary.ovsp})/\sum length(x,y)$. In the 3D mode, both the count of non-connected elements and the length of the skeletonized filament are called from the Skan library using $I_{sk}$ computed from $I_{binary}$ as the input.

**Total branch:** the number of all graph theory-defined branches. $I_{sk}$ is obtained from $I_{binary.ovsp}$ for the 2D mode or $I_{binary}$ for the 3D mode, respectively, and is next processed by the Skan library. The total number of recognized branches is collected. Total branch is a non-standardized intermediate index for the developer's use.

**Total node:** the number of all graph theory-defined nodes. $I_{sk}$ is processed by the Skan library and the total number of recognized nodes is collected. Total node is a non-standardized intermediate index for the developer's use.

**Static branching activity:** the branching point count per unit length of skeletonized filament. $I_{sk}$ is obtained from $I_{binary.ovsp}$ for the 2D mode or $I_{binary}$ for the 3D mode, respectively, and is next input into the Skan library. The total number of type-3 and type-4 branches (i.e., graph theory-defined branches at least greater than three branches at one node, or "biologically defined branch"; Nunez-Iglesias et al., 2018) is collected and then divided by the length of the skeletonized filament.

**Local anisotropy:** We performed a local averaging of the filament alignment tensor, which is constructed as follows. First, we calculate the unit direction vector for each straight filament segment $g_i$. Then, the covariance matrix for each segment is obtained from the following equation:

$$t_i = g_i g_i^t.$$

This rank-2 tensor is independent of the orientation of the line segment and can thus be averaged over a region containing a collection of unoriented line segments. We weighed each filament tensor in a circular/spherical neighborhood by the length of every filament to produce a smoothed tensor field. The eigenvector corresponding to the largest eigenvalue indicates the primary orientation of filaments in this region. The difference between the maximum and the minimum eigenvalues is an indicator of the anisotropy in this region. If all the eigenvalues are the same, the indicator is 0, which implies an isotropic region. If the eigenvalues other than the maximum are all nearly 0, all the filaments in this region are parallel to each other. In this case, they are all aligned with the maximum eigenvector, the dominant filament direction of this region.

### Judging actin-like false-positive filaments

For visual comparison of algorithms, we judge whether a false-positive filament is actin-like by its shape and connectivity to matched filaments. First, we define a slim index (SI), which is the ratio of the skeleton length/perimeter of a segment. If a false-positive segment does not connect to any known matched pixel, then those with $SI > 0.3$ are judged as actin-like. Otherwise (if a false-positive segment connects to know matched pixel), we additionally require that a segment containing actin-like false-positive pixels should have higher SI and skeleton length versus the segment without these pixels, plus the 0.3 SI threshold.

### Simulation model of 3D artificial actin image

For each of the 31 training samples $I(x,y,z)$, the binary image $I_{binary}$ and skeleton image $I_{sk}$ were previously obtained via ILEE, as described above. Three independent fractions reflecting artificial actin ($I_{atf.a}$), artificial ground noise ($I_{atf.gn}$), and artificial diffraction noise ($I_{atf.d}$) are simulated independently (Fig. S16).

To calculate $I_{atf.a}$, the brightness of the real actin ($I_a$) of all training samples is collected as:

$$I_a = I \circ I_{binary}.$$

Here, we define the "reference voxel" of a non-zero voxel in $I_a$ as its nearest voxel whose position has a positive value in $I_{sk}$. In other words, for each non-zero voxel of $I_a(x,y,z)$, the distance from $(x,y,z)$ to its nearest reference voxel ($P_R = (x_R, y_R, z_R)$) on the skeleton image is defined as $D_R(x,y,z)$, where

$$D_R(x,y,z) = \min_{I_{sk}(x_i,y_i,z_i)=1} \sqrt{(x-x_i)^2 + (y-y_i)^2 + (z-z_i)^2},$$

and

$$P_R(x,y,z) = \underset{I_{sk}(x_i,y_i,z_i)=1}{\arg\min} \sqrt{(x-x_i)^2 + (y-y_i)^2 + (z-z_i)^2}.$$

Meanwhile, the distance from the reference voxel to the nearest non-actin voxel, $D_{R2N}(x,y,z)$, is calculated as:

$$D_N(x,y,z) = \min_{I_a(x_i,y_i,z_i)=0} \left| P_R(x,y,z) - (x_i, y_i, z_i) \right|.$$

Next, the "relative eccentric distance," $D_{rlt}(x,y,z)$, and the relative brightness compared to the reference voxel, $I_{rlt}(x,y,z)$ are defined as:

and

$$D_{rlt}(x,y,z) = \frac{D_R(x,y,z)}{D_{R2N}(x,y,z)},$$

$$I_{rlt}(x,y,z) = \frac{I_a(x,y,z)}{I_a(P_R(x,y,z))}.$$

Note that $D_{rlt}$ and $I_{rlt}$ default to 0 whenever the denominator is 0. As $D_{rlt}$ is discrete, each possible value of $D_{rlt}$ can be counted

as $n(D_{rlt})$. Using the $I_{rlt}$ and $D_{rlt}$ of all voxels with $D_{rlt} \leq 1$ in the entire 31 images, a polynomial regression model between $D_{rlt}$ and $I_{rlt}$ is fitted using $\log_{10} n(D_{rlt})$ as weights and adjusted into a piecewise polynomial function:

$$\widehat{\mu}_{I.rlt}(D_{rlt}) = \begin{cases} 1, (D_{rlt} \leq 0.2) \\ -0.647 D_{rlt}^2 + 0.226 D_{rlt} + 0.989, (0.2 < D_{rlt} \leq 1.4) \end{cases}.$$

Similarly, the STD of the $I_{rlt}$ is also fitted as below:

$$\widehat{\sigma}_{I.rlt}(D_{rlt}) = -0.562 D_{rlt}^2 + 0.778 D_{rlt} + 0.065, \ (D_{rlt} \leq 1.4).$$

Finally, the artificial actin is generated using random values subject to:

$$I_{atf.a}(x, y, z) = \begin{cases} 0, if \ D_{rlt}(z, x, y) > 1.4 \ or \ I_{binary}(z, x, y) = 0 \\ I(z_R, x_R, y_R) \widehat{I}_{rlt}(D_{rlt}(x, y, z)) \big| \widehat{I}_{rlt} \sim N(\widehat{\mu}_{I.rlt}(D_{rlt}(x, y, z)), \\ \widehat{\sigma}_{I.rlt}^2(D_{rlt}(x, y, z))), otherwise. \end{cases}$$

where $N$ means normal distribution. The random values are clamped to [0,4095].

To calculate $I_{atf.gn}$, two different strategies are utilized to generate two groups of voxels. For the voxels positioned in the coarse background of $I$, we apply a fold of change that is subject to a normal distribution to each $I(x,y,z)$ as the new ground noise plus a Gaussian filter ($O_{Gauss}'$, see ILEE section of Methods) to neutralize the magnified gradient; for the rest beyond the coarse background, we gave random values that are subject to a Burr-12 distribution (API Ref. [3]) fitted by voxels in the coarse background of $I(x,y,z)$, that is:

$$I_{atf.gn}(x, y, z) = \begin{cases} 2^k I(x, y, z) * O_{Gauss}' \big| k \sim N(0, 0.25^2), if \ I(x, y, z) < t_{cbg} \\ b \big| b \sim Burr12(c, d, \mu_b, \sigma_b), otherwise. \end{cases}$$

where $c$, $d$, $\mu_b$, and $\sigma_b$ are the fitted parameters of the Burr-12 distribution using the training voxels $\{ (x,y,z) \mid I(x,y,z) \leq t_{cbg} \}$. Similarly, the random values are clamped to [0, 4095].

Third, for $I_{atf\_d}$, voxels of the positions in $I$ that are neither coarse background nor real actin are used to fit an exponential distribution:

$$I_{atf.d}(x, y, z) = \begin{cases} e \big| e \sim Exp(\mu_b, \sigma_b), if \ I_{atf.a}(x, y, z) = 0 \ and \ I(x, y, z) \geq t_{cbg} \\ 0, otherwise. \end{cases}$$

where $\mu_e$ and $\sigma_e$ are the fitted parameters of the exponential distribution using the training voxels $\{ (x,y,z) \mid I(x,y,z) \geq t_{cbg} \}$. Similarly, the random values are clamped to [0, 4095].

Finally, these fractions of brightness are summed to generate the complete artificial image $I_{atf}$, as:

$$I_{atf} = I_{atf.a} + I_{atf.gn} + I_{atf.d}.$$

## Other image thresholding approaches
Otsu (API Ref. [4]), triangle (API Ref. [5]), Li (API Ref. [6]), Yen (API Ref. [7]), Niblack (API Ref. [1]), and Sauvola (API Ref. [8]) thresholding algorithms are obtained from the "scikit-image" library (van der Walt et al., 2014).

## Library API
Python library API Ref. [1]-[14], for additional image thresholding/processing, statistical analysis, modeling, and simulations, are as follows:

API Ref. [1] Niblack thresholding: https://scikit-image.org/docs/stable/api/skimage.filters.html#skimage.filters.threshold_niblack; API Ref. [2] Distance transformation (DT): https://docs.scipy.org/doc/scipy/reference/generated/scipy.ndimage.distance_transform_edt.html; API Ref. [3] statistical analysis; data distribution fitting and simulation: https://docs.scipy.org/doc/scipy/reference/stats.html; API Ref. [4] Otsu thresholding: https://scikit-image.org/docs/stable/api/skimage.filters.html#skimage.filters.threshold_otsu; API Ref. [5] Triangle thresholding: https://scikit-image.org/docs/stable/api/skimage.filters.html#threshold-triangle; API Ref. [6] Li thresholding: https://scikit-image.org/docs/stable/api/skimage.filters.html#skimage.filters.threshold_li; API Ref. [7] Yen thresholding: https://scikit-image.org/docs/stable/api/skimage.filters.html#skimage.filters.threshold_yen; API Ref. [8] Sauvola thresholding: https://scikit-image.org/docs/stable/api/skimage.filters.html#skimage.filters.threshold_sauvola; API Ref. [9] Skeletonization: https://scikit-image.org/docs/stable/api/skimage.morphology.html#skimage.morphology.skeletonize; API Ref. [10] Visualization ("matplotlib"): https://matplotlib.org/stable/index.html; API Ref. [11] Visualization ("seaborn"): https://seaborn.pydata.org; API Ref. [12] T-test (multiple comparison): https://scikit-posthocs.readthedocs.io/en/latest/generated/scikit_posthocs.posthoc_ttest; API Ref. [13] modeling: https://scikit-learn.org/stable/modules/classes.html?highlight=linear+fit.

## Online supplemental material
Fig. S1 shows NNES (Non-connected negative elements scanning) that can identify the coarse background. Fig. S2 shows the performance of NNES-based adaptive global thresholding and its prediction model (3D). Fig. S3 shows enhanced 3D NNES-based adaptive coarse thresholding model for samples with greater diversity. Fig. S4 shows visualized explanation of the core ILEE algorithm. Fig. S5 shows a significant difference filter. Fig. S6 shows that the mean and STD gradient magnitude of ground noise is directly proportional to STD of the noise. Fig. S7 shows that the ratio of $x$-$y$ unit and z unit influences the proportional coefficient of $\sigma_{Noise}$–$\mu_G$ and $\sigma_{Noise}$–$\sigma_G$ relationship. Fig. S8 shows determination of the global gradient threshold. Fig. S9 shows enhanced global gradient threshold estimation model for 3D mode. Fig. S10 shows the impact of $K$ and the training for $K_2$ estimation. Fig. S11 shows visualized demonstration of concepts of cytoskeleton indices. Fig. S12 shows the stability of ILEE and other classic image thresholding approaches for cytoskeleton segregation in confocal images. Fig. S13 shows the visualized comparison of robustness of ILEE and other algorithms by segmentation accuracy. Fig. S14 shows the quantificational comparison of robustness of ILEE and other algorithms by segmentation accuracy. Fig. S15 shows the quantificational comparison of robustness of ILEE and other algorithms by index rendering stability. Fig. S16 shows an illustrated introduction of the actin image simulation model to generate 3D cytoskeleton images with ground truth. Fig. S17 shows the comparison of ILEE vs MGT over stability and capability to distinguish biological differences among real biological samples. Fig. S18 shows the correlation of occupancy and linear density. Fig. S19 shows the

performance of ILEE on other type of biological sample. Fig. S20 shows that ILEE and the human eye have different tendencies to judge the topological structure of the cytoskeleton, especially between two bright bundles.

## Acknowledgments

We would like to thank Dr. Yi-Ju Lu (National Taiwan University), Dr. Silke Robatzek (Ludwig-Maximilians University—Munich), and Dr. Weiwei Zhang (Purdue University) for generously providing raw images of actin and tubulin for algorithm development and cytoskeleton analysis. We would like to thank Dr. Yi-Ju Lu and Dr. Masaki Shimono (University of Nevada-Reno) for providing individual MGT evaluations for our experiments. We would like to thank Rongzi Liu (University of Florida) for providing critical advice on statistical analysis and modeling. We would like to thank Dr. Richard Neubig (Michigan State University) and Dr. Noel Day (Zoetis), Dr. Brittni Kelley (Michigan State University), and Emily Ribando-Gros (Michigan State University) for critical advice on manuscript preparation. We would like to thank Sarah Khoja (Michigan State University) for improving the reading experience of the ILEE_CSK documentation website.

Research in the laboratory of B. Day was supported by grants from the National Science Foundation (MCB-1953014) and the National Institutes of General Medical Sciences (1R01GM125743). Research in the laboratory of Y. Tong is supported by a grant from National Science Foundation (III-1900473).

The authors declare no competing financial interests.

Author contributions: P. Li, Z. Zhang, B. Day, and Y. Tong participated in the design and conception of ILEE pipeline and experiments. P. Li and Z. Zhang developed the ILEE algorithm and ILEE_CSK library. P. Li and B.M. Foda performed the experiments. P. Li, Z. Zhang, B. Day, and Y. Tong analyzed and interpreted the data. P. Li, Z. Zhang, and B. Day wrote the manuscript. All authors actively edited and agreed on the manuscript before submission.

Submitted: 6 March 2022

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

# Supplemental material

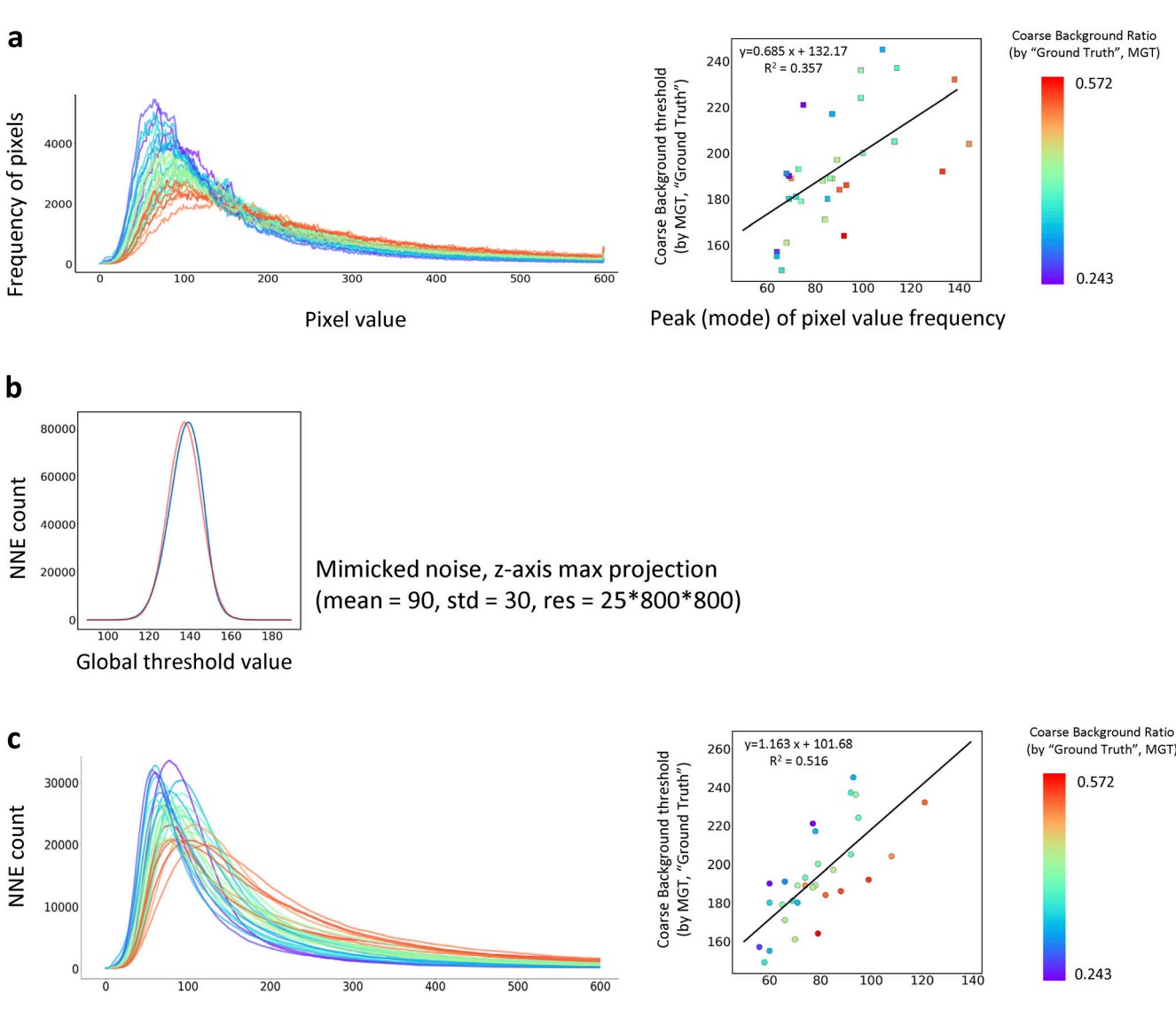

Figure S1.    **NNES (Non-connected negative elements scanning) can identify the coarse background. (a)** Performance of a brightness-based adaptive global thresholding using peak-of-frequency of brightness as a feature value. A random set of 31 actin image of 25*800*800 in our database are used to test the performance. Left, the frequency distribution curves of the pixel brightness of the 31 training samples; right, the linear correlation of MGT (manual global threshold)-determined coarse background (as ground truth) vs. the threshold at peak-of-frequency. This method cannot identify the peak accurately due to the turbulent curve and the results do not have good correlation. **(b)** NNE count of pure ground noise has a normal-like distribution. A ground noise image was mimicked by a random normal distribution (mean = 90, std = 30) into a 25*800*800 array. The maximum projection is conducted by choosing the maximum value of the third axis to make an 800*800 image (distribution shown as blue), and its mean and STD are calculated to make a true normal distribution (shown as red), both of which finely overlap. **(c)** Performance of the NNES-based adaptive global thresholding and its prediction. The same samples of (a) is used for the assay. Left, the NNES curves reflecting the relationship between the threshold and NNE count; right, correlation of ground truth coarse background evaluated by MGT vs. peak of NNE count for individual samples. Comparing to the brightness-based methods, NNES has a much smoother shape that enables the utilization of peak as a feature value. Also, NNES has a more robust correlation to coarse background determined by MGT.

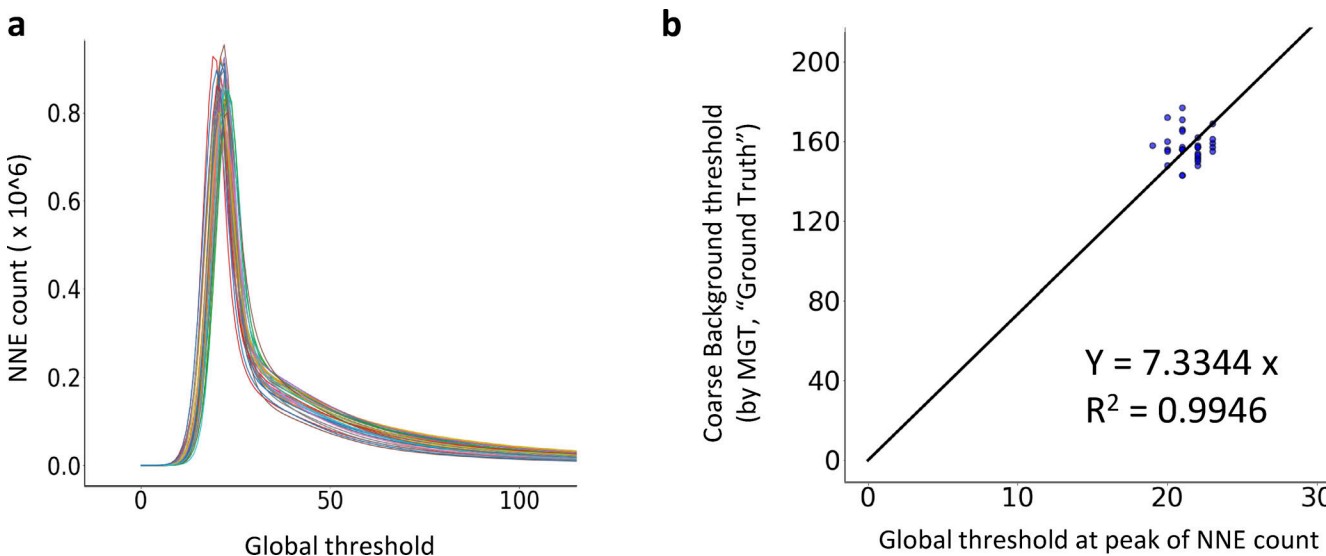

Figure S2.  **Performance of NNES-based adaptive global thresholding and its prediction model (3D). (a)** The NNES curve of the same sample set as Fig. S1. While they vary in actin features (density, bundling, etc.), they have very similar NNES shape. **(b)** The correlation of coarse background (evaluated by an approach mimicking 2D MGT) vs the peak of NNE count for individual samples and corresponding coarse background prediction model. As suggested by NNES curve shape, they have very similar peak as well as coarse background, which represents the norm of sensor performance in 3D imaging (2D projection therefore is distorted by information loss). Therefore, we directly used a simple proportional function to establish the regression model.

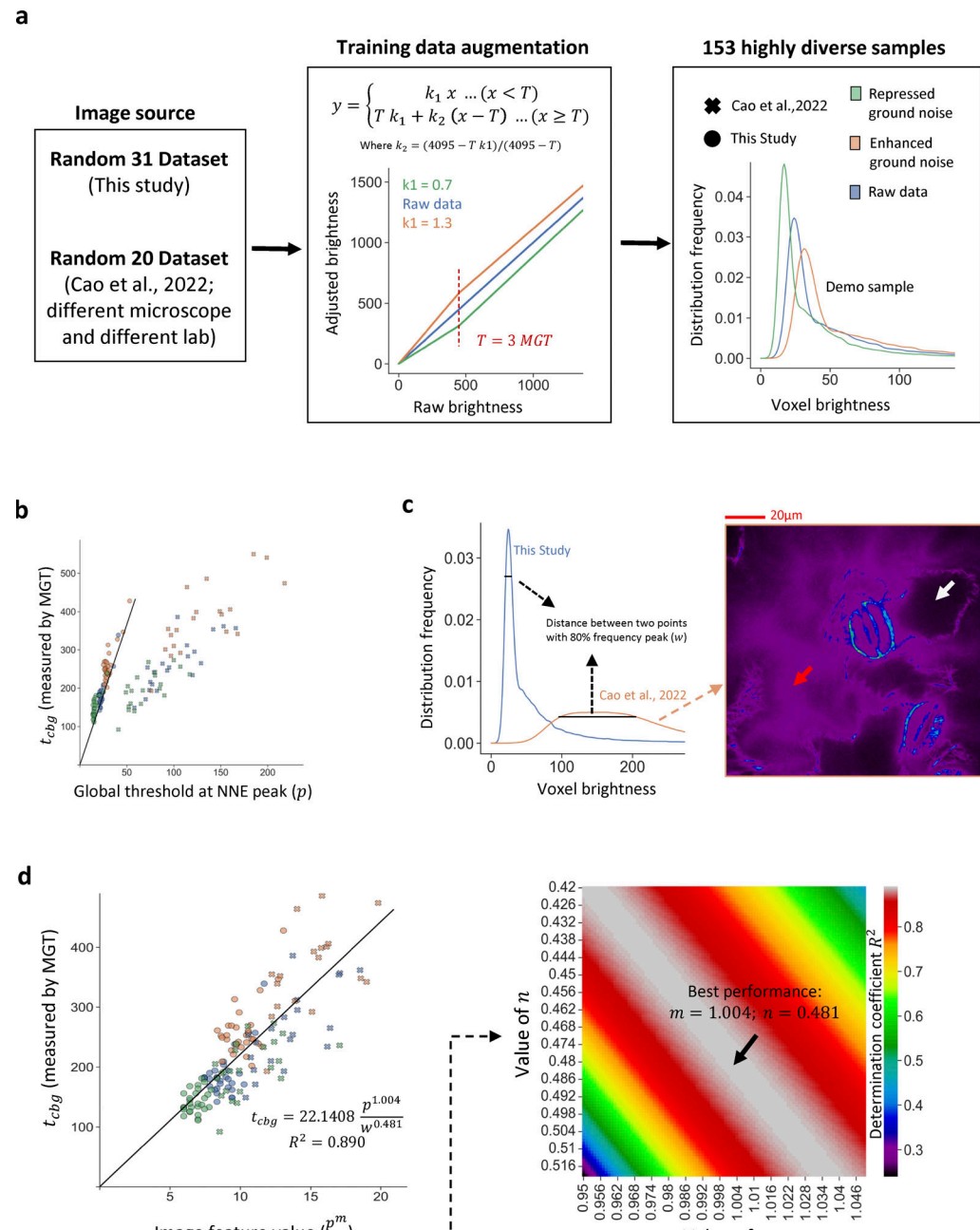

Figure S3. **Enhanced 3D NNES-based adaptive coarse thresholding model for samples with greater diversity.** As the original training dataset of Fig. S2 may not have satisfactory variation of threshold-at-peak range, we introduced published actin image samples (Cao et al., 2022) from a different microscope device in a different lab and applied image augmentation to generate a greatly diverse dataset for model training, resulting an enhanced 3D coarse thresholding model. **(a)** Scheme of datasets. 31 and 20 random samples, by different microscope machines and photographers, are collected. The total 51 images are then processed through a linear LUT transformation (see formula) to repress ($k_1 = 0.7$) or enhance ($k_1 = 1.3$) the ground noise (see altered brightness distribution, right panel). **(b)** the correlation between global threshold at NNE peak (p) and coarse global threshold ($t_{cbg}$) measure through MGT, of the total training dataset. The black line suggests the "1st edition" model (Fig. S2) used for the rest of this study, which fits very well with all raw and augmented samples from our lab collection but not with the other dataset. **(c)** Differences in brightness distribution between datasets. We inspected both datasets to investigate why they have different slope (see b). It is determined that our lab collection have a single-pattern, condensed distribution of ground noise brightness, while the Cao dataset has a wide-spread, dispersive peak. This is because the Cao dataset has multi-layer ground noises, as some of the cells have higher baseline brightness potentially due to out-of-focus light or non-binding actin fluorescence marker. Arrows with different color indicates two background areas with different ground noise level. **(d)** Performance of enhanced 3D coarse global threshold model $t_{cbg} = k_{cbg}\frac{p^m}{w^n}$. As the ground noise concentration level have an impact to the relationship between p and $t_{cbg}$, we measured a second feature value—the distance between two points with 80% top frequency (w; see c)—for modeling improvement. To introduce of w, we constructed a new model $t_{cbg} = k_{cbg}\frac{p^m}{w^n}$, where $k_{cbg}$, m, and n are trained constants. The left panel shows the correlation between $\frac{p^m}{w^n}$ and $t_{cbg}$ at the best performance ($t_{cbg} = 22.1408\frac{p^{1.004}}{w^{0.481}}$); the right panel shows the relationship between m, n, and correlation coefficient $R^2$. Note: This model will be provided as the default model for 3D $t_{cbg}$ estimation in the PyPI library release but not used for the rest of the study unless specifically announced.

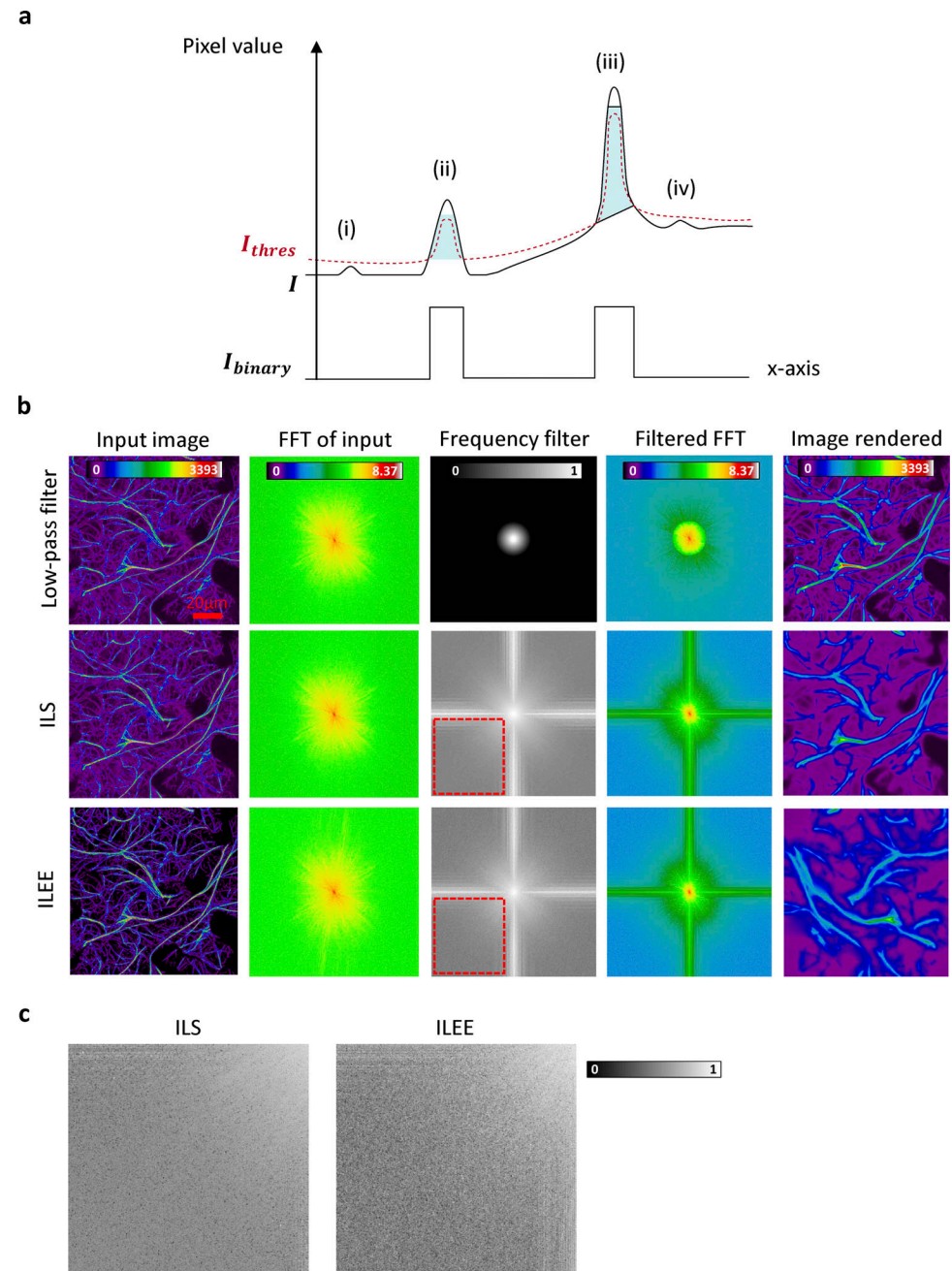

Figure S4. **Visualized explanation of core ILEE algorithm.** The core strategy of ILEE thresholding is explained from the perspective of time domain (i.e., image itself) and frequency domain (i.e., after Fourier transformation, FFT). **(a)** Schematic diagram describing how ILEE generates local threshold based on detected edges. For the purposed of simplicity, we use a "1D image" for demonstration. Suppose there is an example grayscale image ($I$) with 4 peaks of pixel value (i, ii, iii, iv); the higher peaks ii and iii are true cytoskeleton fluorescence but the lower peaks i and iv are random noise. ILEE first identify edges of cytoskeleton—area with a gradient magnitude higher than a computed threshold (marked cyan). Then these edges are used as "reference values" to generate a threshold image ($I_{thres}$, in red color) that smoothly links all the reference area. Finally, the bona-fide cytoskeleton will be defined as area where $I$ is higher than $I_{thres}$. The areas not selected as edges will not be referred, which means true florescence with locally high values (ii, iii) are selected and background with locallay low values (i, iv) are excluded, regardless of whether the local level is generally low (i, ii) or high (iii, iv). **(b)** The comparison of effect of ILEE, classic low-pass filter, and implicit Laplacian smoothing (ILS) on image frequency domain. For low-pass filter, the input image ($I$) is transformed into frequency domain pattern by FFT, and we artificially define a filter where 0–1 indicates the passing rate of each frequency fraction. We pass the frequency pattern through the filter to render the filtered FFT pattern and restore the image by reverse FFT. For ILS and ILEE, the input image is transformed to frequency domain pattern by FFT; on the other side, the result of ILS and ILEE are pre-computed and transformed to frequency domain pattern as the "filtered FFT." The (equivalent) frequency filter is deduced by subtracting filtered FFT from FFT of input and make the value relative to the FFT of input, which is comparable to the classic low-pass filter. ILS and ILEE have more fractions low frequency fractions on either x- or y-direction, which are potentially thick line cytoskeleton structures. Scale bar = 20 μm. **(c)** Comparison of high-frequency filtering of ILS and ILEE. The red rectangle in subplot (b) is maximized to present the detail. ILEE has uneven but well-directed selection of high frequency fractions because ILEE particularly preserves the cytoskeleton edge and tends to neglect the coarse background.

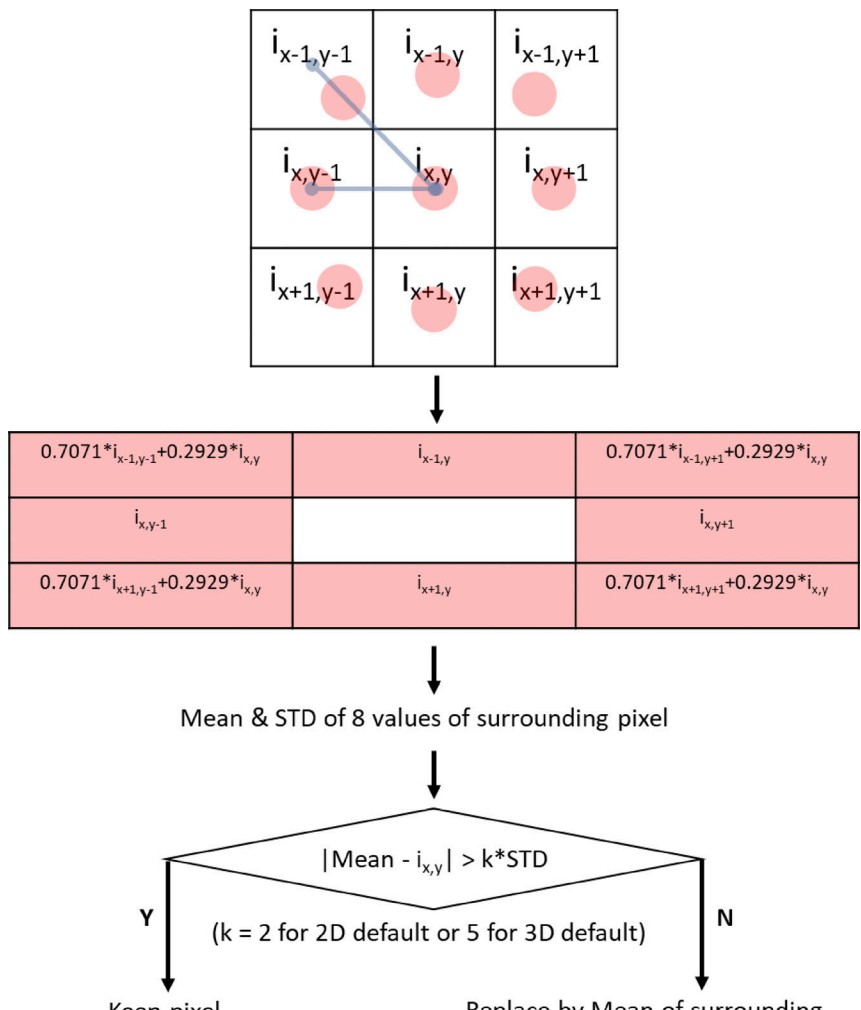

**Figure S5. Significant difference filter.** Significant difference filter changes a pixel to the mean of its surrounding pixel if the object pixel is "significantly different" from its surrounding pixels (shown as red dots in the figure). For the diagonal pixels only, we use an estimated by linear interpolation of the position that is 1-pixel distance to the referred pixel. "Significant difference" is defined as a difference greater than twofold of standard deviation (STD) of surrounding pixels for 2D mode and fivefold for 3D mode. The rationale behind this is that there is no detectable independent actin element as tiny as a 1-pixel element, so a 1-pixel region that is significantly different from its surrounding tends to be noise or bias due to z-axis projection.

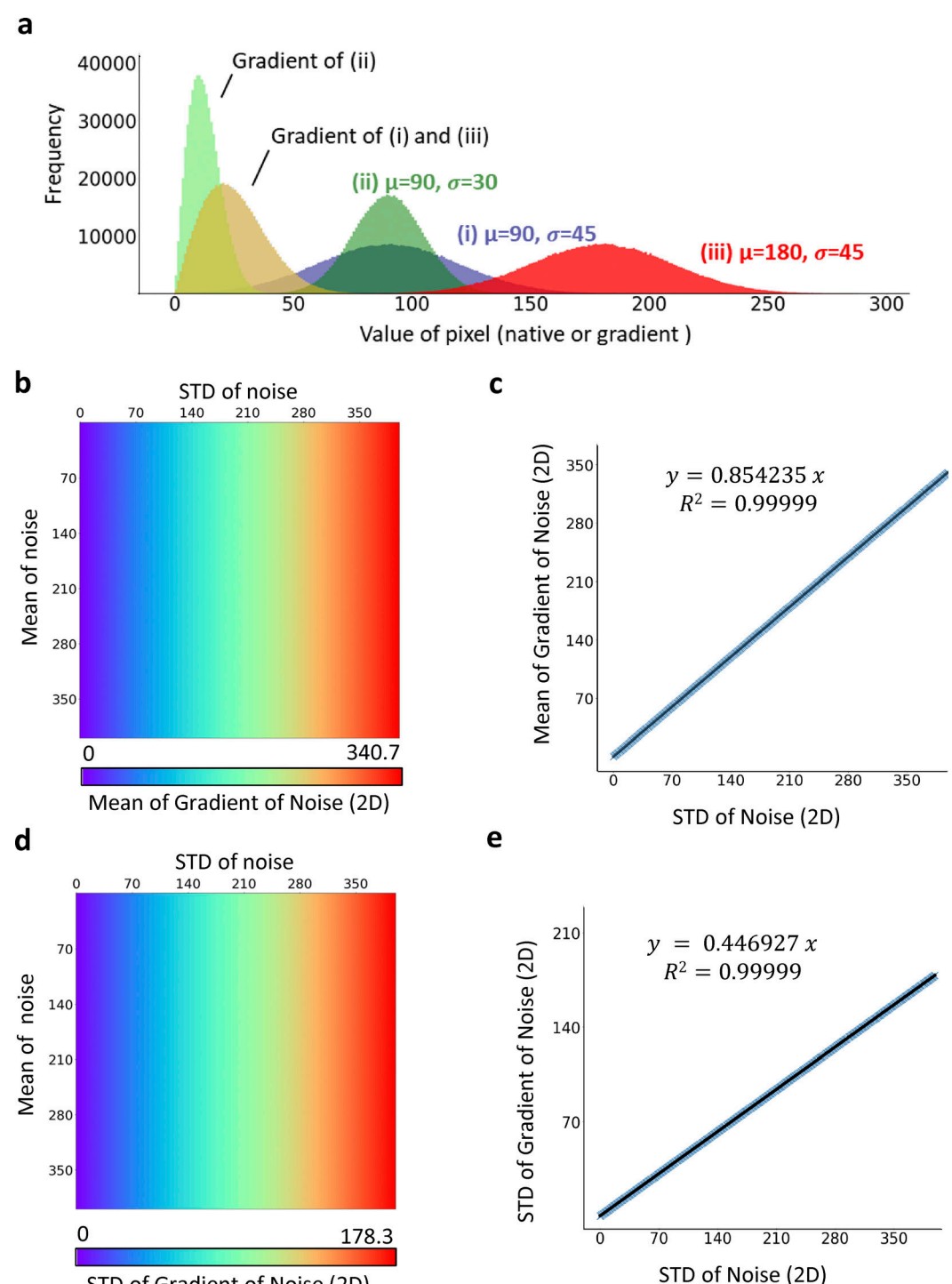

Figure S6. **The mean and standard deviation (STD) of gradient magnitude of ground noise is directly proportional to the STD of noise.** To investigate the mathematical relationship between statistical distribution of ground noise (represented by coarse background located by NNES) and the gradient magnitude of ground noise, we used random normal array that mimics the data structure of our image sample and generated its Scharr gradient image for statistical analysis. **(a)** Normal-like distribution of Scharr gradient. Simulated 800*800 ground noise image subject to normal distribution (i), (ii), and (iii) are processed by Scharr operator and the gradient distribution is shown. The gradient distribution is not influenced by the mean yet by the STD of native data of the simulated ground noise. **(b and c)** The mean of noise gradient is not influenced by the mean of the noise, but the STD of noise. The mean of noise gradient is directly proportional to the STD of native noise. **(d and e)** The STD of the noise gradient is only influenced by STD of the noise as well. The STD of the noise gradient is directly proportional to the STD of native noise.

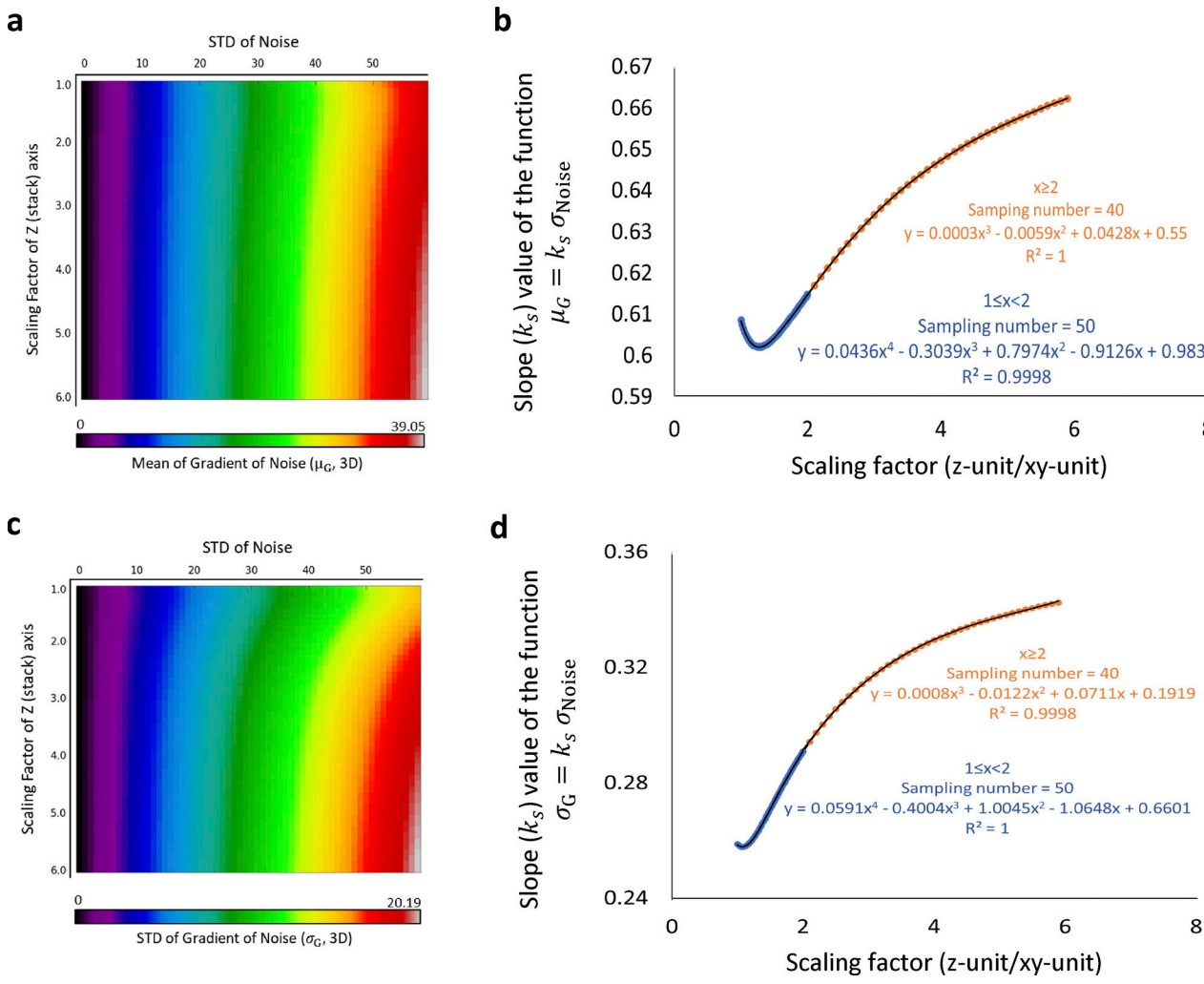

Figure S7.   **The ratio of x-y unit and z unit influences the proportional coefficient of $\sigma_{Noise}$-$\mu_G$ and $\sigma_{Noise}$-$\sigma_G$ relationship.** Using a strategy similar to what is presented in Fig. S5, we simulated 3D normal ground noise images (800*800*25) with a variable standard deviation (STD) and mean fixed to 90. Next, the z-axis was interpolated by the fold of a scaling factor, defined as z-unit/xy-unit, the second variable. Like the 2D data structure, at a given scaling factor, the mean and STD of noise gradient is proportional only to the STD of native noise. Then, we adopt a mathematical model $\mu_G$ or $\sigma_G$ = $k_s$(Scalingfactor)$\sigma_{Noise}$ to accurately describe the numeric relationship between the gradient and native value of background noise. By a piecewise polynomial regression, we obtain a prediction model that accurately ($R^2$ = 1) calculate $k_s$ and therefore determine the relationship between mean/STD of the noise gradient and the native noise values. **(a and b)** the relationship between mean of noise gradient and STD of noise. **(c and d)** the relationship between mean of noise gradient and STD of noise.

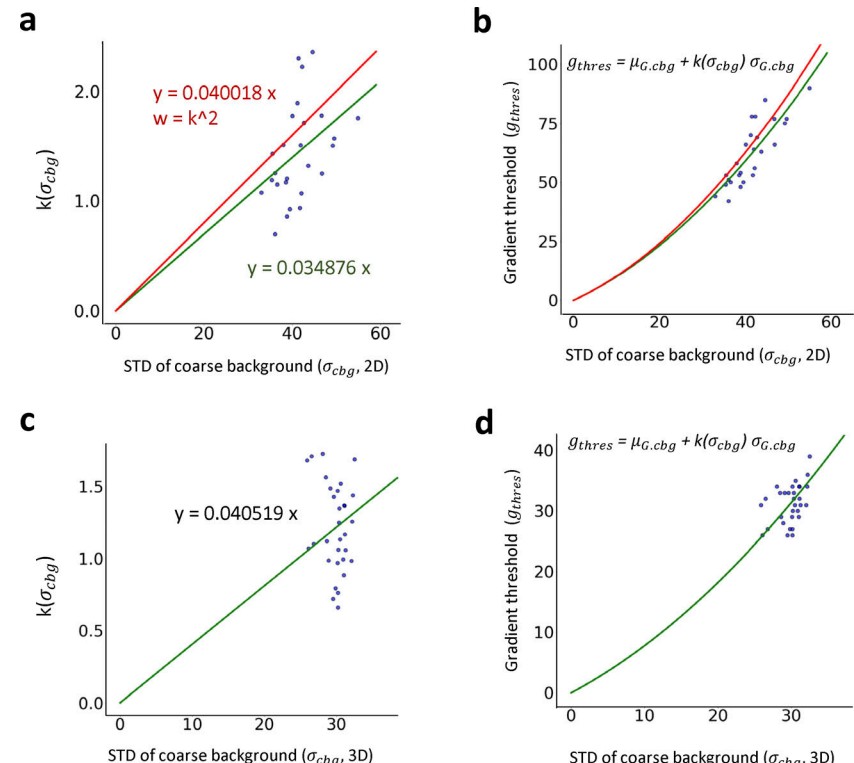

Figure S8. **Determination of global gradient threshold.** In order to determine the global gradient threshold using estimated mean and standard deviation (STD) of coarse background gradient as an input of ILEE, we constructed a non-linear estimation model $g_{thres} = \mu_{G.cbg} + k(\sigma_{cbg}) \sigma_{G.cbg}$. The global gradient threshold of 30 samples with 800*800*25 resolution randomly collected from our actin image database were evaluated manually (similar to MGT), and the mean and STD of gradient values in coarse background are calculated following the approach described in the Methods and Fig. S5. For the 2D mode, images were first projected to 800*800 resolution. Then, the coefficient k of each sample was calculated and corelated with STD of the coarse background. **(a)** The correlation of STD of coarse background and k. Green, standard mode, where each sample have the similar weight of regression; Red, conservative mode (default of this paper), where a weight equal to $k^2$ is applied. **(b)** The $g_{thres}$-$\sigma_{cbg}$ relationship restored from (a). **(c and d)** Similar to (a and b), but the reference global gradient threshold were manually evaluated using a 3D visualizing interface.

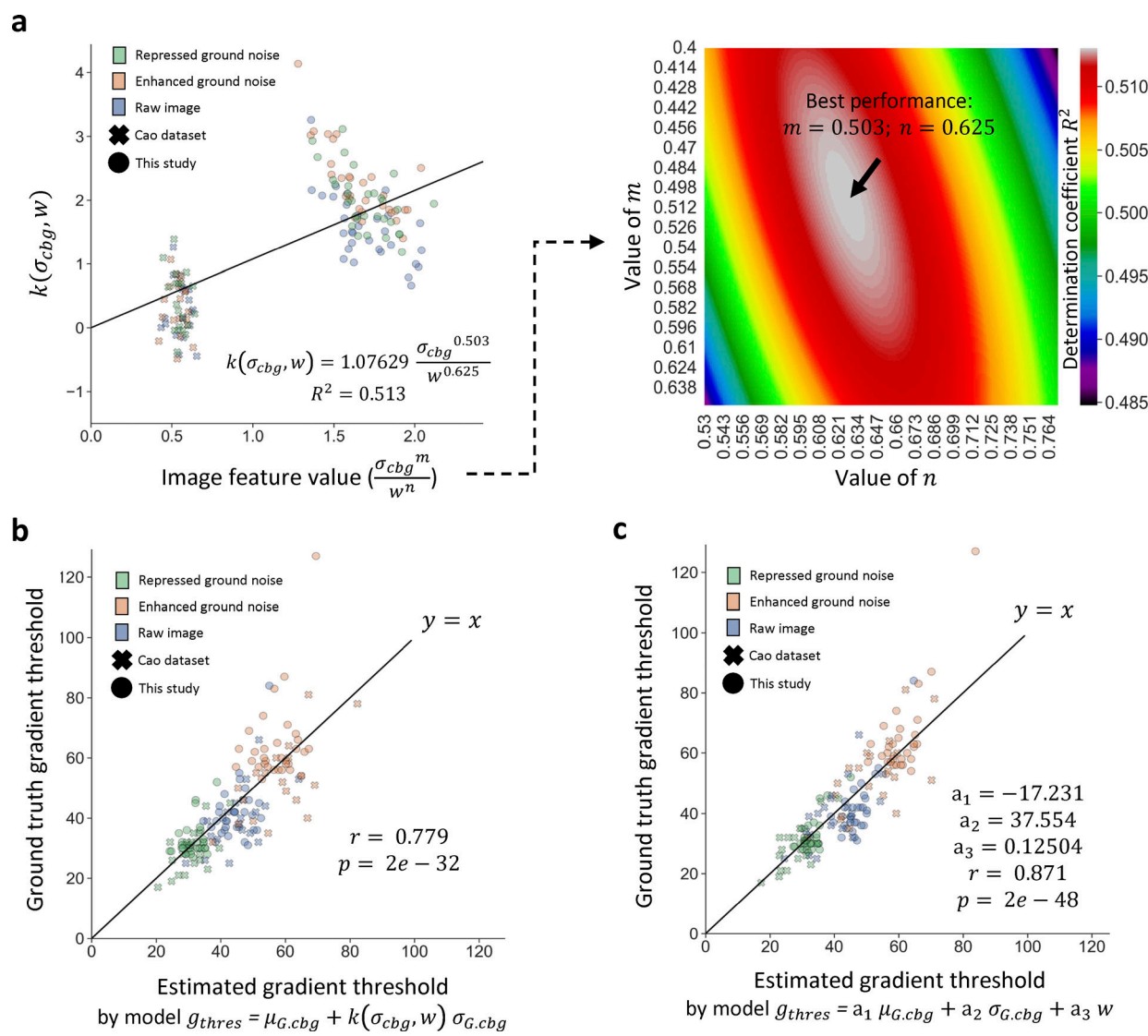

**Figure S9. Enhanced global gradient threshold estimation model for 3D mode.** Similar to Fig. S3, the training dataset for $g_{thres}$ estimation model for 3D mode (Fig. S8, c and d) may have limited variation. To test and improve the applicability of the model, we introduced a large dataset from a different microscope device by a different lab with image augmentation (see Fig. S3 a; $n$ = 153) and constructed two enhanced $g_{thres}$ estimation models for 3D mode. **(a)** Reconstruction of model $g_{thres}$ = $\mu_{G.cbg}$ + $k\sigma_{G.cbg}$ by introducing $w$ (see Fig. S3 c) to improve compatibility to diverse images. $k$ is replaced into a multi-variant function $k(\sigma_{cbg}, w) = a_k \frac{\sigma_{cbg}^m}{w^n}$ to apply the impact of the dispersion level of ground noise. Therefore, we name the model $g_{thres}$ = $\mu_{G.cbg}$ + $k(\sigma_{cbg}, w)$ $\sigma_{G.cbg}$ as "$\sigma_{cbg}$-$w$ interaction model." Left panel displays the model with the best parameters; right panel shows the impact of $m$ and $n$ on model performance. **(b)** Performance of the $\sigma_{cbg}$-$w$ interaction model ($g_{thres}$ = $\mu_{G.cbg}$ + $k(\sigma_{cbg}, w)$ $\sigma_{G.cbg}$) by accuracy. The correlation of the estimated $g_{thres}$ and MGT-determined "ground truth" $g_{thres}$ is measured by Pearson coefficient (r) and P-value (p). The $\sigma_{cbg}$-$w$ interaction model has a satisfactory performance with $r$ = 0.779. **(c)** Performance of the multivariate linear model $g_{thres}$ = $a_1\mu_{G.cbg}$ + $a_2\sigma_{G.cbg}$ + $a_3w$. We constructed an alternative model by a direct multivariate linear regression combining the impact of $\mu_{G.cbg}$, $\sigma_{G.cbg}$, and $w$. This model has a further improved performance with $r$ = 0.871, for users to select according to their demand. The two models are available in the library release with the $\sigma_{cbg}$-$w$ interaction model set as default, but they are not used for the rest of the study unless specifically announced.

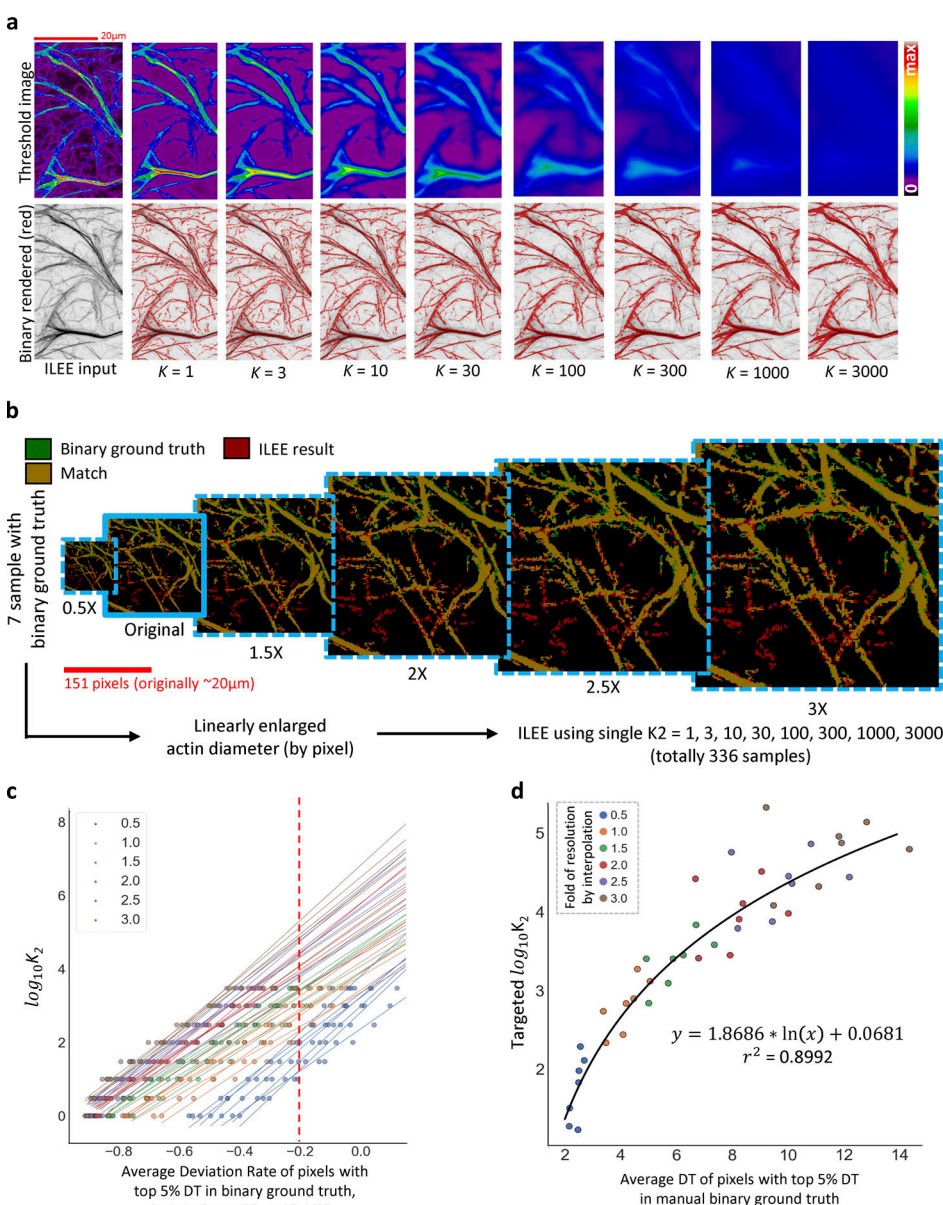

**Figure S10. The impact of *K* and the training for *K₂* estimation.** In order to learn the impact $K$ (implicit Laplacian smoothing coefficient) and determine the $K$ value for a give batch of image sample, we constructed a training database and developed a non-linear estimation model. **(a)** The influence of different $K$ to rendered threshold image and binary result by ILEE. A low $K$ renders a threshold image that assimilates the object sample with finer filament structure (preserved high frequency information), but tends to underestimate the thickness of bright and thick filament; as $K$ increases, less local detail are preserved, and the rendered binary image losses thin and faint filaments but get more accurate for thick and bright filament. There is a trade-off over the performance over faint/thin and bright/thick for a single $K$. Therefore, we used the full-outer-join result of a small $K_1$ and a high $K_2$ (Fig. 3 b and corresponding main text). **(b)** Construction of $K_2$ training database. Initially, we planned using the 7 images with hand-portrayed binary ground truth of filament fraction, but these sample have relatively limited range of filament size by pixel, so we decided to expand out sample pools using these data. Each sample is bicubically interpolated into the resolution of 0.5-, 1,5-, 2-, 2.5-, and threefold of the original and added to the database. Their corresponding binary imaged are converted to float data type with 0.0 and 1.0 to process the same bicubic interpolation, with the pixels over 0.42 are defined as True and False if not. Using this approach, the judgement of matching between the ground truth and ILEE result did not have significant change, as shown. Finally, each sample will be processed by ILEE with a single $K_2$ ranging from 1 to 3000, rendering totally 336 binary images as the training database. **(c and d)** Training algorithm. In (c), we converted the ILEE samples to a feature value—estimated $K_2$ with −0.2 of average deviation rate of pixels with top 5% DT in binary ground truth. Specifically, for each original or interpolated image sample (shown by independent lines, where different color represents different fold of interpolation), ILEE binary results using various single $K_2$ are compared with corresponding ground truth image to calculate the deviation rate, defined as the fold of difference of Euclidian distance transformation (DT) of ILEE result vs ground truth relative to ground truth. The averaged deviation rate of pixels with top 5% DT are taken as a feature value to represent thick filament. Next this feature was correlated to the corresponding $K_2$, and linear regressions were conducted to estimate the $\log_{10}$ of the optimal $K_2$ that renders −0.2 for average deviation rate for each sample. Finally, the estimated optimal $K_2$ and the average DT of pixels with top 5% DT of each sample were utilized (shown in d) to generate an exponential regression model. This model covers the vast majority (if not all) of possible highest filament thickness rendered by confocal microscope images of cytoskeleton. The average DT of pixels with top 5% DT estimated by Niblack thresholding were used as an independent variable for optimal $K_2$ estimation.

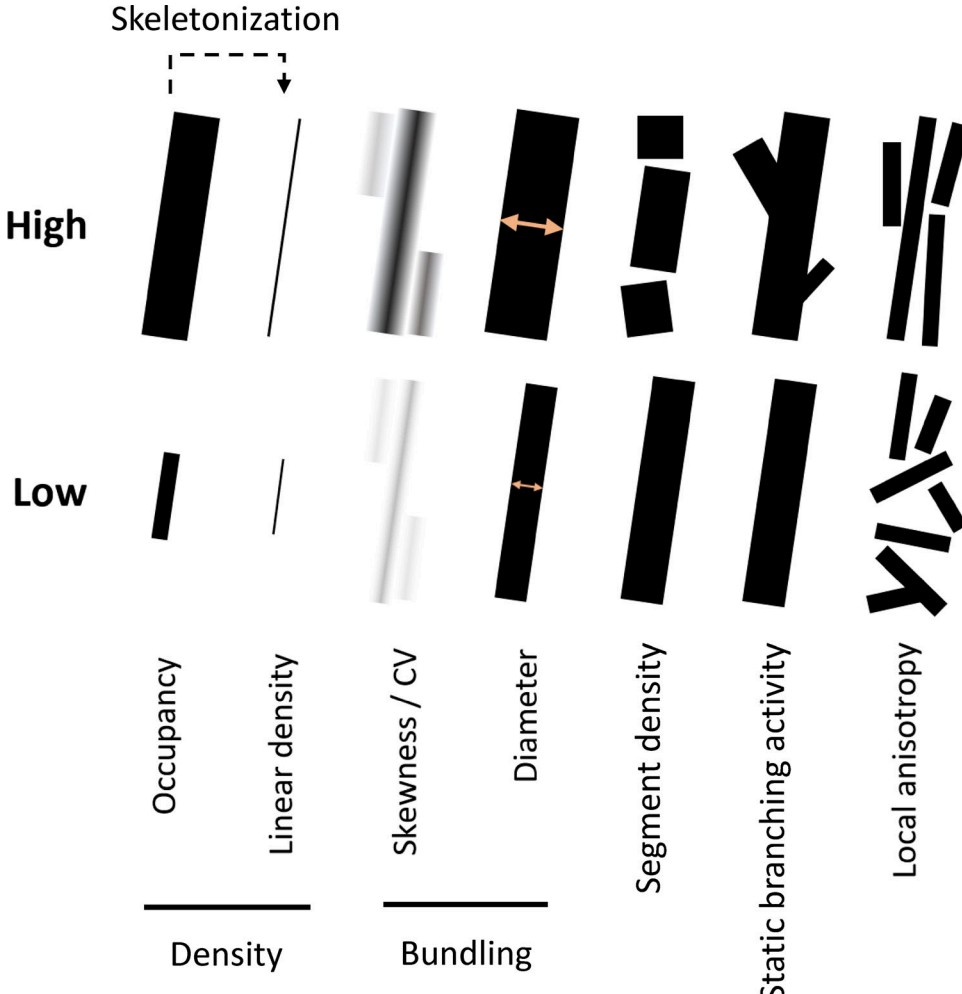

Figure S11. **Visualized demonstration of concepts of cytoskeleton indices.** Major cytoskeleton indices involved in this research are explained by showing schematic cytoskeleton pieces representing high vs low value of the indices. Skewness/CV, as well as diameter (TDT/SDT) have similar concept and reflect similar features of cytoskeleton image, with minor difference in their mathematical definition (see Materials and methods); therefore, their demonstrations are merged.

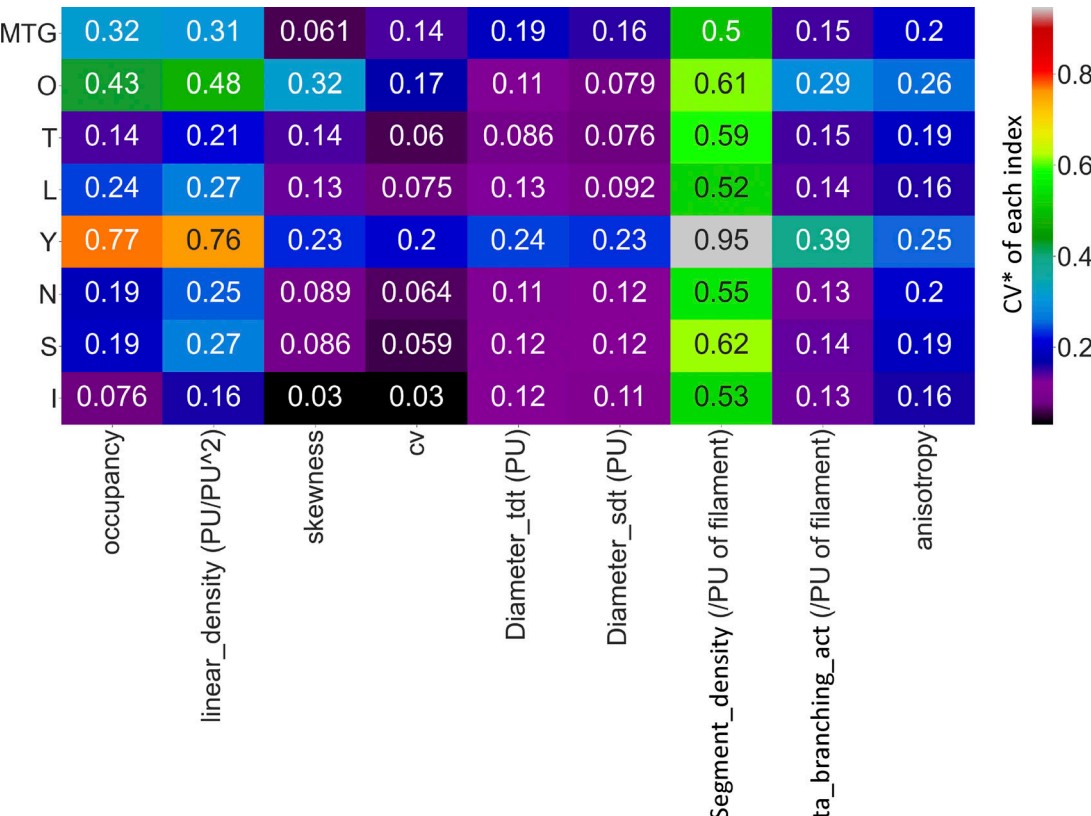

Figure S12. **The stability of ILEE and other classic image thresholding approaches for cytoskeleton segregation in confocal images.** The coefficient of variation (CV*) of each index rendered by each approach in Fig. 4 d were calculated and visualized as a heatmap. O, Ostu; T, Triangle; L, Li; Y, Yan; N, Niblack; S, Sauvola; I, ILEE. ILEE has a dominantly low CV* (i.e., high stability) over diverse sample for *occupancy, linear density, skewness, CV,* or one of the lowest for other indices. Note that the CV* of computed indices in the figure and the fluorescence *CV* of a sample, which is a cytoskeletal index, are different concept.

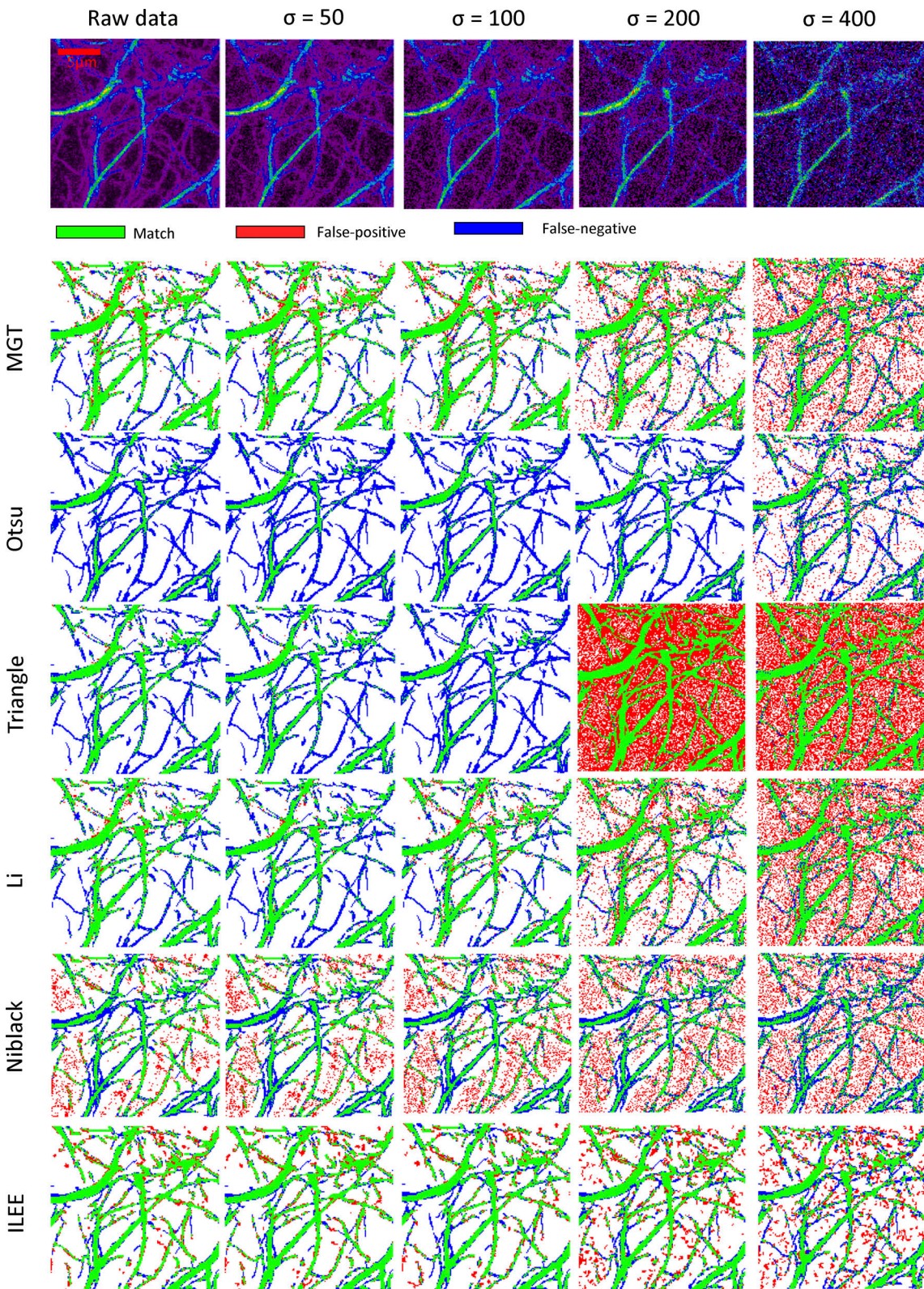

Figure S13. **The visualized comparison of robustness of ILEE and other algorithms by segmentation accuracy.** The raw demonstration image of Fig. 4 a is added with a series of gaussian noise (μ = 0, σ as variable). The binary images are computed by selected algorithms including ILEE and compared with manually portraited ground truth, and pixels of match, false-positive, and false-negative are presented. By visual observation, all of the algorithms are stable when the σ of noise is within 100 (3–4% of max of dynamic range); three of the better performed algorithm, ILEE, MGT, and Li, remain accurate, among which ILEE has the best coverage of ground truth. When σ of noise is higher than 100, all algorithms become visibly unstable. While MGT and Li tend to have single pixel errors while ILEE tends to result in block-shaped errors that are less in number and mimics the thickness of the cytoskeleton in shape, which indicates that the indices derived from ILEE are potentially more accurate at high noise conditions.

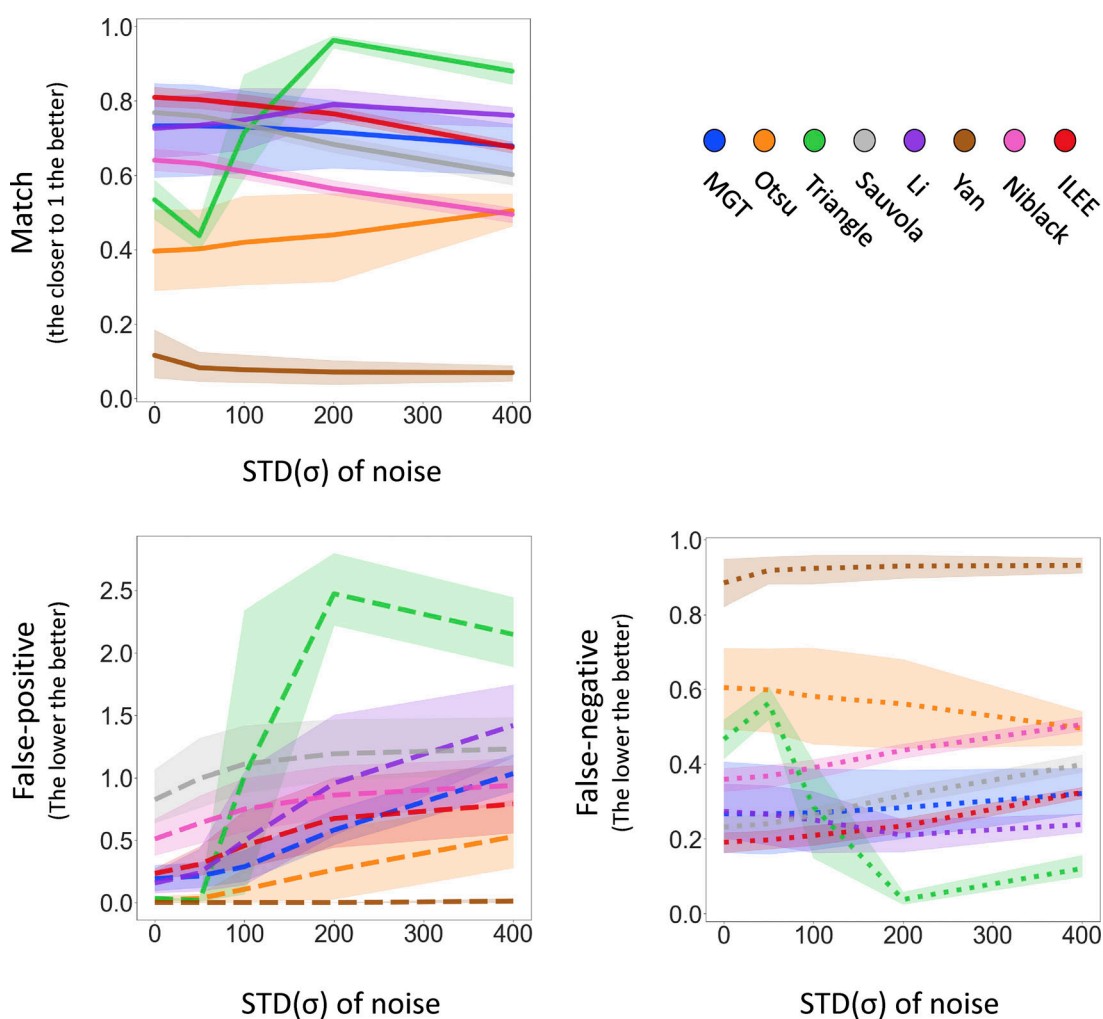

Figure S14.   **The quantificational comparison of robustness of ILEE and other algorithms by segmentation accuracy.** The dataset of Fig. 4 is added with a series of gaussian noise (μ = 0, σ as variable). Since Gaussian noise is random and potential unstable, each sample-noise combination is technically repeated by 12 times and the averaged result is used. The transparent area with light color indicates 95% confidence interval of each algorithm. The binary images rendered by different algorithms are compared with manually portraited ground truth to count the pixels that are matched (ideally 1.0), false-positive (ideally 0), and false-negative (ideally 0). Generally speaking, ILEE is the best-performing algorithm with high match, low false-positive, and low false-negative rate at both low and high noise. Other algorithms have one or more flaws.

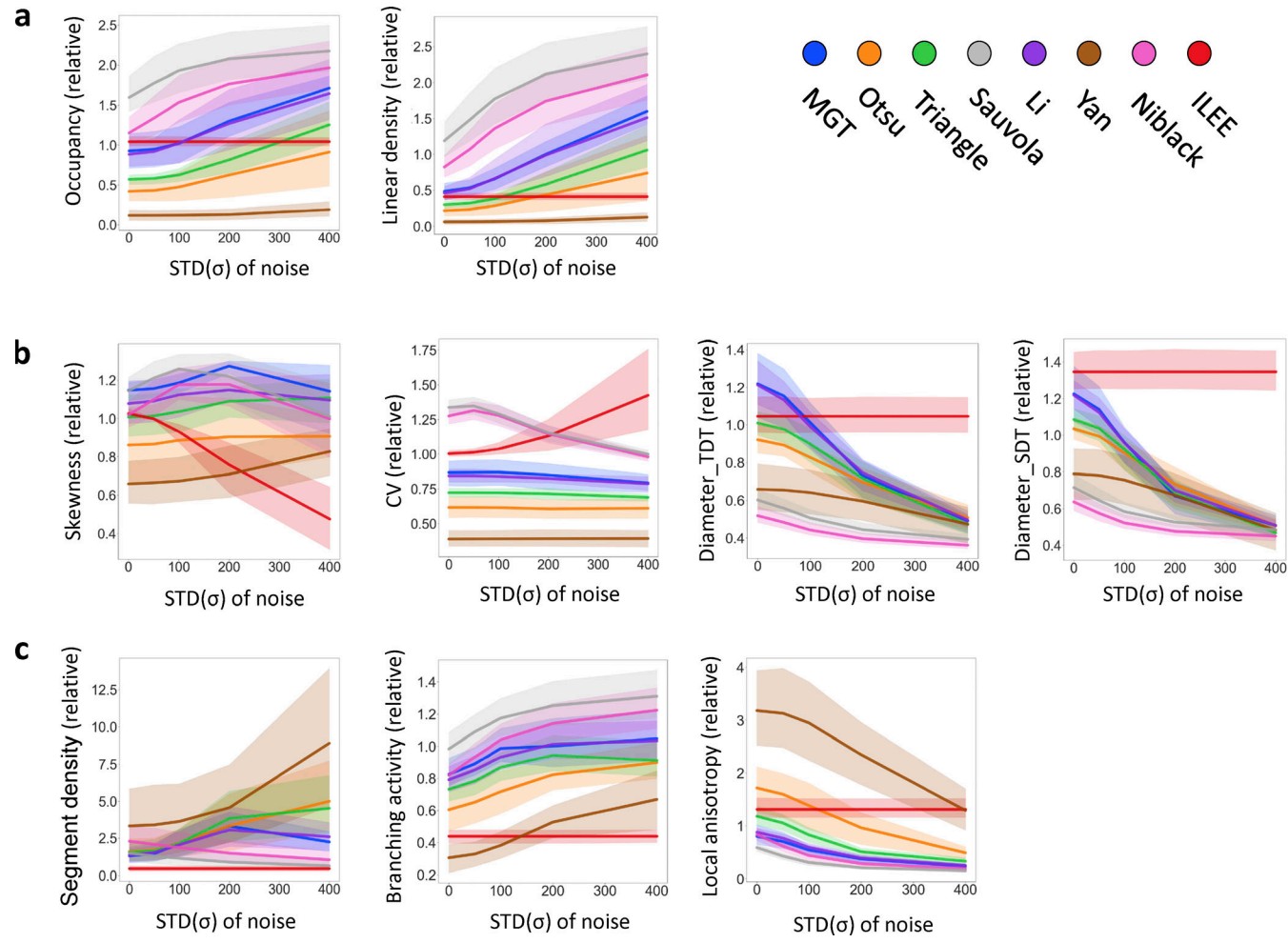

Figure S15. **The quantificational comparison of robustness of ILEE and other algorithms by index rendering stability.** Like Fig. S14, the dataset of Fig. 4 is added with a series of Gaussian noise ($\mu = 0$, $\sigma$ as variable), segmented by aforementioned algorithms, and their indices are computed and presented as the relative value to the ground truth. Each sample-noise combination is technically repeated by 12 times and the averaged result was used. The transparent area with light color indicates 95% confidence interval of each algorithm. For each figure, the lines with plainer slope indicate higher robustness (resistance to noise); being closer to 1.0 by value indicates higher accuracy. **(a–c)** Index class of density; (b), index class of bundling; (c), indices of other classes. For most of the indices, ILEE provide an extremely stable result against increasing noise, while other algorithms have very obvious change of value of output indices and are therefore no longer accurate (if they were), which echoes the visual observation of Fig. S14. Interestingly, *skewness* and *CV* are two exceptions, where ILEE shows more instability and tends to have a bifurcated direction of change.

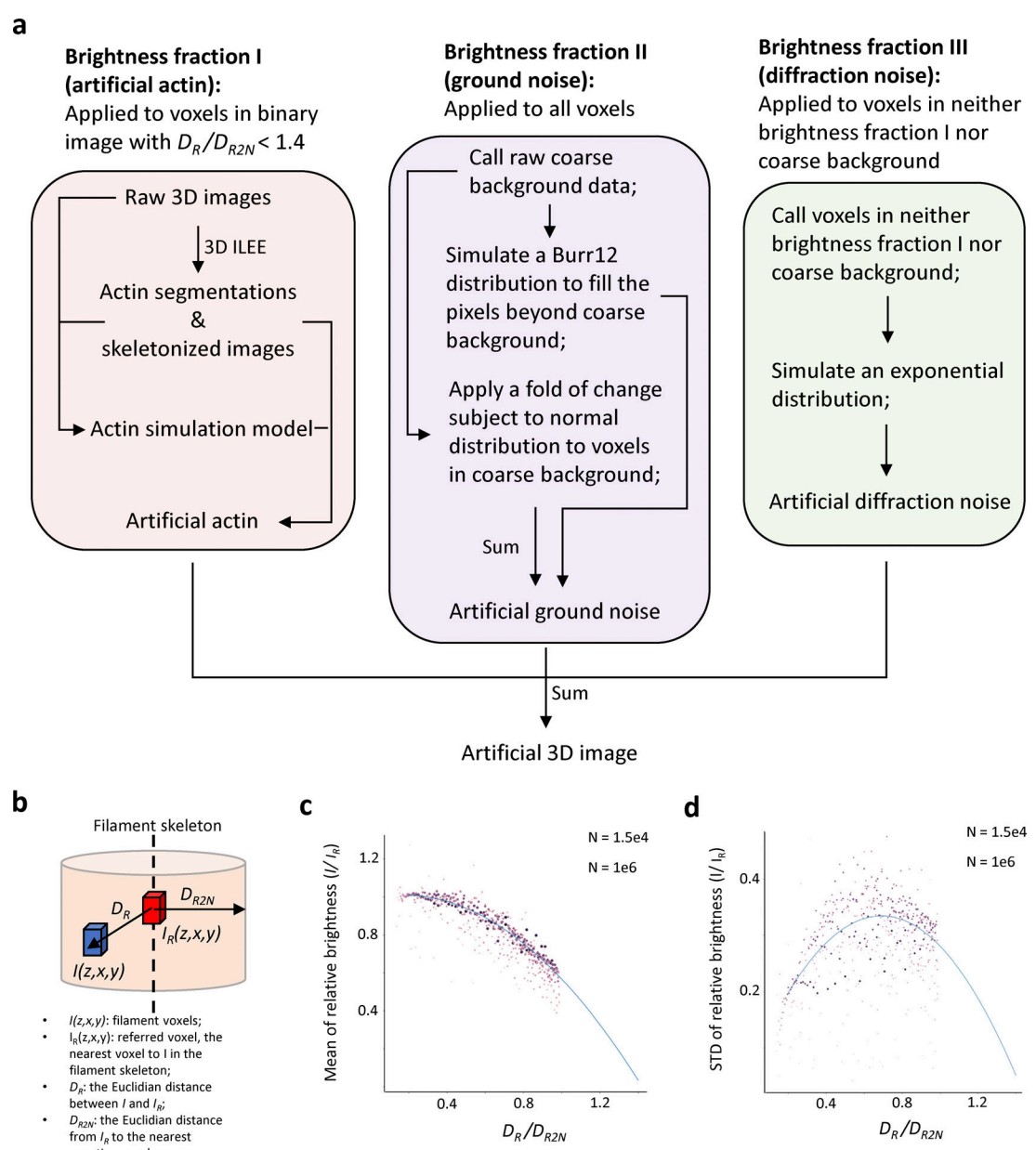

Figure S16. **An illustrated introduction of the actin image simulation model to generate 3D cytoskeleton images with ground truth. (a)** A schematic diagram describing the image simulation model. The training dataset is the same as Fig. 2. As preparation, raw images were processed by 3D ILEE. To mostly simulate the shape of actin, we directly adapted the skeletonized images ($I_{sk}$) generated from 3D ILEE as the central line of new filaments but filled all voxels with novel values. The artificial images comprise three brightness fractions: artificial actin, artificial ground noise, and artificial diffraction noise. First, referring to the binary images of the training samples, for each voxel ($I$) in the filament, we found its referred voxel ($I_R$), the nearest voxel to $I$ in the filament skeleton. **(b and c)** Next, we calculated the $D_R$ (the Euclidian distance between $I$ and $I_R$) and $D_{R2N}$ (the Euclidian distance from IR to the nearest negative voxel) of all voxels that belongs to the filament (also see b) and constructed a polynomial regression model (also see c) between the relative distance to $I_R$ (i.e., $D_R/D_{R2N}$) and the mean ($\mu_{atf}$) as well as the STD($\sigma_{atf}$) of relative brightness (i.e., $I/I_R$) for all on-filament voxels for all images. To generate the new artificial actin fraction, we refilled the on-filament voxel by the normal distribution $I/IR \sim N(\mu_{atf},\sigma_{atf}^2)$ based on the brightness of voxels belonging to the corresponding $I_{sk}$, and we abandoned the voxels with $D_R/D_{R2N} >1.4$ because they are more likely to be the structural noise—the cytosol and PM with fluorescence markers not binding to the actin. Second, for the ground noise fraction, we had different strategies to generate two groups of voxels. For the volume previously belonging to the coarse background, we directly apply a fold of change that is subject to a normal distribution to each voxel; for the other volume beyond coarse background, we gave random values that subject to a Burr-12 distribution fitted by voxels of the coarse background, thereby all voxels in the new artificial image were given with a ground noise value. Third, the volume previous belonging to neither coarse background nor real actin is considered as the space impacted by the diffraction (Fig. 2 a). Using the raw data of this region, we fitted an exponential distribution to describe the diffraction signal and refilled the diffraction region by this distribution. The choice of Burr-12 and exponential distribution is based on the best performance over a fitting test over 60 statistical models. Finally, we add the three fractions of brightness together and linearly align the value range to 0-4095 (12 bit). (b) a schematic diagram explaining the data utilized to fit the actin simulation model. **(d)** (c and d) the fitting result of the expected Mean of STD of the relative brightness of the actin simulation model. Only data with $D_R/D_{R2N}$ lower than 1 were used to fit the model because they were less likely to be influenced by the structural noise. As $D_R/D_{R2N}$ were discretely distributed, we added weights equal to $\log_{10}N$, where N means the number of voxels among the 31 samples, to each $D_R/D_{R2N}$.

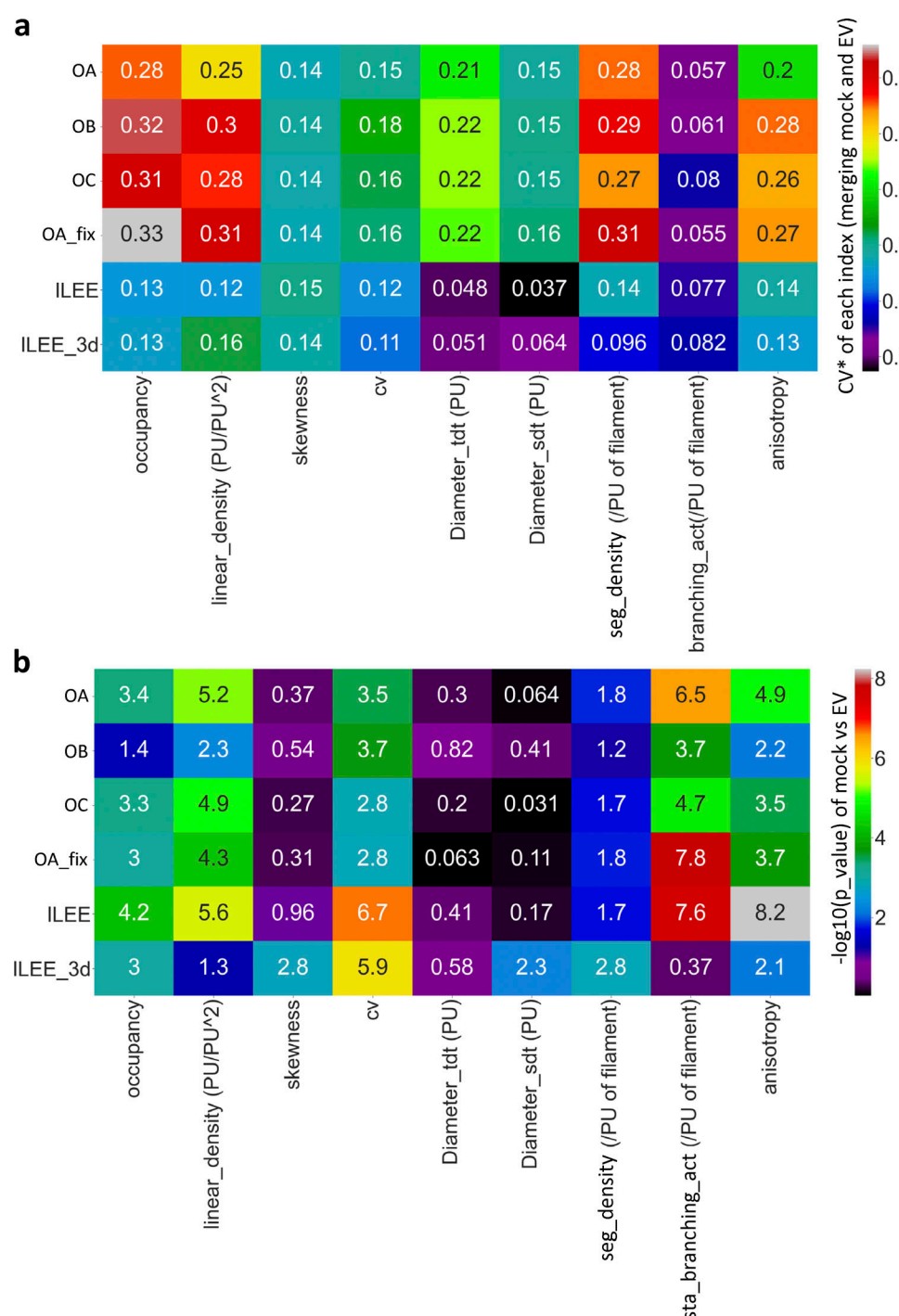

Figure S17. **The comparison of ILEE vs MGT over stability and capability to distinguish biological differences among real biological samples. (a)** The coefficient of variation (CV*) of each index rendered by ILEE and MGT by different operators. The individual CV*s of mock and EV group for each index-method combination are merged. ILEE or ILEE_3d got the lowest CV* (or highest stability) on *occupancy, linear density,* (fluorescence) *CV, Diameter_TDT/SDT, segment density,* and *local anisotropy.* **(b)** A comparison of the t-test P value of mock versus *P. syringae* (EV)-inoculated sample rendered by ILEE and MGT by different operators. ILEE or ILEE_3d has the highest -log$_{10}$ P value (i.e., lowest P value) on *occupancy, linear density, skewness,* (fluorescence) *CV, Diameter_SDT, segment density,* and *anisotropy.* Note that the CV* of computed indices in the figure and the fluorescence *CV* of the samples, which is a cytoskeletal index, are different concept.

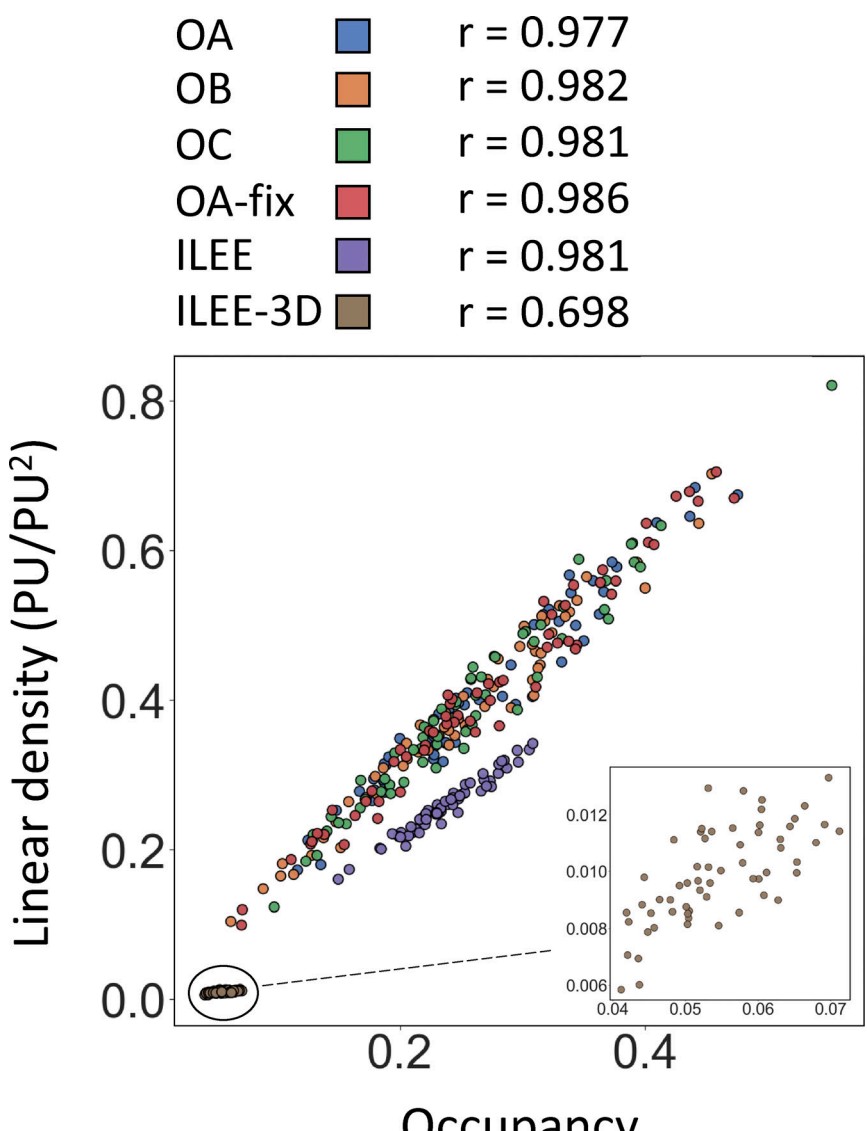

Figure S18.   **The correlation of *occupancy* and *linear density*.** The Pearson correlation coefficients (r) of *occupancy* vs. *linear density* rendered by ILEE and MGT (shown in Fig. 6) are calculated. The *occupancy* and *linear density* generally have a very high linear correlation, indicating that they agree with each other on evaluating the cytoskeletal density. ILEE and MGT does not display the same mathematical relationship, indicating that ILEE and MGT possess different tendencies in rendering the topological structure (mostly influencing linear density). In the 3D mode, they only have medium-strong correlation, potentially because (1) the 3D topological structure cannot be perfectly rendered due to concave/convex structures present in the skeletonized images and (2) 3D mode abandoned the oversampling process that 2D mode supports, due to insufficient computational power.

|  | Raw image | Binary image (segmented cytoskeleton) | Merge |
|---|---|---|---|

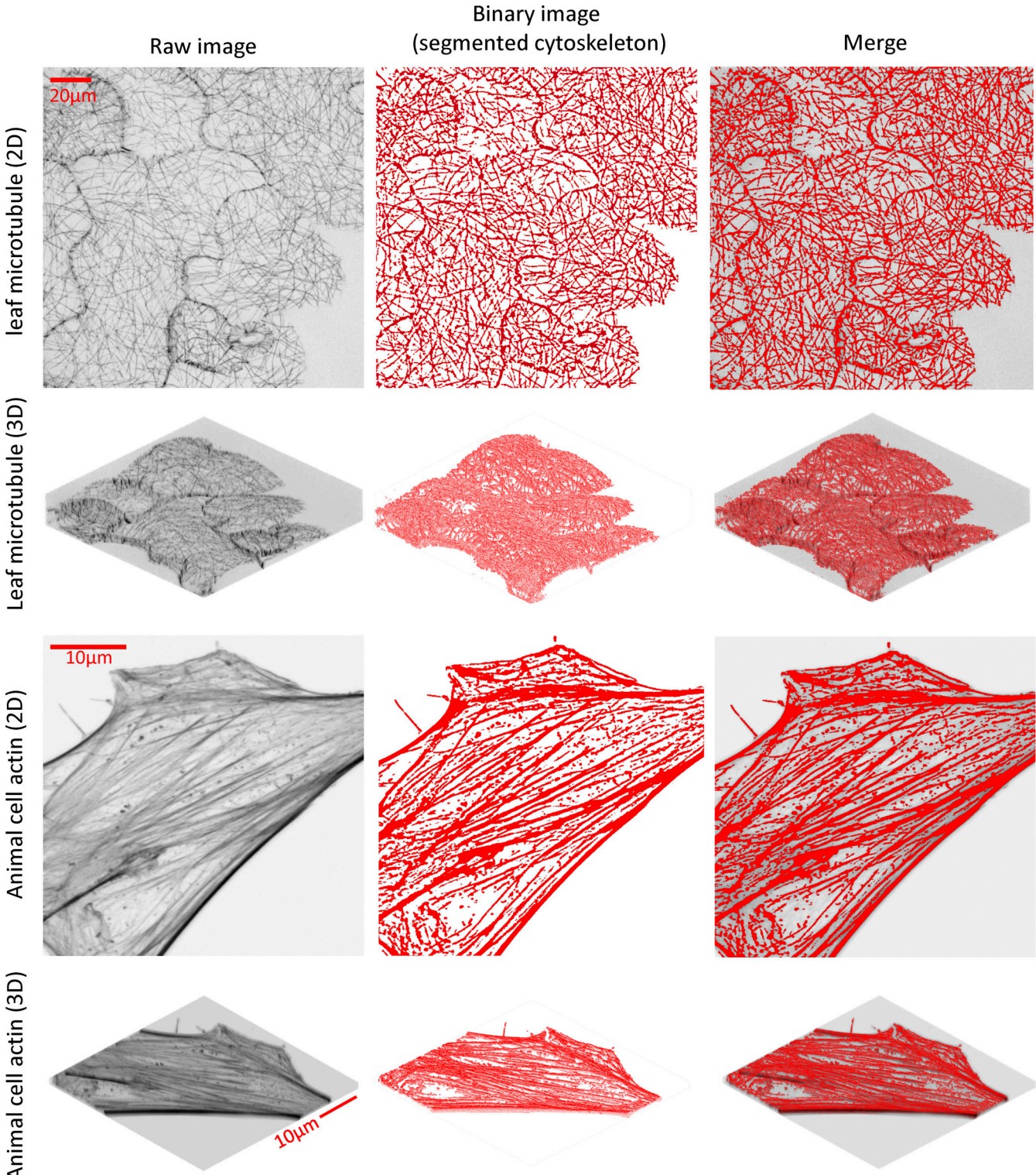

Figure S19.    **The performance of ILEE on other type of biological sample.** We tested ILEE on leaf microtubule images of Arabidopsis Col-0/GFP-MAP4 and animal cell actin images of YUMMER1.7D4 line. The Col-0/GFP-MAP4 images were generously provided by Dr. Silke Robatzek (Ludwig-Maximilians-Universität München) from a previously published dataset (Faulkner et al., 2017). The result indicates ILEE accurately covers the cytoskeleton samples with locally dynamic brightness, thickness, and shape, with no visible difference compared with its performance on leaf actin images.

**Raw Data**

**ILEE vs ground truth comparison of binary image**

**ILEE vs ground truth comparison of skeleton image**

Match

False-positive

False-negative

Figure S20. **ILEE and human eye have different tendency to judge the topological structure of cytoskeleton, especially between two bright bundles.** A cropped portion of the demonstration image from Fig. 3 a and its derived images are generated and demonstrated to explain the discrepancy related to the determination of the topological structure between ILEE and hand-drawn binary ground truth (i.e., human eye evaluation). When ILEE binary image is compared with ground truth, they mostly match with each other with only minor and non-influential differences. When both binary images were skeletonized based on their topological structure, a numerous branches were detected from the ground truth, but not from ILEE. This phenomenon is very obvious in the highlighted (i.e., circled) area, where two bright actin bundles are very close to each other. This is intriguing, as where the human eye identified many branches and intervals, ILEE did not. These structures have a strong impact on the topological structure. Therefore, the topology-sensitive indices, *linear density*, *segment density*, and *branching activity* display contrast and stable inconformity between ground truth and ILEE (Fig. 4 d). However, since the hand-drawn ground truth image is not a rigorously defined ground truth, we surmise that inconformity cannot be interpreted to any inaccuracy of ILEE. At the same time, the skeletonization algorithm may be replaced/upgraded as an entire module, for further improved performance.

