## [Peer Review File · The Journal of Cell Biology]

ILEE: Algorithms and Toolbox for Unguided and Accurate Quantitative Analysis of Cytoskeletal Images

Pai Li, Ze Zhang, Yiyong Tong, Bardees Foda, and Brad Day

Corresponding Author(s): Pai Li, Michigan State University

Review Timeline:

Submission Date:	2022-03-06
Editorial Decision:	2022-05-01
Revision Received:	2022-08-04
Editorial Decision:	2022-08-30
Revision Received:	2022-11-01
Editorial Decision:	2022-11-02
Revision Received:	2022-11-10

Monitoring Editor: Alex Mogilner

Scientific Editor: Dan Simon

Transaction Report:

DOI: <https://doi.org/10.1083/jcb.202203024>

May 1, 2022

Re: JCB manuscript #202203024

Dr. Pai Li
Michigan State University
Department of Plant, Soil and Microbial Sciences
1066 Bogue St.
East Lansing, Michigan 48824

Dear Dr. Li,

Thank you for submitting your manuscript "ILEE: Algorithms and Toolbox for Unguided and Accurate Quantitative Analysis of Cytoskeletal Images." Your manuscript was examined by two reviewers, whose comments are appended below. Although the reviewers express potential interest in this work, significant concerns unfortunately preclude publication of the current version of the manuscript in JCB.

You will see that while Reviewer #1 is enthusiastic about your study, Reviewer #2 raises significant concerns that would absolutely have to be addressed. Specifically, the reviewer #2 asks for more detailed information about the segmentation methodology and for a thorough comparison with other relevant algorithms, which is a requirement of the JCB Tools format.

Please let us know if you are able to address the major issues outlined above and wish to submit a revised manuscript to JCB. Note that a substantial amount of additional data would likely be needed to satisfactorily address the concerns of the reviewers.

The typical timeframe for revisions is three to four months. While most universities and institutes have reopened labs and allowed researchers to begin working at nearly pre-pandemic levels, we at JCB realize that the lingering effects of the COVID-19 pandemic may still be impacting some aspects of your work, including the acquisition of equipment and reagents. Therefore, if you anticipate any difficulties in meeting this aforementioned revision time limit, please contact us and we can work with you to find an appropriate time frame for resubmission. Please note that papers are generally considered through only one revision cycle, so any revised manuscript will likely be either accepted or rejected.

If you choose to revise and resubmit your manuscript, please also attend to the following editorial points. Please direct any editorial questions to the journal office.

GENERAL GUIDELINES:

Text limits: Character count is < 40,000, not including spaces. Count includes title page, abstract, introduction, results, discussion, and acknowledgments. Count does not include materials and methods, figure legends, references, tables, or supplemental legends.

Figures: Your manuscript may have up to 10 main text figures. To avoid delays in production, figures must be prepared according to the policies outlined in our Instructions to Authors, under Data Presentation, <https://jcb.rupress.org/site/misc/ifora.xhtml>. All figures in accepted manuscripts will be screened prior to publication.

Supplemental information: There are strict limits on the allowable amount of supplemental data. Your manuscript may have up to 5 supplemental figures. You currently exceed this limit. We will likely be able to give you more space if necessary but please consolidate as much as possible. Up to 10 supplemental videos or flash animations are allowed. A summary of all supplemental material should appear at the end of the Materials and methods section.

Please note that JCB now requires authors to submit Source Data used to generate figures containing gels and Western blots with all revised manuscripts. This Source Data consists of fully uncropped and unprocessed images for each gel/blot displayed in the main and supplemental figures. If your paper will contain cropped gel and/or blot images, please be sure to provide one Source Data file for each figure that contains gels and/or blots along with your revised manuscript files. File names for Source Data figures should be alphanumeric without any spaces or special characters (i.e., SourceDataF#, where F# refers to the associated main figure number or SourceDataFS# for those associated with Supplementary figures). The lanes of the gels/blots should be labeled as they are in the associated figure, the place where cropping was applied should be marked (with a box), and molecular weight/size standards should be labeled wherever possible. Source Data files will be made available to reviewers during evaluation of revised manuscripts and, if your paper is eventually published in JCB, the files will be directly linked to

specific figures in the published article.

If you choose to resubmit, please include a cover letter addressing the reviewers' comments point by point. Please also highlight all changes in the text of the manuscript.

Regardless of how you choose to proceed, we hope that the comments below will prove constructive as your work progresses. We would be happy to discuss them further once you've had a chance to consider the points raised. You can contact the journal office with any questions, cellbio@rockefeller.edu or call (212) 327-8588.

Thank you for thinking of JCB as an appropriate place to publish your work.

Sincerely,

Alex Mogilner, PhD
Monitoring Editor
Journal of Cell Biology

Dan Simon, PhD
Scientific Editor
Journal of Cell Biology

Reviewer #1 (Comments to the Authors (Required)):

In this manuscript, the authors presented a methodology for automated extraction and analysis of filamentous structures in 2D and 3D fluorescent imaging data. The authors showed that their approach outperforms a number of existing methods in terms of accuracy, stability, and robustness. They introduced novel quantitative metrics that enrich characterization of diverse and complex filamentous patterns. They also illustrated the approach using both simulated and experimental data. Finally, the authors provided their methodology as a toolbox with Google Colab interface to be accessible to the broader research community.

Overall, I find this work very important to the field of biological and biomedical image analysis. The manuscript is well-written with convincing validations and sufficient details for reproducibly and further developments. It is a high-quality work with a high-quality presentation. So, I believe this paper will have a significant impact and I enthusiastically support its publication at JCB, which is a perfect platform for this kind of work.

I do not have any major concerns, but just minor suggestions that, in my opinion, could improve clarity:

1. Line 54. "two types of cytoskeleton, actin and microtubule". I agree that actin and microtubules are the major/most-studied components, but I believe more accurate (recent) view is that filamentous components of the cytoskeleton include actin, microtubules, intermediate, and septin filaments.
2. Lines 152-154. "However, due to limited computational biological resources, most studies have exclusively employed the z-axis-projected 2D image..." I don't think limited computational resources is the major/only reason for using maximum projection. It is often used to boost the signal-to-noise ratio, in situations when the fluorescent signal has to be set weak to avoid/minimize perturbations to the natural cell state.
3. Line 180. Typo in "the true fluorescence, that which is emitted by"
4. Lines 196-197 "...will be determined using a linear model trained by the representative value rendered by NNES and manual global thresholding (MGT)..." This part is a bit confusing. The whole method is presented as "fully automated" (line 124), but here it sounds like manual global thresholding is still a part of the processing. Please clarify.
5. Lines 261-262 "To overcome this dilemma, we applied a strategy using a lower K1 and a higher K2 to compute two different binary images that focus on thin/faint components and thick components, respectively." Does that still preserve the detection of filaments with intermediate thickness?
6. Line 275 "...used Niblack thresholding..." A reference is needed here.

7. Line 276 "... we calculated the mean of the top 5% of the Euclidian distance transformation". Is 5% an arbitrary choice?
8. Line 304. Typo "...cytoskeleton. static severing activity assumes..." Should be capital 'S' in the beginning of the sentence.
9. Lines 449-450. "In total, nine indices that cover features of density, bundling, severing, branching, and directional order are measured and compared." Nine out of twelve? Why not the other three?
10. Line 463. Typo "As suggested by Supplemental Fig. 14b, we that ILEE indeed has..."
11. Lines 477-478 "... and diameter-SDT (as a representative of direct indices) have a striking zero correlation". Exactly zero? Sounds suspicious to have a perfectly zero correlation of two real experimental signals.
12. Lines 559-560 "human eyes tend to "hallucinate" imaginary filaments that do not statistically exist." I find it strange to blame researcher's hallucination for discrepancies with a statistical analysis. I would suggest rephrasing this statement.
13. Line 677 "parameter $k = -0.36$ " The way it is written, it is not clear if this parameter is derived or chosen.
14. Lines 681-682 "Finally, the mean of all individual K_2 will be output as the recommend K_2 for the total batch". How big is the spread and does it matter?
15. Line 688. As I understand, the coefficients 0.293 and 0.707 are used to interpolate the value on the diagonal 1 pixel length from point (x,y). If yes, may be worth mentioning in the text. If not, may be worth explaining.
16. Line 697. "is rendered through Scharr operator". A reference is needed here.
17. Line 703. It is not clear to me, where the numbers 1.163, 101.68, and 7.33 come from? I think, having a fully automated processing, it is important to comment on any chosen parameters because such choices may be a source of inconsistencies in the analysis of "new" data (not used for testing in this particular study).
18. Lines 709-710. The same concern as in 17.
19. Line 737. Typo "hence it can be further be represented as"
20. Line 785-787. A well-known issue with that type of skeletonization is appearance of many spur branches sensitive to boundary fluctuations. Many algorithms rely on pruning to reduce errors associated with such artifactual branches. Did you consider this issue? Based on what I see in the description, I suspect this may be a significant source of inaccuracy. Please comment of that.
21. Line 795. Typo. A duplicated incomplete formula on the second line.
22. Line 824. "The total number of type-3 and type-4 branches is collected" Although the reference 22 is provided, for the sake of self-sufficiency, I would suggest defining these two types of branches.
23. Section "Simulation model of 3D artificial actin image". I appreciate all the details, but this section tells what was used and doesn't say why or how it works. What is the logic behind all these different distance metrics? How do they define the noise? For a general reader, who may not have experience simulating different types of imaging noise, I would add some rationale for this part of the algorithm.

Reviewer #2 (Comments to the Authors (Required)):

In this work, Li et al. describe a pipeline to segment cytoskeletal filaments in fluorescence images, and then to quantitatively characterize their organization using various "cytoskeleton indices". The segmentation approach is based on global and local thresholding, and employs intensity as well as intensity derivative information. The segmented binary image is skeletonized in order to calculate the cytoskeleton indices. The pipeline is applicable in both 2D and 3D, which is in principle a useful feature.

I do have major concerns about this work. The manuscript as written is difficult to follow. The analysis pipeline seems quite complicated with many steps that are often not justified well other than that they "work." There are many seemingly arbitrary procedures. The comparison to other algorithms is incomplete and does not demonstrate strong superiority of ILEE. The work does not mention or cite many of the more recent advances in the field of segmenting curvilinear structures in images.

Manuscript itself:

There is quite some jargon. Terms, including acronyms, are often introduced without proper definition, making it difficult to follow what exactly is done. Various analysis steps are described without proper rationale. The results section becomes (somewhat) easier to follow after reading the detailed methods, but this is not the order in which the manuscript will be read.

Many sentences are difficult to understand. For example: Lines 467-468: The authors write "Next, we attempted to understand whether different indices of the same class, particularly density and bundling, can reflect the quantitative level of the class in accordance, or instead show inconsistency." I think I get what the authors are trying to say, but please have someone read the manuscript and give you feedback on clarity!!!

Segmentation pipeline:

As mentioned above, there are many steps in their pipeline without strong rationale and sometimes operating as a black box. Examples:

In the method for determining the "course background," I cannot find any reference for the "non-connected negative element scanning" (NNES) algorithm used to estimate it. How the authors then go from the "peak" of the NNES curve to the presumed optimal threshold (Fig. 2) feels like a black box. The "straight-line fit" employed in Suppl. Fig. 2 is not convincing. In this section, the authors also mention a "peak-of-frequency brightness" as an alternative method (line 202). What is this?

In the procedure to determine g_thres , the line and curve fits in Suppl. Fig. 7 are not convincing.

The authors correctly state that segmenting both thin and thick filaments at the same time is challenging. Thus, this is an important problem that the authors are working on. However, why do they only employ 2 values of K , a small one for "thin" filaments and a larger one for "thick" filaments? In reality, the filaments most likely have a continuum of thickness, and I am not sure why just 2 values of K will be sufficient to cover the full thickness range. Also, how the authors estimate $K2$, described on lines 272-279, is rather heuristic.

Cytoskeleton indices:

After segmentation, to describe the segmented cytoskeleton, the authors calculate 12 indices, but then only focus on 9 because "they are normalized and ready for biological interpretation" (lines 289-290). The other 3 should either be made biologically interpretable or just not get mentioned. It is also not clear to me why some of the indices require image oversampling. And if they indeed require oversampling, then why is oversampling done in 2D but not in 3D?

The difference between $diameter_TDT$ and $diameter_SDT$ is not clear.

I am not convinced by the "static severing activity" index. The authors state that "a severing event generates two visible cytoskeleton filaments." This is of course true, but how do they assess this in static images? If there are two filaments aligned head-to-tail, how do you know in a static image whether these two filaments were generated by severing vs. by two nucleation events?

Assessment and comparisons:

As far as I can tell, none of the other algorithms that ILEE is compared to in Fig. 4 employ derivative information to identify curvilinear structures in an image. They seem to be different ways of thresholding the raw intensity. This is far from the state of the art in this field. The authors do not introduce or discuss key concepts in curvilinear object detection, such as steerable edge and ridge filters based on first and second order derivatives, as well as nonmaximum suppression, which is a more accurate procedure to determine the detected line centers than skeletonization (as is done in this work). See e.g. Canny IEEE PAMI 1986, Jacob & Unser IEEE PAMI 2004, Puspoki et al. IEEE TIP 2016, Gan et al. Cell Systems 2016, Kittisopikul et al. Bioinformatics 2020. These concepts should be at least discussed, and even better the performance of ILEE should be compared to pipelines based on some of these more advanced concepts.

The authors also do not state anywhere whether they implemented the algorithms to which ILEE is compared, or whether they employed software written by others.

In evaluating the segmentation, the use of the terms "pseudo-positive" and "pseudo-negative" is rather unusual. Why not use the common terms "false positive" and "false negative"? Also, the "actin-like pseudo-positive" category is not justified - if the ground truth is not good enough, then it cannot serve as the ground truth. More problematically, the procedure for classifying segmentations as "actin-like pseudo-positive" is not explained, and in fact, in looking at the segmentations of Fig. 4, those segmentations do not look very actin-like. The formulae for calculating the different fractions are not explicitly defined.

In terms of ILEE's performance, specifically regarding the calculated cytoskeletal indices (Fig. 4d), ILEE's improved performance

is heterogeneous. It systematically underestimates three of the 9 indices. Its error bars are comparable to those of MGT and Li, except for the severing activity index where MGT and Li have higher error bars, but in fact at the same time they have less systematic error. So it is not clear which method is really better.

More minor comments:

In the introduction, the authors state that "shared by plants and animals are two types of cytoskeleton, actin and microtubule." What about intermediate filaments?

Lines 266-268: The authors state "Since the theoretical minimum thickness of a distinguishable cytoskeletal component is approximately equal to one pixel/voxel unit". Well, this depends on the pixel size vs. the microscope resolution. With proper spatial sampling, the minimum thickness hopefully covers about 3 pixels.

Line 355: The authors state that "the adaptive global thresholding approaches (from Otsu to Yan) tend to be relatively accurate when judging the thick and bright bundles." I disagree with the authors' judgment regarding the performance of the Yan approach (Fig. 4a). As their quantitative analysis shows, it has many false negatives.

Fig. 5c: what are the p's shown on each graph?

Fig. 6c-h, l-n: What are the little a's and b's etc. inside the bar graphs?

The text discussing Fig. 6 does not cite the many panels of Fig. 6 as needed.

Suppl. Fig. 14 is not cited anywhere, and all the Suppl. Figures after it are cited incorrectly (i.e. what the text refers to as Suppl. Fig. 14 is actually Suppl. Fig. 15, etc.).

Response to Reviewers

We greatly appreciate the highly professional, critical comments and constructive advice by the reviewers and the editorial board. Your efforts significantly enable us to considerably improve our manuscript and contribute to the community with a better ILEE toolbox. Herein, we would like to provide a point-by-point response to the aforementioned issues; the corresponding edits have been applied to the revised manuscript.

The original reviewer's comments are shown in **black**, and our response are shown in **green**.

Reviewer #1 (Comments to the Authors (Required)):

In this manuscript, the authors presented a methodology for automated extraction and analysis of filamentous structures in 2D and 3D fluorescent imaging data. The authors showed that their approach outperforms a number of existing methods in terms of accuracy, stability, and robustness. They introduced novel quantitative metrics that enrich characterization of diverse and complex filamentous patterns. They also illustrated the approach using both simulated and experimental data. Finally, the authors provided their methodology as a toolbox with Google Colab interface to be accessible to the broader research community.

Overall, I find this work very important to the field of biological and biomedical image analysis. The manuscript is well-written with convincing validations and sufficient details for reproducibly and further developments. It is a high-quality work with a high-quality presentation. So, I believe this paper will have a significant impact and I enthusiastically support its publication at JCB, which is a perfect platform for this kind of work.

I do not have any major concerns, but just minor suggestions that, in my opinion, could improve clarity:

1. Line 54. "two types of cytoskeleton, actin and microtubule". I agree that actin and microtubules are the major/most-studied components, but I believe more accurate (recent) view is that filamentous components of the cytoskeleton include actin, microtubules, intermediate, and septin filaments.

Re #1:

Agreed. We have already corrected this information and reorganized the description related to this issue. See line 54-60 of the revised manuscript.

2. Lines 152-154. "However, due to limited computational biological resources, most studies have exclusively employed the z-axis-projected 2D image..." I don't think limited computational resources is the major/only reason for using maximum projection. It is often used to boost the signal-to-noise ratio, in situations when the fluorescent signal has to be set weak to avoid/minimize perturbations to the natural cell state.

Re #2:

We actually agree that enhancing signal-to-noise ratio is a major reason z-axis projection; however, we meant to include this by "computational biological resources", but our language was too vague. We have added more specific descriptions there for better delivery. See line 169-170 of the revised manuscript.

3. Line 180. Typo in "the true fluorescence, that which is emitted by"

Re #3:

Thank you and it is now corrected.

4. Lines 196-197 "...will be determined using a linear model trained by the representative value rendered by NNES and manual global thresholding (MGT)..." This part is a bit confusing. The whole method is presented as

"fully automated" (line 124), but here it sounds like manual global thresholding is still a part of the processing. Please clarify.

Re #4:

Sorry for the confusion. The total developed ILEE pipeline is indeed full-automated as we claim. However, during our development, we do need to train a few models using MGT (the only possible and trusted way as ground truth input) to make sure that our automated pipeline indeed uses parameters/calculation models that are both accurate and compatible to varying images.

The model here is used to transfer the global threshold at peak of NNE as a feature value (see Fig. 2b (i), Supplemental Fig. 1c, Supplemental Fig. 2, and *Method* line 740-744 of the revised manuscript,) to the global threshold for coarse background (t_{cbg}). However, we indeed found that our previously submitted version was very perplexing here. For better delivery, we had a major edition of the total subsection of *Identification of coarse background*, which generally described our rationale with clearer logic. The theoretical reason for the wide adaptability is our original approach actually uses topological/statistical features of ground noise, regardless of how high it is overall, so it is supposed to handle all realistic samples, even non-existent samples. Our tests also found the model can obtain satisfactory t_{cbg} for extreme samples. Therefore, as we have confirmed and built into the library, the "global threshold" is exactly automatically determined and there is zero input by users. Also, this library is modularized and can be easily upgraded if unpredictable issues were met and reported by users.

5. Lines 261-262 "To overcome this dilemma, we applied a strategy using a lower K1 and a higher K2 to compute two different binary images that focus on thin/faint components and thick components, respectively." Does that still preserve the detection of filaments with intermediate thickness?

Re #5:

This is a very good question. The answer is yes; it does handle intermediate thickness well for all "general" images. This is because the thickness (by pixel) of cytoskeleton images is restricted in a certain range: the magnification level should neither too low to recognize/measure thin filaments, nor too high to maintain effective optical resolution or keep filament stagnant when doing live imaging. Therefore, K1 and K2 cannot be unlimitedly too low/high. During our experiments, we do find that intermediate filaments are well covered by both K1 and K2. For these theoretical and practical reasons, we believe 2 Ks are appropriate for the algorithm. However, the adding Ks is always possible in library updates; if we did receive valid evidence suggesting intermediate filaments are not covered in extreme cases, we could add new functions with more Ks to the library for users meeting such unusual problem.

6. Line 275 "...used Niblack thresholding..." A reference is needed here.

Re #6:

Thanks for the reminder and now it is already added to the new version. We also added other algorithm API references that we previously omitted. See line 305 of the revised version.

7. Line 276 "... we calculated the mean of the top 5% of the Euclidian distance transformation". Is 5% an arbitrary choice?

Re #7:

We tested several different choices of top percentile and found 5% is quite representative to thick filaments and reliable. On the other hand, this is not a very important parameter, so we did not test it at very high resolution (such as 4.9 vs 5.1), and hence we did not describe this in detail.

However, this is a good reminder that we should open this percentage as an argument with default 5%, for advanced users in case of unusual scenario. We will add this to our next release.

8. Line 304. Typo "...cytoskeleton. static severing activity assumes... " Should be capital 'S' in the beginning of the sentence.

Re #8:

Thanks for pointing out. This is no longer this issue because we have totally rewritten this subsection to improve the other previous flaws. See line 355-370.

9. Lines 449-450. "In total, nine indices that cover features of density, bundling, severing, branching, and directional order are measured and compared." Nine out of twelve? Why not the other three?

Re #9:

Thank you for the reminder. They are non-normalized intermediate indices for advanced users. We have added the description in line 318-321 of new version; they were also mentioned in the Fig. 1 legend.

10. Line 463. Typo "As suggested by Supplemental Fig. 14b, we that ILEE indeed has..."

Re #10:

Thank you and already corrected.

11. Lines 477-478 "... and diameter-SDT (as a representative of direct indices) have a striking zero correlation". Exactly zero? Sounds suspicious to have a perfectly zero correlation of two real experimental signals.

Re #11:

It is true that this is not absolutely rigorous language. They are -0.11 and 0.06, with very high p-value, which basically means there is barely any correlation, if we need to be too rigorous to call it "zero". However, we choose to call it "striking zero" because this is too much out of our expectation and it means one type of bundling indices is not actually trust-worthy. Although we are subjectively on the side to support direct indices according to our theory and experience, currently there is no experimental evidence as it requires very advanced confocal systems and approaches which are not available in our lab. We wish use "striking zero" to address attention to readers to realize this issue, so that the problem is more likely to be settled in the future.

12. Lines 559-560 "human eyes tend to "hallucinate" imaginary filaments that do not statistically 560 exist." I find it strange to blame researcher's hallucination for discrepancies with a statistical analysis. I would suggest rephrasing this statement.

Re #12:

Thank you and already rephrased. See line 597-598.

13. Line 677 "parameter $k = -0.36$ " The way it is written, it is not clear if this parameter is derived or chosen.

Re #13:

Sorry for this omission. They are optimized selection of parameter by comparing with thick filaments of ground truth, which we previously failed to mention because we thought it was not very important. We have added this information in the version. See line 715-716.

14. Lines 681-682 "Finally, the mean of all individual K2 will be output as the recommend K2 for the total batch". How big is the spread and does it matter?

Re #14:

Good question. We tested the spread (data distribution) of the image sample of Fig. 5 as an example to answer your question. The result is shown as **Reviewer Responding Material 1** in the end of this file, following the responding letter. We did not observe wide spreading as the data concentrated surrounding the mean.

A somehow sub-optimal K2 may have minor impact to the ILEE result. However, it is unlikely to change the result when comparing different biological groups. What matter more is to make sure K2 is kept for all samples in all biological groups; this approach mainly serves to find a fixed representative value of K2.

15. Line 688. As I understand, the coefficients 0.293 and 0.707 are used to interpolate the value on the diagonal 1 pixel length from point (x,y). If yes, may be worth mentioning in the text. If not, may be worth explaining.

Re #15:

Sorry for your confusion, and you are right. This is exactly to interpolate 1 pixel length for diagonal. This information is now added into both main text line 729-730 and Supplemental Fig. 4.

16. Line 697. "is rendered through Scharr operator". A reference is needed here.

Re #16:

Thanks for pointing out. It is added now.

17. Line 703. It is not clear to me, where the numbers 1.163, 101.68, and 7.33 come from? I think, having a fully automated processing, it is important to comment on any chosen parameters because such choices may be a source of inconsistencies in the analysis of "new" data (not used for testing in this particular study).

18. Lines 709-710. The same concern as in 17.

Re #17, to Reviewer #1 issue 17 and 18 together:

We totally understand this reasonable and rigorous concern, but we believe such strategy will not actually impact its critical feature of automated process. Here are the reasons:

1. As you have already mentioned (and I have slightly discussed in the respond) in the *Reviewer #1 issue 4*, we have trained some model to provide one-for-all settings to calculate intermediate variables, including these shown in Supplemental Figure 1 and 2. Since the users does not need to input any parameter to automatically get comparable results among biological groups or even across different experiments settings, we believe this level of automation has already solved the basic problem in this area.
2. The reason we believe our model works at satisfactory performance for diverse image is that the cytoskeleton image per se have a certain range of diversity by its nature, which we also mentioned in *Reviewer #1 issue 5*. In brief, for thickness, the diameter by pixel is limited by effective optical resolution; for brightness value, we always linearly normalized image to 12-bit in the pipeline (to be updated to 0-1 float in future library upgrades), so it is also not a problem. Therefore, it is indeed feasible to train a universal model by diverse training datasets, as we did and found good performance. However, if the image is not actual cytoskeleton and lost these features, we may not promise that model would work at satisfactory performance, which is indeed the limit of these models.
3. Actually, we have already tried some "new" data, which is even not the same type of cytoskeleton (plant microtubule and animal cell actin) collected by different labs using exact the same pipeline, shown as Supplemental Fig. 17. While we did not make the ground truth image, we found through our visual inspection that these model work out very well to identify filaments that are different from training samples.

However, it is indeed true that we cannot guarantee that this and other models can work for all cytoskeleton samples without exception, and we are willing to fix them if observed. However, to further optimize/upgrade them, we first need image samples that fails to obtain good segmentations and then inspect the problem.

Therefore, we are happy to get feedbacks from users and collect problematic image samples to learn about the limitation of any models to solve the problem.

Finally, we want to mention that an ultimate automatic 2D/3D pipeline even without any pre-trained parameter calculation model is actually a holy grail for biological fiber segmentation algorithm. A relatively feasible strategy is deep learning, but this is still not quite available because it requires not only a complicated model but also a huge dataset with ground truth. However, there is no massive ground truth available due to unrealistic labor cost – it takes 10 hours for an expert to finish only one 2D image. Therefore, from very long-term perspective, we regard our ILEE library as an automated approach to gain trustworthy ground truths, to open the gate to finally reach the ideal algorithm that you described.

19. Line 737. Typo "hence it can be further be represented as"

Re #18:

Thank you and now it is already corrected.

20. Line 785-787. A well-known issue with that type of skeletonization is appearance of many spur branches sensitive to boundary fluctuations. Many algorithms rely on pruning to reduce errors associated with such artifactual branches. Did you consider this issue? Based on what I see in the description, I suspect this may be a significant source of inaccuracy. Please comment of that.

Re #19:

This is actually a very good consideration. During our development of ILEE, we also found this is a potential performance booster and did try some pruning algorithms. However, we found their performance is very difficult to control and evaluation. This is majorly because it is very difficult to determine the “skeletonization ground truth” of the image, as we are using a very challenging type of image sample – plant live actin. In this case, manually draw the skeleton can be very subjective (more subjective than the binary ground truth) – especially when handling a very short branch (we are not confident to determine whether they are true branches or “fake spur branches sensitive to boundary fluctuations”). Therefore, also we can adjust/training pruning parameters (another pre-set model you may not like) for a seeming neater skeletonization but cannot make sure this is closer to the actual skeleton. Also, we found that the state-of-art pruning algorithms are likely to be based on MATLAB (not open source for all users) and black-boxed C++ program. To develop a high-performance algorithm to automatically handle the pruning problem can be efforts even for another paper.

However, we do provide solution to this issue by oversampling (interpolation). During our test, we found through our visual inspection that interpolating original image to higher resolution can reduce the “boundary fluctuation artifact” while still detecting small branches. Therefore, we added this to the 2D pipeline when calculating topological-sensitive indices; it still has more potential because greater interpolation rate is available. Unfortunately, this is for 2D pipeline exclusively because 3D would take too much memory and is not available for general PCs.

21. Line 795. Typo. A duplicated incomplete formula on the second line.

Re #20:

Thank you and already corrected.

22. Line 824. "The total number of type-3 and type-4 branches is collected" Although the reference 22 is provided, for the sake of self-sufficiency, I would suggest defining these two types of branches.

Re #21:

Agreed. This is now added. See line 876-878.

23. Section "Simulation model of 3D artificial actin image". I appreciate all the details, but this section tells what was used and doesn't say why or how it works. What is the logic behind all these different distance metrics? How do they define the noise? For a general reader, who may not have experience simulating different types of imaging noise, I would add some rationale for this part of the algorithm.

Re #22:

Sorry for your confusion. First of all, I apologize that we had one critical omission – we missed to refer this total subsection in *Methods* to Supplemental Fig. 14, where we have an illustrated description of the rationale of this simulation model, and a relatively detailed literal instruction of each step was provided in the figure legend, in addition to the mathematical description in the *Methods*. If my understanding is correct, this is actually the problem. A brief answer to the questions: this model is based on our framework that the total image is dissected into 3 parts: filament true fluorescence, diffraction noise, and ground noise (Fig. 2a). During our development, we found the three portions have some high-confidence statistical features respectively, which allowed us to fit statistical models (e.g., normal distributions, exponential distribution, Burr12 distribution) to re-render three parts individually and added them up together. Therefore, we can directly get a new image using known binary ground truth. A copy of part of the Supplemental Fig. 14 legend is copied here below, and you may like to go to the figure for a better understanding:

“...The training dataset is the same as Fig. 2. As preparation, raw images were processed by 3D ILEE. To mostly simulate the shape of actin, we directly adapted the skeletonized images (I_{sk}) generated from 3D ILEE as the central line of new filaments but filled all voxels with novel values. The artificial images comprise three brightness fractions: artificial actin, artificial ground noise, and artificial diffraction noise. First, referring to the binary images of the training samples, for each voxel (I) in the filament, we found its referred voxel (I_R), the nearest voxel to I in the filament skeleton. Next, we calculated the D_R (the Euclidian distance between I and I_R) and D_{R2N} (the Euclidian distance from I_R to the nearest negative voxel) of all voxels that belongs to the filament (also see **b**) and constructed a polynomial regression model (also see **c**) between the relative distance to I_R (i.e., D_R/D_{R2N}) and the mean (μ_{atf}) as well as the STD (σ_{atf}) of relative brightness (i.e., I/I_R) for all on-filament voxels for all images. To generate the new artificial actin fraction, we refilled the on-filament voxel by the normal distribution $I/I_R \sim N(\mu_{atf}, \sigma_{atf}^2)$ based on the brightness of voxels belonging to the corresponding I_{sk} , and we abandoned the voxels with $D_R/D_{R2N} > 1.4$ because they are more likely to be the structural noise – the cytosol and PM with fluorescence markers not binding to the actin. Second, for the ground noise fraction, we had different strategies to generate two groups of voxels. For the volume previously belonging to the coarse background, we directly apply a fold of change that is subject to a normal distribution to each voxel; for the other volume beyond coarse background, we gave random values that subject to a Burr-12 distribution fitted by voxels of the coarse background, thereby all voxels in the new artificial image were given with a ground noise value. Third, the volume previous belonging to neither coarse background nor real actin is considered as the space impacted by the diffraction (Fig. 2, **a**). Using the raw data of this region, we fitted an exponential distribution to describe the diffraction signal and refilled the diffraction region by this distribution. The choice of Burr-12 and exponential distribution is based on the best performance over a fitting test over 60 statistical models. Finally, we add the three fractions of brightness together and linearly align the value range to 0-4095 (12-bit)...”

However, although here we newly added the note to refer to Supplemental Fig. 14, we would not like to move this detailed description to the *Results* or *Methods* for an overall consideration. While we totally understand such detail is very helpful to peer reviewers and readers with computer vision background, it may be non-important contents for some readers from the cell biology side. Therefore, we would still like to leave the choice

readers and direct the advanced readers to the supplemental figure for more detail.

Reviewer #2 (Comments to the Authors (Required)):

In this work, Li et al. describe a pipeline to segment cytoskeletal filaments in fluorescence images, and then to quantitatively characterize their organization using various "cytoskeleton indices". The segmentation approach is based on global and local thresholding, and employs intensity as well as intensity derivative information. The segmented binary image is skeletonized in order to calculate the cytoskeleton indices. The pipeline is applicable in both 2D and 3D, which is in principle a useful feature.

I do have major concerns about this work. The manuscript as written is difficult to follow. The analysis pipeline seems quite complicated with many steps that are often not justified well other than that they "work." There are many seemingly arbitrary procedures. The comparison to other algorithms is incomplete and does not demonstrate strong superiority of ILEE. The work does not mention or cite many of the more recent advances in the field of segmenting curvilinear structures in images.

Re #23:

Thank you for your comments. After a thorough reading of your comments and rigorous consideration, we believe they generally make good sense and indeed reflect certain weaknesses of our manuscript. Therefore, we highly appreciate your comments for our improvement and will correct/renew the corresponding contents. We are responding to each of the specific concerns as below:

Manuscript itself:

There is quite some jargon. Terms, including acronyms, are often introduced without proper definition, making it difficult to follow what exactly is done. Various analysis steps are described without proper rationale. The results section becomes (somewhat) easier to follow after reading the detailed methods, but this is not the order in which the manuscript will be read.

Many sentences are difficult to understand. For example: Lines 467-468: The authors write "Next, we attempted to understand whether different indices of the same class, particularly density and bundling, can reflect the quantitative level of the class in accordance, or instead show inconsistency." I think I get what the authors are trying to say, but please have someone read the manuscript and give you feedback on clarity!!!

Re #24:

Thank you for the feedback. We inspected the total text and indeed agree with your opinion. To polish the language for better delivery, we first checked and edited all definitions of concept to make sure they are properly described using clear and easy language at accessible position. After we re-edit the general languages first by ourselves, we then invited two scientists to help assess the accessibility of the language. To ensure our language of friendly for all potential readers, their areas are (1) only cell/molecular biology without background of computer vision, and (2) computer vision without knowledge of biology, respectively. The two volunteering peers are very insightful and responsible with a lot of significant efforts. According to their feedback, languages of concerns are further improved. We believe the revised paper has an essentially improved delivery.

Segmentation pipeline:

As mentioned above, there are many steps in their pipeline without strong rationale and sometimes operating as a black box. Examples:

In the method for determining the "course background," I cannot find any reference for the "non-connected negative element scanning" (NNES) algorithm used to estimate it. How the authors then go from the "peak" of the NNES curve to the presumed optimal threshold (Fig. 2) feels like a black box. The "straight-line fit" employed in Suppl. Fig. 2 is not convincing. In this section, the authors also mention a "peak-of-frequency brightness" as an alternative method (line 202). What is this?

Re #25:

Thank you and sorry for your confusion. Previously we thought if we include too much algorithm detail in the *Results* section, readers (especially those of only biological background) may feel overwhelmed and confused, we somehow simplified the description of "the content in the black box" and move them to supplemental figures and *Methods*. However, we do realize this may cause additional confusion and we are sorry for these problems.

Therefore, to make sure the "black boxes" are easily understandable while not perplexing, we added simple descriptions on the rationale and mechanism of each of our original algorithms. Meanwhile, we added more explanatory contents in the *Methods* to help understand the rationale.

For NNES, it is actually one of our original algorithms developed particularly for this paper, so there is no reference. This is because we have tried several direct approaches to extract a "feature value" of the coarse background, but none of them can get stable or estimable results, just like the example we have shown in Supplemental Fig. 1c (now called Supplemental Fig. 1a in the revised version). However, we realized that no matter how the brightness distribution of background and filament changes among different image samples, a nature of the background noise remains that all pixels are independent, which explains why the background noise is always subject to a normal-like distribution (or Burr12 distribution, if more accurately). That means, in the background, two adjacent pixels would mostly like to fall to different value (one 0 and another 1) if we binarize the image with a threshold of the peak (mean). Conversely, if we find a binarization threshold that renders the most non-connected negative element (or count of "reversed connected component" in the language of computer vision), it should reflect the real peak of background noise distribution, which can be used as the feature value. After we tried this algorithm, we got smooth curve of NNE distribution, which means the peak is callable, so we stuck to this approach.

The NNES section, line 191-213, is re-written to provide the rationale of this algorithm with a straightforward logic. To serve this purpose, the subplot order of Supplemental Fig.1 is adjusted, and the legend is re-written for better delivery.

In the procedure to determine g_thres , the line and curve fits in Suppl. Fig. 7 are not convincing.

Re #26:

Due to lack of further explanation, we are unfortunately not sure why the reviewer believed this is not convincing. But we may guess at our best criticism from the perspective of a reviewer and further explain why we choose this model.

The reviewer might have thought that there are other models to fit the correlation even with better R value, such as simple linear regression. However, we believe a good model in this scenario must be engineered upon these three considerations:

- (1) the more turbulent (higher $\sigma_{G.cbq}$) is the coarse background of the gradient image, the higher gradient threshold should be set to make sure the selected edge is clean.
- (2) $k(\sigma_{cbq})$ and $\sigma_{G.cbq}$ must pass origin point, because we are developing a solution for the total community, and their confocal image may have less gradients in the background beyond our tested samples but we should be ready to process such data. Although we had particularly used training dataset with high variation to cover most of the scenarios, images with less background gradients will still be generated due to advanced microscopy or dye technology that renders better signal-to-noise ratio.
- (3) for ILEE, including false positive edge is actually more devastating to the final accuracy than false negative, because a selected low brightness pixel will cause a low neighbor area in the I_{thres} . Therefore, the model needs to be relatively conservative when selecting the high gradient area.

For these reasons, we decided on training the non-linear model $g_{thres} = \mu_{G.cbq} + k(\sigma_{cbq}) \sigma_{G.cbq}$ that has both satisfactory compatibility and accuracy during our tests, which reaches our requirements above.

The authors correctly state that segmenting both thin and thick filaments at the same time is challenging. Thus, this is an important problem that the authors are working on. However, why do they only employ 2 values of K, a small one for "thin" filaments and a larger one for "thick" filaments? In reality, the filaments most likely have a continuum of thickness, and I am not sure why just 2 values of K will be sufficient to cover the full thickness range. Also, how the authors estimate K2, described on lines 272-279, is rather heuristic.

Re #27:

From a purely technical perspective, it is indeed true that one could have use many Ks. However, we believe using 2 Ks is most appropriate upon three major reasons: (1) the thickness of cytoskeleton has certain range based on biology and cannot be unlimitedly contrasting. Through our tests, we found that the results by K1 and K2 mostly cover each other (as shown in Supplemental Figure 8) and 2Ks are already enough to cover all filaments with the continuum of thickness. (2) extra Ks will may fail to match with any K selection logic/criteria. While K1 is based on theoretical thinnest filament and K2 is based on thickest filament among all samples in a batch (described in line 268-281, line 682-791, old version), there is no reference or criteria to decide any extra K, so we did not find a good reason for more Ks. (3) each round of ILEE linear solver (using one K) consumes certain time, and more Ks means longer time and more computational power required. This is extremely obvious in 3D mode, so on the contrary, we provide a single K mode for the 3D pipeline even it means minor sacrifice on accuracy.

However, just like what you indicated, we cannot absolutely promise that using 2Ks always works on all kinds of image sample, especially for those beyond of experience. We do encourage users to send data that generate strange results to us for diagnose and bug-fixing. If we did receive valid evidence suggesting 2Ks are not covering in certain types of samples, we could add new functions with more Ks to the library for users meeting such issue.

This question is also addressed by review #1, which is discussed in Re #5, as a potential reference.

Cytoskeleton indices:

After segmentation, to describe the segmented cytoskeleton, the authors calculate 12 indices, but then only focus on 9 because "they are normalized and ready for biological interpretation" (lines 289-290). The other 3 should either be made biologically interpretable or just not get mentioned. It is also not clear to me why some of the indices require image oversampling. And if they indeed require oversampling, then why is oversampling done in 2D but not in 3D?

Re #28:

Sorry for these confusions. For the three indices that are not mentioned, we did realize that they make some confusion, but we believe it is better to include this information properly because they may be useful to advanced users who intend to develop original methods (e.g., new indices), which potentially benefits the community of this area. We apologize for omitting this information. To eliminate such confusion, we added the description in line 318-321 of the revised manuscript; they were also mentioned in the Fig. 1 legend.

For oversampling, we indeed found we did not put this clear. We deliberately omit this because we do not want to burden readers from pure biological background. The original image is always the "truth", but there are additional considerations:

(1) some indices require topological structure (e.g., linear density, branching), so the original thin filaments may get minorly inaccurate results, so it is better to oversample the data;

(2) for 3D images particularly, because of the optical nature of any confocal system that x-y unit and z unit is different, the image must be stretched back to 1:1:1 unit length (cubic voxel instead of cuboid) for computation of most of the indices. So, we necessarily oversampled the z-axis (800*800*25 to 800*800*97 in our case).

(3) processing the topological image of an oversampled 3D image (800*800*97 to 2400*2400*220, as a general scenario) to enhance topological structure consumes too much computational power (CPU/GPU and memory) for index computation, which is not available for most of the current PCs, so we excluded this in the 3D pipeline. We understand this may lead to a certain level of confusion for readers in a computer vision background, so we added a clarification in line 323-327.

The difference between diameter_TDT and diameter_SDT is not clear.

Re #29:

Thank you for pointing out. To clarify this issue, we added a brief description (see line 332-334 in the revised version) and refer to the details in the Methods for interested readers (line 844-858). We also found and corrected an error describing the math of these two indices in Methods and have already corrected them.

I am not convinced by the "static severing activity" index. The authors state that "a severing event generates two visible cytoskeleton filaments." This is of course true, but how do they assess this in static images? If there are two filaments aligned head-to-tail, how do you know in a static image whether these two filaments were generated by severing vs. by two nucleation events?

Re #30:

Absolutely agreed. This is indeed our critical mistake. Severing and nucleation events cannot be distinguished by static imaging approaches. Therefore, we would like to rename this index to "Segment density", and we have totally rewritten/edited all contents that involve severing computation (see line 336-370, plus other places using the wrong nomenclature).

Again, thank you for pointing out this problem.

Assessment and comparisons:

As far as I can tell, none of the other algorithms that ILEE is compared to in Fig. 4 employ derivative information to identify curvilinear structures in an image. They seem to be different ways of thresholding the raw intensity. This is far from the state of the art in this field. The authors do not introduce or discuss key concepts in curvilinear object detection, such as steerable edge and ridge filters based on first and second order derivatives, as well as nonmaximum suppression, which is a more accurate procedure to determine the detected line centers than skeletonization (as is done in this work). See e.g. Canny IEEE PAMI 1986, Jacob & Unser IEEE PAMI 2004, Puspoki et al. IEEE TIP 2016, Gan et al. Cell Systems 2016, Kittisopikul et al. Bioinformatics 2020. These

concepts should be at least discussed, and even better the performance of ILEE should be compared to pipelines based on some of these more advanced concepts.

Re #31:

Agreed. Please let me further explain about this issue. It is true that the algorithms that we compared are mostly raw intensity-based, and we did know there are some “state of the art” algorithms that conducts image segmentation using edge and ridge based on first or second order derivative. However, during our exploration, we found these approaches are not very accurate for cytoskeleton and are generally difficult to handle, basically because they are originally designed for photos where different component generally have a sharp edge, unlike live image of cytoskeleton. This is potentially an important reason why most researcher in this field still use MGT so far. However, we still inspected the four papers that the reviewer recommended. They are good papers, which is related to, but not necessarily facilitating our study:

- (Canny IEEE PAMI 1986, Jacob & Unser IEEE PAMI 2004): they are discussing about the famous Canny-based approaches, which detects a single-pixel edge (not the same as the “edge” during ILEE) and cannot directly generate the whole segment (i.e., binary image). It is possible, however, to enable this algorithm for our purpose by filling the edge it detects. So, we decided to give it a try (discussed below).
- (Puspoki et al. IEEE TIP 2016): they are talking about an application of established steerable filter, edge detection, and mostly calculating the directionality of biological/biomedical images, but unfortunately not too much of new image segmentation (not only the edge) algorithms like ILEE. It seems that our paper and this one serves for different purposes.
- (Gan et al. Cell Systems 2016): this is a biological paper whose focus is proving a co-functionality of IF and microtubule, where algorithm is not a focus. The novelty of their algorithm mainly focuses on filament directionality, which may help with our anisotropy indices in the future, but this is not our focus. They do mention their image segmentation approach, which is very interesting because it is from ridge to the full element as a different strategy, but it is somehow disappointing that they seemed to believe this is not important because they do not mention anything else (reference, library...) except for it is called “iterative graph matching”.
- (Kittisopikul et al. Bioinformatics 2020): an algorithm paper focusing on directionality of biological curvilinear structures. Just like Gan et al., 2016, a major difference between this publication vs ours is that their calculation is based on the (1-pixel-wide) skeleton image, rather the binary image with filament thickness. Segmentation is actually conducted but not described as an unimportant issue. However, we downloaded their developed MATLAB interface and found actually they are using Otsu for initial segmentation, which we proved is an inferior approach for challenging image samples. However, the paper actually shows great strategy and superior performance of their ridge-based “skeletonization” algorithm, which may substitute our classic skeletonization approach – rather than ILEE, the segmentation approach. We are willing to put this into our plan for future of update release.

In summary of these 5 aforementioned papers, the state-of-the-art practice of the steerable filters can be used for biological filament computation, but they mostly bypassed the filament segmentation to directly focus on the skeleton image identified through ridge analysis. While this may still contribute to a part of our indices, they serve different research aims. Therefore, we cannot direct use such algorithms for comparison.

To exactly answer the reviewer’s question whether such steerable edge detection approach can be used for cytoskeleton segmentation, we designed and optimized the pipeline testing two approaches for filament segmentation: Canny, a classic edge detection algorithm as mentioned by the reviewer, and Active Contour Without Edge (ACWE), a well-developed high-performance approach that is very popular in computer vision (see Reviewer Responding Material 2 in the end of this file following the responding letter; details are described in the legend). The results are within our expectation, canny shows one of the worst performances while ACWE

has a mediocre performance but still slightly inaccurate comparing to MGT and Li's method, not to mention ILEE, even though all parameters are optimized to the best performance. However, we cannot exclude the possibility that these strategies can reach good performance if re-designed to with novel algorithms, but such effort is beyond this publication.

Since ILEE indeed uses both first and second derivatives of the image, we agree with the reviewer that it is indeed appropriate to introduce/discuss the state-of-the art application of steer filter for biological filaments in the article, which is now added to line 116-131 of the revised manuscript. However, according to the result of our additional tests (Reviewer Responding Material-2), we assume it is not necessary to add them to the results of this paper. While one reason is they do not show good performance when applied to the cytoskeleton, we also suspect that readers with biology background only may feel confounded if too many concepts are introduced.

The authors also do not state anywhere whether they implemented the algorithms to which ILEE is compared, or whether they employed software written by others.

Re #32:

Sorry for this confusion and thank you for pointing out. Now, we have added an additional sub-section in the *Methods* called *API reference of libraries* (see line 971-977), leading to a supplemental file "S2. API references.docx" that describes the libraries we used for this study. Accordingly, all aforementioned algorithms are now referred to the web links by the index number.

In evaluating the segmentation, the use of the terms "pseudo-positive" and "pseudo-negative" is rather unusual. Why not use the common terms "false positive" and "false negative"? Also, the "actin-like pseudo-positive" category is not justified - if the ground truth is not good enough, then it cannot serve as the ground truth. More problematically, the procedure for classifying segmentations as "actin-like pseudo-positive" is not explained, and in fact, in looking at the segmentations of Fig. 4, those segmentations do not look very actin-like.

Re #33:

For the term names, we assume there is not too much difference between "pseudo-" vs "false-". We searched Google Scholar and found "pseudo-positive/negative" are also generally used in academic journals. Since this may be a neutral issue, we would like to consult to the editor whether changing all terms to "false-" is preferred.

For "actin-like pseudo-positive", we are very sorry for omitting description of classifying actin-like pseudo-positive and appreciate for pointing out. We believe the most important difference between a piece of actin vs a clot of noise is the shape: actin tends to be thinner and longer while noise clots are dot-like. Therefore, we set rules regarding their parameter, skeleton-length, and connectivity to true actin. Basically, the segment should have a skeleton length/perimeter value greater than 0.3 (approximately thinner than a 1:1.5 rectangle); if the segment connects to any matched pixels, then we have an additional criterion: the segment containing the pseudo-positive should have higher a skeleton length as well as the skeleton length/perimeter, comparing to the segment excluding these pixels. It is noteworthy that we understand this criterion include some level of subjective opinion (i.e., the 0.3 threshold can be set to other value), but so far, this is a relatively fair way to estimate whether a pseudo-positive segment is noise or omitted true actin, according to our knowledge. We have added this information to *Methods*, see line 895-901, and referred to the *Methods* in the corresponding section of *Results*, see line 384-386.

According to our 10+ year lab experience in plant live actin imaging, we found these actin-like segmentations have shapes that is very common for live plant epidermis actin, and we had chosen a typical image for Fig. 4a that has diverged thin/thick, bright/faint, and straight/curvy filaments, to illustrate how challenging this work is.

Animal actin and plant fixed actin do have some nuance versus plant live actin, which may deliver an impression that our shape is “strange” (but actually not).

As for the issue that “ground truth is not good enough”, we would like to argue that it is technically impossible to obtain an “absolute ground truth”. The ground truth is generated by human visual recognition through drawing pixel by pixel by an expert with 6-year experience working on actin and inspected by peers. Among similar types of study, this is a widely used approach, and perhaps the only relatively accurate way to get “ground truth”. Plus, we are handling the most challenging plant live samples whose actin is actively moving and the GFP marker is permeating, so certain level of anticipation is required over minor blurs. Therefore, the “ground truth” inevitably suffers from minor subjective error and it is impossible to claim that the “ground truth” is always truer than ILEE. This is also the reason we try classifying “actin-like pseudo-positive” as an investigation over ground truth. Still, this is the ground truth at our best; if the using such ground truth was not allowed, there would not be practical approaches to develop any cytoskeleton recognition algorithms.

In terms of ILEE's performance, specifically regarding the calculated cytoskeletal indices (Fig. 4d), ILEE's improved performance is heterogeneous. It systematically underestimates three of the 9 indices. Its error bars are comparable to those of MGT and Li, except for the severing activity index where MGT and Li have higher error bars, but in fact at the same time they have less systematic error. So it is not clear which method is really better.

Re #34:

We do agree that ILEE's improved performance is heterogeneous with three indices systematically underestimated if only looking at the mean compared with ground truth; however, we believe this cannot be simply interpreted as ILEE is not accurate. As we are well concerned of this phenomenon, we discussed it in an entire subsection, see line 582-602 of the new version. The abnormality of these three indices (i.e., *linear density*, *static severing activity/segment density*, and *static branching activity*) are majorly due to their high dependence on the topological structure of the actin (i.e., I_{sk}). Unfortunately, the topological structure, especially the connectivity pattern between two adjacent filament that is very near, is very sensitive to the subjective judgment by the maker of ground truth. Therefore, as we had discussed, currently there is no conclusive approach to determine whether it is ILEE or “ground truth” that is the most accurate reflection of the topological structure and the absolute accuracy of the three indices specifically, and this systemic difference should not be interpreted as “systemic error”.

However, it is very certain that, even including these three indices, ILEE provide a huge boost on stability, which evidently benefits most users whose goal is to explore biological questions by comparing experiment groups vs control. Therefore, we conclude ILEE is generally superior over an overall consideration on its accuracy (Fig. 4), stability (Supplemental Fig. 10), and robustness (Supplemental Fig. 11-13). To better help understand our conclusion, we summarized each item of its performance as a table. Please see **Reviewer Responding Material 3** in the end of this file, following the responding letter.

For these reasons, we do believe ILEE is a prior choice. Even if ILEE cannot reach the “best” simultaneously for all these individual items, it is still the best-over-all. It is true that other single algorithms may dominate a few items but are likely to have horrible performance on other items at the same time. However, we do agree that Li's approach is somehow satisfactory among classic algorithms even if it is below ILEE. As modularization and easy customization our library's important feature, we can also transplant Li's approach to our library's 2D pipe in future releases, as an alternative choice for users who are interested.

More minor comments:

In the introduction, the authors state that "shared by plants and animals are two types of cytoskeleton, actin and microtubule." What about intermediate filaments?

Re #35:

Previously, intermediate filament is believed as animal-exclusive; so far, we did not find any identical/typical IF but some cases showed existence of partial-homologs of IF in plant. Doubts exist as their function may be somehow different to call them "intermediate filament". However, we agree to edit these descriptions to be more instructive and accurate. See line 54-60.

Lines 266-268: The authors state "Since the theoretical minimum thickness of a distinguishable cytoskeletal component is approximately equal to one pixel/voxel unit". Well, this depends on the pixel size vs. the microscope resolution. With proper spatial sampling, the minimum thickness hopefully covers about 3 pixels.

Re #36:

Thank you and we agree. We have edited the language to be more accurate, see line 291-297. We will also articulate in our library documentation that users may change K1 when their resolution is extremely high.

Line 355: The authors state that "the adaptive global thresholding approaches (from Otsu to Yan) tend to be relatively accurate when judging the thick and bright bundles." I disagree with the authors' judgment regarding the performance of the Yan approach (Fig. 4a). As their quantitative analysis shows, it has many false negatives.

Re #37:

Thank you for pointing out. We have changed the description as "able to detect some thick and bright filaments", see line 391-392.

Fig. 5c: what are the p's shown on each graph?

Re #38:

Sorry for omitting the clarification. "p" represents the possibility to reject the non-hypothesis of no existent correlation, which is generally regard as another index to show level of correlation in addition to r. We have added this to the legend.

Fig. 6c-h, l-n: What are the little a's and b's etc. inside the bar graphs?

Re #39:

They are results of multiple comparison, groups with totally different alphabets suggests significant difference. This is a common way to show significant difference among many groups. We confirmed that the description is in the legend.

The text discussing Fig. 6 does not cite the many panels of Fig. 6 as needed.

Re #40:

Sorry for this omission. The information is added accordingly, see line 480, 482, and 491.

Suppl. Fig. 14 is not cited anywhere, and all the Suppl. Figures after it are cited incorrectly (i.e. what the text refers to as Suppl. Fig. 14 is actually Suppl. Fig. 15, etc.).

Re# 41:

Thank you for pointing out. The citations after Supplemental Fig. 14 are now corrected.

Distribution of Top 5% DTs of Fig. 5 dataset

Distribution of K2s of Fig. 5 dataset

Reviewer Responding Material 1: Data distribution of individual K2s estimated by top 5% DTs.

We used the random 31 samples of Fig. 5 to show an example of the general data distribution of top 5% DTs as well as K2s, shown as violin plot plus single data points. As the distribution concentrates to the mean, with a normal-like shape, we believe it is available to use the mean of top 5% DT as a manner to estimate optimal K2.

Reviewer Responding Material 2: Performance of “Edge-based” segmentation algorithms.

Two commonly-used edge-based segmentation algorithms, Canny and Active Contours Without Edges (ACWE) are introduced and developed into four methods for binary image of cytoskeleton. The total experiment scheme is the same as Fig. 4a and the comparison map is shown. The parameters of all algorithms are already optimized to their best performance. For Canny, because this algorithm *per se* can only determine edge without finding the connected component (pieces of filament), we filled all closed contour to obtain the binary image. As result, Canny is almost a failure because most of the edges it found are not closed contours, therefore unable to be recognized as a piece of filament. For this reason, we also tried the closing algorithm following Canny. However, it generates a huge clot that destroy the filament structure, even at the minimum intensity of closing only 1 pixel (as shown), so Canny is not likely to recognized filaments accurately without a complex downstream processing. For ACWE, we compared the best result vs the converged result. The best result takes only 2 iterations, and the performance style is similar to MGT or Li method, yet with more pseudo-positive and pseudo-negative results. However, if we keep iterating until the image is converged (theoretically “most confident”), it failed detect most of the filaments. Therefore, both commonly used edge-based methods cannot accurately detect cytoskeleton and performed worse than ILEE, MGT, or Li. We assume this is because the confocal image of live cytoskeleton does not have a contrast change of brightness (high gradient) at their physical boarder, which is totally different from traditional photos, for which these algorithms are originally developed.

Additional Library API used for this study:

ACWE:

https://scikit-image.org/docs/dev/api/skimage.segmentation.html?highlight=morphological_chan_vese#skimage.segmentation.morphological_chan_vese

Canny:

<https://scikit-image.org/docs/dev/api/skimage.feature.html#skimage.feature.canny>

Closing:

https://docs.scipy.org/doc/scipy/reference/generated/scipy.ndimage.binary_closing.html

Criteria	Occupancy	Linear density	Diameter (TDT)	Diameter (SDT)	Skewness	CV	Segmentation density	Branching activity	Local anisotropy
Accuracy (Absolute accuracy)	Best	Uncertain, ground truth bias*	Great	Great	Great	Great	Uncertain, ground truth bias*	Uncertain, ground truth bias*	Good
Stability (Statistical power for comparison)	Best	Best	Best	Best	Best	Best	Good	Best	Best
Robustness (tolerating noise)	Best	Best	Best	Best	Worst, index feature**	Worst, index feature**	Best	Best	Best

Reviewer Responding Material 3: Itemized summary of 2D_ILEE performance among all approaches.

“Best” means literally the highest performance; “Great” means in the top-tier and comparable to the best; “good” means satisfactory performance and still above the average of all algorithm. *, Unable to reach conclusion because ground truth may not be absolute accurate on some challenging topological structure. **, the index *per se* – by its definition – is very sensitive to gaussian noise; it is the lease robust because it is accurate, while other algorithms introduces many false pixels that neutralize the change of these indices.

August 29, 2022

Re: JCB manuscript #202203024R

Dr. Pai Li
Michigan State University
Department of Plant, Soil and Microbial Sciences
1066 Bogue St.
East Lansing, Michigan 48824

Dear Dr. Li,

Thank you for submitting your revised manuscript entitled "ILEE: Algorithms and Toolbox for Unguided and Accurate Quantitative Analysis of Cytoskeletal Images." The revised manuscript has been seen by Reviewer #2 whose full comments are appended below. While the reviewer is positive about the work in terms of its suitability for JCB, some important issues remain that need to be addressed before we can move forward.

You will see that the reviewer raises a number of points asking for additional clarification of your model fitting, the algorithm description, and several other text edits requests. We agree that these comments are important and should be fully addressed by revisions to the manuscript rather than a rebuttal.

Our general policy is that papers are considered through only one revision cycle; however, given that the suggested changes are relatively minor we are open to one additional short round of revision. Please note that although we expect to make a final decision without additional reviewer input, the editors will look carefully over the responses and changes before recommending acceptance.

JCB formatting guidelines allow for a maximum of 10 main figures and 5 supplemental figures. You currently exceed the supplemental limit but we do understand that some papers require more supplemental figures than others. In this case, we will be able to give you the extra space, but since you only have 6 main figures currently we ask that you please move some of the supplemental data to the main figures and also consolidate the remaining supplemental materials as much as possible. Each figure can be full page length.

Please submit the final revision within one month, along with a cover letter that includes a point by point response to the remaining reviewer comments.

Thank you for this interesting contribution to Journal of Cell Biology. You can contact me or the scientific editor listed below at the journal office with any questions, cellbio@rockefeller.edu or call (212) 327-8588.

Sincerely,

Alex Mogilner, PhD
Monitoring Editor
Journal of Cell Biology

Dan Simon, PhD
Scientific Editor
Journal of Cell Biology

Reviewer #2 (Comments to the Authors (Required)):

I thank the reviewers for improving the clarity of their manuscript (although more could be done), and for addressing many of my concerns.

I still do have one major concern, namely the model fits in Suppl. Fig. 7, and in fact similarly in Suppl. Fig. 2b. The authors did get the main idea behind my previous comment that the fits are not convincing. Specifically, the model is fitted to a much wider range of values than the actual data used for fitting. In Fig. 2b, the scatter of points is nothing but an unstructured "blob," and the fit is primarily driven by the authors' assumptions. This is largely the case in Suppl. Fig. 7, especially in c and d. In fact, in panel c, the scatter of data seems to have nothing to do with the fitted model. Again, the fits are largely based on the assumptions that the authors stated in their response to the reviews, through which they designed the model that they fitted. These assumptions

must be stated clearly in the manuscript (I apologize if they are stated so clearly and I missed them), as they are critical and I believe they influence the fitted model more than the actual data do. To justify their choice of model, the authors state that they "are developing a solution for the total community, and their confocal image may have less gradients in the background beyond our tested samples but we should be ready to process such data." But without taking some new dataset with a dramatically different background and showing that their algorithm works without any additional tuning, this is for now merely an assumption that the algorithm is applicable to images of a different nature.

I appreciate the clarification and better explanation of the NNE algorithm. A "minor" point regarding the new text. The authors state that "NNE generates a normal-like curve ..." (lines 222-223). While the rest of the sentence is warranted, that curve is far from normal-like. This is not relevant to the point anyway, and there is no need for the "normal-like" description.

I also thank the reviewers for incorporating some of the literature on steerable filters for edges and ridges in their manuscript. I do believe that they are missing key citations though in this regard by e.g. not citing work from the Unser lab, especially the Jacob & Unser 2004 paper. That paper involves more algorithm development than the Gan 2016 paper that they decided to cite, for example. The authors' statement that "this strategy cannot support the computation of filament thickness" should be however toned down. After all, the thickness information is implicitly there (in the filter design) and could be utilized with appropriate algorithmic developments.

But the authors missed one of my main points, regarding the concept of non-maximum suppression, which is a more robust way to "skeletonize" a collection of curved lines in an image by using directionality information (this generally avoids spurious skeleton branching around thick filaments). I brought up the Canny 1986 publication because it was the first to develop this concept (I think), and not because of the gradient filter per se.

I agree that there is no need to add the comparison to Canny and ACWE to the manuscript.

The term "segment density" is indeed more accurate than "static severing activity." However, the authors changed the term in some places but not in others. Please go over the manuscript and make sure this change is done everywhere.

I disagree with using the terms "pseudo-positive" and "pseudo-negative". "Pseudo" gives the impression that something may be not 100% correct, but it is not 100% wrong either. "False," on the other hand, is a much less ambiguous term. Maybe the "actin-like false positives" are "pseudo-positives"?

Line 399: "imaging algorithms": I assume you mean "image analysis algorithms."

Lines 417-418: "the performance of these two algorithms among diverse and complex biological samples was not as stable as ILEE." Is this deduced from the standard deviations in Fig.4 ? If so, please indicate this, to justify your statement.

Fig. 5c. What test is used to generate the displayed p-values? Also, it is "null hypothesis" and not "non-hypothesis".

Response to Reviewers

We greatly appreciate the very critical and constructive advice by the reviewers and the editorial board. Your efforts significantly enable us to considerably improve our manuscript and contribute to the community with a better ILEE toolbox. Herein, we would like to provide a point-by-point response to the proposed issues; the corresponding edits have been applied to the revised manuscript.

In order to address some of the issues, we generated new data, including two Supplemental Figures (Supplemental Fig. 2-2, 7-2), which have already been added to the independent S1 PDF file and also copied to the end of this file for your convenience.

The original reviewer's comments are shown in **black**, and our responses are shown in **green**.

Reviewer #2 (Comments to the Authors (Required)):

I thank the reviewers for improving the clarity of their manuscript (although more could be done), and for addressing many of my concerns.

I still do have one major concern, namely the model fits in Suppl. Fig. 7, and in fact similarly in Suppl. Fig. 2b. The authors did get the main idea behind my previous comment that the fits are not convincing. Specifically, the model is fitted to a much wider range of values than the actual data used for fitting. In Fig. 2b, the scatter of points is nothing but an unstructured "blob," and the fit is primarily driven by the authors' assumptions. This is largely the case in Suppl. Fig. 7, especially in c and d. In fact, in panel c, the scatter of data seems to have nothing to do with the fitted model. Again, the fits are largely based on the assumptions that the authors stated in their response to the reviews, through which they designed the model that they fitted. These assumptions must be stated clearly in the manuscript (I apologize if they are stated so clearly and I missed them), as they are critical and I believe they influence the fitted model more than the actual data do. To justify their choice of model, the authors state that they "are developing a solution for the total community, and their confocal image may have less gradients in the background beyond our tested samples but we should be ready to process such data." But without taking some new dataset with a dramatically different background and showing that their algorithm works without any additional tuning, this is for now merely an assumption that the algorithm is applicable to images of a different nature.

Re #1:

Thank you for pointing out a very important issue. You are absolute right that the model fitting results depended more on the assumed model than the data and that our training samples have limited variation, as shown in previous Supplemental Fig. 2b and Supplemental Fig. 7c, d. So, we investigated this issue in detail and put great efforts into improving both models by introducing an additional dataset and through image processing. We also edited the paper according to your suggestion. At the same time, we also verified that the quality of our results is not too sensitive to the parameters these models led to. We provide a more detailed explanation in the following:

The reason that these plots are like "blobs" – the training dataset with limited variation – is that we could not find raw 3D images with varying ground noise patterns, although we had already chosen diverse training samples on purpose. We had chosen different batches of unrelated experiments, but still saw the similar distribution of ground noise brightness, so we had thought this is the nature of cytoskeleton imaging (not necessarily true, though). Since these models produced reasonable results, we had decided to use them in our pipeline.

However, just like what you commented, we should not "anticipate" good applicability without testing truly diverse samples. This time, to solve the problem, we introduced images from another lab (images from a different device by a different researcher) and additionally conducted image augmentation to enhance/reduce the ground noise for a larger, extended dataset. Finally, we got a highly diverse dataset of 153 images (see Supplemental Fig. 2-2a) with MGT-based ground truth to test our model. It turned out that your concerns are valid, as there is a very interesting and noteworthy phenomenon – our previous model fitted with the images from our lab collection (including augmented images, see Supplemental Fig. 2-2b), but failed to generalize to the images from another

lab. However, their samples also seemed to be a proportional function just like ours, but one with a different slope. This suggested that the rationale of the model (previously described in the responding letter of the first resubmission) may be correct, but the proportional coefficient is influenced somehow. After investigation, we found this was because the images from the other lab had some unexpected features. They have “multiple layers of ground noise”. Many cells have a higher baseline level of fluorescence than the others with a cytosolic permissive pattern (see Supplemental Fig. 2-2c), which may be out-of-focus light or excessive fluorescence dye not binding to actin. Therefore, their brightness distribution of ground noise is wider with a lower peak value, where ground noise covers some of the real actin signal. Since NNE peak and brightness frequency peak have the same trend, theoretically, the proportional coefficient should be lowered to exclude the real actin. This discovery inspired us: the question becomes how to extract a feature value representing the degree of concentration of ground noise and how to use it to adjust the proportional coefficient.

After testing several models, we finally created a solution with satisfactory and stable performance (see Supplemental Fig 2-2d) for both sets of data. The basic idea is the same, but we introduced another variable w (i.e., the width between two 80% peaks of brightness frequency; see Supplemental Fig 2-2c) to reflect the degree of centralization of ground noise and its impact on the thresholding strategy. This is a multivariate interaction model, expressed as $t_{cbg} = k_{cbg} \frac{p^m}{w^n}$, where k_{cbg} , m , and n are parameters to be trained. On the one hand, as previously described, a greater NNE peak means higher coarse background threshold; on the other hand, the more dispersed the ground noise, the more cytoskeleton edge voxels are overshadowed, which means we should apply a lower threshold to include adequate edge voxels. This modified model reflects this logic, and we found the best parameters as $t_{cbg} = 22.1408 \frac{p^{1.004}}{w^{0.481}}$, with a very satisfying determination coefficient $R^2 = 0.89046$. Since the new training dataset is much more diverse and widespread (see Supplemental Fig 2-2d), we believe the applicability of the new model is greatly improved and better tested, and may be applied to various 3D images for a relatively accurate result.

A similar strategy is applied to the g_{thres} estimation model (see Supplemental Fig. 7-2). Previously, the model was defined as $g_{thres} = \mu_{G.cbg} + k(\sigma_{cbg}) \sigma_{G.cbg}$. While the logic is to apply more conservative threshold on samples with higher noise (as previously explained the response letter of our first resubmission), our training sample for 3D mode indeed centralized into a “blob”, arousing concerns over the range of applicability over the model (see Supplemental Fig. 7-1c,d). Therefore, we used the same multi-lab augmented datasets as Supplemental Fig. 2-2 to test and improve the model. Likewise, the trend of $\sigma_{cbg} - g_{thres}$ relationship still diverges only as a function of image resource (data not shown), suggesting a similar issue as described above. To solve the problem, we again used w to adjust the estimation through a $\sigma_{cbg} - w$ interaction model $k(\sigma_{cbg}, w) = a_k \frac{\sigma_{cbg}^m}{w^n}$ to replace the old $k(\sigma_{cbg})$ function. At the best fitting, the model got an acceptable determination coefficient $R^2 = 0.513$ (Supplemental Fig. 7-2a). It can be observed that the two image resources formed two blobs, but now it makes better sense to predict over a broader range of input by linear extrapolation, compared to a single blob. When introducing the new k function to $g_{thres} = \mu_{G.cbg} + k(\sigma_{cbg}, w) \sigma_{G.cbg}$, the model displays satisfactory accuracy by Pearson coefficient $r = 0.779$ and false-positive rate $p = 2e - 32$. Besides, we also established an alternative g_{thres} model with even higher performance. We used the simple multivariate linear regression model $g_{thres} = a_1 \mu_{G.cbg} + a_2 \sigma_{G.cbg} + a_3 w$ to fit the dataset and obtained a model with $r = 0.871$ and $p = 2e - 48$. As for the pros and cons over the two models: the $\sigma_{cbg} - w$ interaction model makes better sense of logic over statistics principles, while the simple multivariate linear model provides slightly better accuracy so far. We will offer the users options to decide which model to use. In sum, both of them fix the problem of limited variation of the training dataset, and we believe the g_{thres} estimation is improved by a significant leap forward.

We are going to replace the t_{cbg} and g_{thres} model in the next release of the library, and the newly generated data, Supplemental Fig 2-2 and 7-2 are added to the manuscript supplemental material PDF for a detailed explanation. All the corresponding content related to t_{cbg} and g_{thres} model has been edited (see line 229-233, 278-281, 750-751, 778-780). However, since the previous estimation models are already quite accurate for samples used in this study (leading to reasonable ILEE results), we would like to maintain the data only for this publication and avoid massive reconstruction over all data throughout the manuscript.

On the other hand, we are committed to maintain the library and improve its performance with better parameter estimation models. Since the architecture of the ILEE_CSK pipeline is extremely modularized where each step is relatively independent, it is not difficult to update these estimation models if we discover new limitations according to users' feedback.

I appreciate the clarification and better explanation of the NNE algorithm. A "minor" point regarding the new text. The authors state that "NNE generates a normal-like curve ..." (lines 222-223). While the rest of the sentence is warranted, that curve is far from normal-like. This is not relevant to the point anyway, and there is no need for the "normal-like" description.

Re #2:

Thank you for pointing this out. In fact, our further analysis suggested this is more like a Burr12 distribution rather than a normal one. However, pure ground noise is indeed normal-like (Supplemental Fig. 1b; we also confirmed this in real blank images), which helps justify our estimation model. Therefore, we deleted the "normal-like" description (see line 222), and additionally edited the language in the Supplemental Fig. 1b legend to make it more appropriate.

I also thank the reviewers for incorporating some of the literature on steerable filters for edges and ridges in their manuscript. I do believe that they are missing key citations though in this regard by e.g. not citing work from the Unser lab, especially the Jacob & Unser 2004 paper. That paper involves more algorithm development than the Gan 2016 paper that they decided to cite, for example. The authors' statement that "this strategy cannot support the computation of filament thickness" should be however toned down. After all, the thickness information is implicitly there (in the filter design) and could be utilized with appropriate algorithmic developments.

Re #3:

Agreed. About Jacob & Unser 2004, we meant to say they are not readily optimized for cytoskeletons, but it was exactly wrong to say "this strategy cannot support the computation of filament thickness". We now deleted this statement. Moreover, we believe steerable filter is a crucial concept for our ILEE pipeline as we do use a modified gradient filter (Schar) to determine edges, so we cited Jacob & Unser 2004 in Line 125.

But the authors missed one of my main points, regarding the concept of non-maximum suppression, which is a more robust way to "skeletonize" a collection of curved lines in an image by using directionality information (this generally avoids spurious skeleton branching around thick filaments). I brought up the Canny 1986 publication because it was the first to develop this concept (I think), and not because of the gradient filter per se.

Re #4:

Thank you for pointing out this promising direction for skeletonization. We agree that non-maximum suppression is a more robust way for this task. However, we did not find a volumetric NMS tool directly applicable to our 3D images. Although we had experimented with NMS for the 2D/3D skeletonization, careful development would be beyond the scope of the current paper, which focuses on the ILEE algorithm and the entire framework of our open-source toolbox. Thus, we leave it as future work.

I agree that there is no need to add the comparison to Canny and ACWE to the manuscript.

Re #5:

Thank you and we do not include these data.

The term "segment density" is indeed more accurate than "static severing activity." However, the authors changed the term in some places but not in others. Please go over the manuscript and make sure this change is done everywhere.

Re #6:

Sorry for the previous omission of these errors. We searched this keyword and found them in Line 411, 499, 533, 613, 616, and they are all corrected now. The main and supplemental figures are also inspected again.

I disagree with using the terms "pseudo-positive" and "pseudo-negative". "Pseudo" gives the impression that

something may be not 100% correct, but it is not 100% wrong either. "False," on the other hand, is a much less ambiguous term. Maybe the "actin-like false positives" are "pseudo-positives"?

Re #7:

Thank you for pointing out the changes we previously missed. Now, the terminology has been consistently replaced by "false-positive/negative", including the main text, main figures, and supplemental figures.

Line 399: "imaging algorithms": I assume you mean "image analysis algorithms."

Re #8:

Yes and thank you. It has been corrected.

Lines 417-418: "the performance of these two algorithms among diverse and complex biological samples was not as stable as ILEE." Is this deduced from the standard deviations in Fig.4 ? If so, please indicate this, to justify your statement.

Re #9:

Yes, it is derived from the standard deviation. Now, we adjusted the sentence to clarify this information and make a better connection toward the next paragraph, where we conducted more accurate analyses to describe the stability advantage of ILEE.

Fig. 5c. What test is used to generate the displayed p-values? Also, it is "null hypothesis" and not "non-hypothesis".

Re #10:

We used a function in the Scipy Library (`scipy.stats.pearsonr`; link below) to calculate the p-values of Pearson correlation, which uses the t-test.

Thank you for pointing out the typo. It is now corrected.

Scipy function link:

<https://docs.scipy.org/doc/scipy/reference/generated/scipy.stats.pearsonr.html>

Supplemental Figure 2-2: Enhanced 3D NNE-based adaptive coarse thresholding model for samples with greater diversity

a

b

c

d

Supplementary Figure 2-2. Enhanced 3D NNE-based adaptive coarse thresholding model for samples with greater diversity. As the original training dataset of Supplemental Fig. 2-1 may not have satisfactory variation of threshold-at-peak range, we introduced published actin image samples (Cao et al., 2022) from a different microscope device in a different lab and applied image augmentation to generate a greatly diverse dataset for model training, resulting an enhanced 3D coarse thresholding model. **a**, Scheme of datasets. 31 and 20 random samples, by different microscope machines and photographers, are collected. The total 51 images are then processed through a linear LUT transformation (see formula) to repress ($k_1 = 0.7$) or enhance ($k_1 = 1.3$) the ground noise (see altered brightness distribution, right panel). **b**, the correlation between global threshold at NNE peak (p) and coarse global threshold (t_{cbg}) measure through MGT, of the total training dataset. The black line suggests the “1st edition” model (Supplemental Fig. 2-1) used for the rest of this study, which fits very well with all raw and augmented samples from our lab collection but not with the other dataset. **c**, Differences in brightness distribution between datasets. We inspected both datasets to investigate why they have different slope (see **b**). It is determined that our lab collection have a single-pattern, condensed distribution of ground noise brightness, while the Cao dataset has a wide-spread, dispersive peak. This is because the Cao dataset has multi-layer ground noises, as some of the cells have higher baseline brightness potentially due to out-of-focus light or non-binding actin fluorescence marker. Arrows with different color indicates two background areas with different ground noise level. **d**, Performance of enhanced 3D coarse global threshold model $t_{cbg} = k_{cbg} \frac{p^m}{w^n}$. As the ground noise concentration level have an impact to the relationship between p and t_{cbg} , we measured a second feature value – the distance between two points with 80% top frequency (w ; see **c**) – for modeling improvement. To introduce of w , we constructed a new model $t_{cbg} = k_{cbg} \frac{p^m}{w^n}$, where k_{cbg} , m , and n are trained constants. The left panel shows the correlation between $\frac{p^m}{w^n}$ and t_{cbg} at the best performance ($t_{cbg} = 22.1408 \frac{p^{1.004}}{w^{0.481}}$); the right panel shows the relationship between m , n , and correlation coefficient R^2 . Note: This model will be provided as the default model for 3D t_{cbg} estimation in the PyPI library release but not used for the rest of the study unless specifically announced.

Supplemental Figure 7-2: Enhanced global gradient threshold estimation models for 3D mode.

Supplementary Figure 7-2. Enhanced global gradient threshold estimation model for 3D mode. Similar to Supplemental Fig. 2-1, the training dataset for g_{thres} estimation model for 3D mode (Supplemental Fig. 7-1c, d) may have limited variation. To test and improve the applicability of the model, we introduced a large dataset from a different microscope device by a different lab with image augmentation (see Supplemental Fig. 2-2a) and constructed two enhanced g_{thres} estimation models for 3D mode. **a**, Reconstruction of model $g_{thres} = \mu_{G.cbg} + k \sigma_{G.cbg}$ by introducing w (see Supplemental Fig. 2-2c) to improve compatibility to diverse images. k is replaced into a multi-variant function $k(\sigma_{cbg}, w) = a_k \frac{\sigma_{cbg}^m}{w^n}$ to apply the impact of the dispersion level of ground noise. Therefore, we name the model $g_{thres} = \mu_{G.cbg} + k(\sigma_{cbg}, w) \sigma_{G.cbg}$ as “ σ_{cbg} - w interaction model”. Left panel displays the model with the best parameters; right panel shows the impact of m and n on model performance. **b**, Performance of the σ_{cbg} - w interaction model ($g_{thres} = \mu_{G.cbg} + k(\sigma_{cbg}, w) \sigma_{G.cbg}$) by accuracy. The correlation of the estimated g_{thres} and MGT-determined “ground truth” g_{thres} is measured by Pearson coefficient (r) and P value (p). The the σ_{cbg} - w interaction model has a satisfactory performance with $r = 0.779$. **c**, Performance of the multivariate linear model $g_{thres} = a_1 \mu_{G.cbg} + a_2 \sigma_{G.cbg} + a_3 w$. We constructed an alternative model by a direct multivariate linear regression combining the impact of $\mu_{G.cbg}$, $\sigma_{G.cbg}$, and w . This model has a further improved performance with $r = 0.871$, for users to select according to their demand. The two models are available in the library release with the σ_{cbg} - w interaction model set as default, but they are not used for the rest of the study unless specifically announced.

November 2, 2022

RE: JCB Manuscript #202203024RR

Dr. Pai Li
Michigan State University
Department of Plant, Soil and Microbial Sciences
1066 Bogue St.
East Lansing, Michigan 48824

Dear Dr. Li,

Thank you for submitting your revised manuscript entitled "ILEE: Algorithms and Toolbox for Unguided and Accurate Quantitative Analysis of Cytoskeletal Images." We would be happy to publish your paper in JCB pending final revisions necessary to meet our formatting guidelines (see details below).

A. MANUSCRIPT ORGANIZATION AND FORMATTING:

- 1) Text limits: Character count for Tools is < 40,000, not including spaces. Count includes title page, abstract, introduction, results, discussion, and acknowledgments. Count does not include materials and methods, figure legends, references, tables, or supplemental legends.
- 2) Figure formatting: Tools may have up to 10 main text figures. Scale bars must be present on all microscopy images, including inset magnifications. Please add scale bars to Figures 2a/b, 3a/b, 4a, 5a, S2c, S3c, S8a/b, S11, S17, & S18.
- 3) Statistical analysis: Error bars on graphic representations of numerical data must be clearly described in the figure legend. The number of independent data points (n) represented in a graph must be indicated in the legend. Statistical methods should be explained in full in the materials and methods. For figures presenting pooled data the statistical measure should be defined in the figure legends. Please also be sure to indicate the statistical tests used in each of your experiments (both in the figure legend itself and in a separate methods section) as well as the parameters of the test (for example, if you ran a t-test, please indicate if it was one- or two-sided, etc.). Also, if you used parametric tests, please indicate if the data distribution was tested for normality (and if so, how). If not, you must state something to the effect that "Data distribution was assumed to be normal but this was not formally tested."
- 4) Materials and methods: Should be comprehensive and not simply reference a previous publication for details on how an experiment was performed. Please provide full descriptions (at least in brief) in the text for readers who may not have access to referenced manuscripts. The text should not refer to methods "...as previously described."
- 5) For all cell lines, vectors, constructs/cDNAs, etc. - all genetic material: please include database / vendor ID (e.g., Addgene, ATCC, etc.) or if unavailable, please briefly describe their basic genetic features, even if described in other published work or gifted to you by other investigators (and provide references where appropriate). Please be sure to provide the sequences for all of your oligos: primers, si/shRNA, RNAi, gRNAs, etc. in the materials and methods. You must also indicate in the methods the source, species, and catalog numbers/vendor identifiers (where appropriate) for all of your antibodies, including secondary. If antibodies are not commercial, please add a reference citation if possible.
- 6) Microscope image acquisition: The following information must be provided about the acquisition and processing of images:
 - a. Make and model of microscope
 - b. Type, magnification, and numerical aperture of the objective lenses
 - c. Temperature
 - d. Imaging medium
 - e. Fluorochromes
 - f. Camera make and model
 - g. Acquisition software
 - h. Any software used for image processing subsequent to data acquisition. Please include details and types of operations involved (e.g., type of deconvolution, 3D reconstitutions, surface or volume rendering, gamma adjustments, etc.).

7) References: There is no limit to the number of references cited in a manuscript. References should be cited parenthetically in the text by author and year of publication. Abbreviate the names of journals according to PubMed.

8) Supplemental materials: While Tools generally have up to 5 supplemental figures, we will be able to give you the extra space for 18 supplemental figures. However, as mentioned previously, please adjust the panel sizes in Figures S2 and S7 so that the entire figure fits on a single page. Please also note that tables, like figures, should be provided as individual, editable files. A summary of all supplemental material should appear at the end of the Materials and methods section. Please include one brief sentence per item.

9) eTOC summary: A ~40-50 word summary that describes the context and significance of the findings for a general readership should be included on the title page. The statement should be written in the present tense and refer to the work in the third person. It should begin with "First author name(s) et al..." to match our preferred style.

10) Conflict of interest statement: JCB requires inclusion of a statement in the acknowledgements regarding competing financial interests. If no competing financial interests exist, please include the following statement: "The authors declare no competing financial interests." If competing interests are declared, please follow your statement of these competing interests with the following statement: "The authors declare no further competing financial interests."

11) A separate author contribution section is required following the Acknowledgments in all research manuscripts. All authors should be mentioned and designated by their first and middle initials and full surnames. We encourage use of the CRediT nomenclature (<https://casrai.org/credit/>).

12) ORCID IDs: ORCID IDs are unique identifiers allowing researchers to create a record of their various scholarly contributions in a single place. At resubmission of your final files, please consider providing an ORCID ID for as many contributing authors as possible.

B. FINAL FILES:

Thank you for this interesting contribution, we look forward to publishing your paper in Journal of Cell Biology.

Sincerely,

Alex Mogilner, PhD
Monitoring Editor
Journal of Cell Biology

Dan Simon, PhD
Scientific Editor
Journal of Cell Biology